theoretical biology/statistics/computer modelling and simulation

meta-research, meta-science, reproducibility, bias, knowledge, pseudoscience

**Author for correspondence:**
Daniele Fanelli
e-mail: email@danielefanelli.com

# A theory and methodology to quantify knowledge

## Daniele Fanelli

Department of Methodology, London School of Economics and Political Science, London, UK

DF, 0000-0003-1780-1958

This article proposes quantitative answers to meta-scientific questions including 'how much knowledge is attained by a research field?', 'how rapidly is a field making progress?', 'what is the expected reproducibility of a result?', 'how much knowledge is lost from scientific bias and misconduct?', 'what do we mean by soft science?', and 'what demarcates a pseudoscience?'. Knowledge is suggested to be a system-specific property measured by *K*, a quantity determined by how much of the information contained in an *explanandum* is compressed by an *explanans*, which is composed of an information 'input' and a 'theory/methodology' conditioning factor. This approach is justified on three grounds: (i) *K* is derived from postulating that information is finite and knowledge is information compression; (ii) *K* is compatible and convertible to ordinary measures of effect size and algorithmic complexity; (iii) *K* is physically interpretable as a measure of entropic efficiency. Moreover, the *K* function has useful properties that support its potential as a measure of knowledge. Examples given to illustrate the possible uses of *K* include: the knowledge value of proving Fermat's last theorem; the accuracy of measurements of the mass of the electron; the half life of predictions of solar eclipses; the usefulness of evolutionary models of reproductive skew; the significance of gender differences in personality; the sources of irreproducibility in psychology; the impact of scientific misconduct and questionable research practices; the knowledge value of astrology. Furthermore, measures derived from *K* may complement ordinary meta-analysis and may give rise to a universal classification of sciences and pseudosciences. Simple and memorable mathematical formulae that summarize the theory's key results may find practical uses in meta-research, philosophy and research policy.

## 1. Introduction

A science of science is flourishing in all disciplines and promises to boost discovery on all research fronts [1]. Commonly branded 'meta-science' or 'meta-research', this rapidly expanding

literature of empirical studies, experiments, interventions and theoretical models explicitly aims to take a 'bird's eye view' of science and a decidedly cross-disciplinary approach to studying the scientific method, which is dissected and experimented upon as any other topic of academic inquiry. To fully mature into an independent field, meta-research needs a fully cross-disciplinary, quantitative and operationalizable theory of scientific knowledge—a unifying paradigm that, in simple words, can help tell apart 'good' from 'bad' science.

This article proposes such a meta-scientific theory and methodology. By means of analyses and practical examples, it suggests that a system-specific quantity named 'K' can help answer meta-scientific questions including 'how much knowledge is attained by a research field?', 'how rapidly is a field making progress?', 'what is the expected reproducibility of a result?', 'how much knowledge is lost from scientific bias and misconduct?', 'what do we mean by soft science?', and 'what demarcates a pseudoscience?'.

The theoretical and methodological framework proposed in this article is built upon basic notions of classic and algorithmic information theory, which have been rarely used in a meta-research context. The key innovation introduced is a function that, it will be argued, quantifies the essential phenomenology of knowledge, scientific or otherwise. This approach rests upon a long history of advances made in combining epistemology and information theory. The concept that scientific knowledge consists in pattern encoding can be traced back at least to the polymath and father of positive philosophy August Comte (1798–1857) [2], and the connection between knowledge and information compression *ante litteram* to the writings of Ernst Mach (1838–1916) and his concept of 'economy of thought' [3]. Claude Shannon's theory of communication gave a mathematical language to quantify information [4], whose applications to physical science were soon examined by Léon Brillouin (1889–1969) [5]. The independent works of Solomonoff, Kolmogorov and Chaitin gave rise to algorithmic information theory, which dispenses of the notion of probability in favour of that of complexity and compressibility of strings [6]. The notion of learning as information compression was formalized in Rissanen's minimum description length principle [7], which has fruitful and expanding applications in statistical inference and machine learning [8,9]. From a philosophical perspective, the relation between knowledge and information was explored by Fred Dretske [10], and a computational philosophy of science was elaborated by Paul Thagard [11]. To the best of the author's knowledge, however, the main ideas and formulae presented in this article were never proposed before (see Discussion for further details).

The article is organized as follows. In §2, the core mathematical approach is presented. This verges on a single equation, the K function, whose terms are described in §2.1, and whose derivation and justification are described in §2.2 by a theoretical, a statistical and a physical argument. Section 2.3 explains and discusses properties of the K function. These properties further support the claim that K is a universal quantifier of knowledge, and they lay out the bases for developing a methodology. The methodology is illustrated in §3, which offers practical examples of how the theory may help answer typical meta-research questions. These questions include: how to quantify theoretical and empirical knowledge (§3.1 and 3.2, respectively), how to quantify scientific progress within or across fields (§3.3), how to forecast reproducibility (§3.4), how to estimate the knowledge value of null and negative results (§3.5), how to compare the knowledge costs of bias, misconduct and QRP (§3.6) and how to define a 'soft' science (§3.8) and a pseudoscience (§3.7). These results are expressed in simple and memorable formulae (table 1), and are further summarized in §4, where the theory's predictions, limitations and testability are discussed. The essay's sections make cross-reference to each other but can be read in any order with little loss of comprehensibility.

# 2. Analysis

## 2.1. The quantity of knowledge

At the core of the theory and methodology proposed, which will henceforth be called 'K-theory', is the claim that knowledge is a system-specific property measured by a quantity symbolized by a 'K' and given by the function

$$K(Y^{n_Y}; X^{n_X}, \tau) \equiv \frac{n_Y H(Y) - n_Y H(Y|X, \tau)}{n_Y H(Y) + n_X H(X) - \log p(\tau)} \tag{2.1}$$

in which each term represents a quantify of information. What is information? In a very general and

**Table 1.** $K$ theory's answers to meta-scientific questions.

| question | formula | interpretation | section |
|---|---|---|---|
| How much knowledge is contained in a theoretical system? | $K = h$ | Logico-deductive knowledge is a lossless compression of noise-free systems. Its value is inversely related to complexity and directly related to the extent of domain of application. | 3.1 |
| How much knowledge is contained in an empirical system? | $K = k \times h$ | Empirical knowledge is lossy compression. It is encoded in a theory/methodology whose predictions have a non-zero error. It follows that $K_{\text{empirical}} < K_{\text{theoretical}}$. | 3.2 |
| How much progress is a field making? | $m\Delta X + \Delta\tau < nY\frac{\Delta k}{K}$ | Progress occurs to the extent that explanandum and/or explanatory power expand more than the explanans. This is the essence of consilience. | 3.3 |
| How reproducible is a research finding? | $K_r = KA^{-\lambda \cdot d}$ | The ratio between the $K$ of a study and its replication $K_r$ is an exponentially declining function of the distance between their systems and/or methodologies. | 3.4 |
| What is the value of a null or negative result? | $K_{\text{null}} \leq \frac{h}{\gamma}\log\frac{\|\mathcal{T}\|}{\|\mathcal{T}\|-1}$ | The knowledge yielded by a single conclusive negative result is an exponentially declining function of the total number of hypotheses (theories, methods, explanations or outcomes) $\|\mathcal{T}\|$ that remain untested. | 3.5 |
| What is the cost of research fabrication, falsification, bias and QRP? | $K_{\text{corr}} = K - \frac{h_u}{h_b}B$ | The $K$ corrected for a questioned methodology is inversely proportional to the methodology's relative description length times the bias it generates ($B$). | 3.6 |
| When is a field a pseudoscience? | $K < \frac{h_u}{h_b}B$ | A pseudoscience results from a hyper-biased theory/methodology that produces net negative knowledge. Conversely, a science has $K > B\frac{h_u}{h_b}$. | 3.7 |
| What makes a science 'soft'? | $\frac{k_H}{k_S} > \frac{h_S}{h_H}$ | Compared to a harder science (H), a softer science (S) yields relatively lower knowledge at the cost of relatively more complex theories and methods. | 3.8 |

intuitive sense, information consists in questions we do not have answers to, or, equivalently, it consists in answers to those questions. Any object or event $y$ that has a probability $p(y)$ carries a quantity of information equal to

$$-\log_A p(y) = \log_A \frac{1}{p(y)} \tag{2.2}$$

that quantifies the number of questions with $A$ possible answers that we would need to ask to determine $y$. The logarithm's base, $A$, could have any value, but we will always assume that $A = 2$ and therefore that

information is measured in 'bits', i.e. in binary questions. Shannon's entropy

$$H(Y) \equiv -\sum p_Y(y) \log p_Y(y) = \sum p_Y(y) \log \frac{1}{p_Y(y)} = E\left[\log \frac{1}{P_Y(Y)}\right] \tag{2.3}$$

is the expected value of the information in a random variable $Y$. A sequence of events, objects or random variables, for example, a string of bits $101100011\cdots$, is of course just another object, event or random variable, and therefore is quantifiable by the same logic [6,12].

The three terms in function (2.1) are defined as follows:

— **Y** constitutes the *explanandum*, latin for 'what is to the explained'. Examples of explananda include: response variables in regression analysis, physical properties to be measured, experimental outcomes, unknown answers to questions.
— **X** and $\tau$ together constitute the *explanans*, latin for 'what does the explaining'. In particular,

 (a) **X** will be referred to as the 'input', and it will represent information acquired externally. Examples of inputs include: results of any measurement, explanatory variables in regression analysis, physical constants, arbitrary methodological decisions and all other factors that are not 'rigidly' encoded in the theory or methodology.
 (b) $\tau$ will be referred to as the 'theory' or 'methodology'. A typical $\tau$ is likely to contain both a description of the relation between $Y$ and $X$, as well as a specification of all other conditions that allow the relationship between $X$ and $Y$ to manifest. Examples of $\tau$ include: an algorithm to reproduce $Y$, a description of a physical law relating $Y$ to $X$, a description of the methodology of a study or a field (i.e. description of how subjects are selected, how measurements are made, etc.).

Specific examples of all of these terms will be offered repeatedly throughout the essay. Mathematically, all three terms ultimately consist of sequences, produced by random variables and therefore characterized by a specific quantity of information. In the cases most typically discussed in this essay, explanandum and input will be assumed to be sequences of lengths $n_Y$ and $n_X$, respectively, resulting from a series of independent identically distributed random variables, $Y$ and $X$, with discrete alphabets $\mathcal{Y}$, $\mathcal{X}$, probability distributions $p_Y$, $p_X$ and therefore Shannon entropy $H(Y)$ and $H(X)$.

The object representing the theory or methodology $\tau$ will be typically more complex than $Y$ and $X$, because it will consist in a sequence of independent random variables (henceforth, RVs) that have distinctive alphabets (are non-identical) and are all uniformly distributed. This sequence of RVs represents the sequence of choices that define a theory and/or methodology. Indicating with $T$ a RV with uniform probability distribution $P_T$, resulting from a sequence of $l$ RVs $T_i \in \{T_1, T_2 \ldots T_l\}$ each with a probability distribution $P_{T_i}$, we have

$$\log \frac{1}{p_T(\tau)} = \log \frac{1}{Pr\{T_1 = \tau_1, T_2 = \tau_2, \ldots T_l = \tau_l\}} = \sum_{i \leq l} \log \frac{1}{P_{T_i}(T_i = \tau_i)} \ . \tag{2.4}$$

The alphabet of each individual RV composing $\tau$ may have size greater than or equal to 2, with equality corresponding to a binary choice. For example, let $\tau$ correspond to the description of three components of a study's method: $\tau = $ ('randomized', 'human subject', 'female'). In the simplest possible condition, this sequence represents a draw from three independent binary choices: $1 = $ 'randomized vs not', $2 = $ 'human vs not', $3 = $ 'female vs not'. Representing each choice as a binary RV $T_i$, the probability of $\tau$ is $Pr\{T_1 = \tau_1\} \times Pr\{T_2 = \tau_2\} \times Pr\{T_3 = \tau_3\} = 0.5^3 = 0.125$ and its information content is 3 bits.

Equivalent and useful formulations of equation (2.1) are

$$K(Y^{n_Y}; X^{n_X}, \tau) = \frac{H(Y) - H(Y|X, \tau)}{H(Y) + \frac{n_X}{n_Y} H(X) - \frac{1}{n_Y} \log p(\tau)} \tag{2.5}$$

and

$$K(Y^{n_Y}; X^{n_X}, \tau) = k \times h \tag{2.6}$$

in which

$$k \equiv \frac{H(Y) - H(Y|X, \tau)}{H(Y)} \tag{2.7}$$

will be referred to as the 'effect' component, because it embodies what is often quantified by ordinary measures of effect size (§2.2.2), and

$$h \equiv \frac{1}{1 + \dfrac{n_X H(X) - \log p(\tau)}{n_Y H(Y)}} \tag{2.8}$$

will be referred to as the 'hardness' component, because it quantifies the informational costs of a methodology, which is connected to the concept of 'soft science', as will be explained in §3.8.

## 2.2. Why $K$ is a measure of knowledge

Why do we claim that equation (2.1) quantifies the essence of knowledge? This section will offer three different arguments. First, a theoretical argument, which illustrates the logic by which the $K$ function was originally derived, i.e. following two postulates about the nature of information and knowledge. Second, a statistical argument, which illustrates how the $K$ function includes the quantities that are typically computed in ordinary measures of effect size. Third, a physical argument, which explains how the $K$ function, unlike ordinary measures of effect size or information compression, has a direct physical interpretation in terms of negentropic efficiency.

### 2.2.1. Theoretical argument: $K$ as a measure of pattern encoding

Equation (2.1) is the mathematical translation of two postulates concerning the nature of the phenomenon we call knowledge:

(i) *Information is finite*. Whatever its ultimate nature may be, reality is knowable only to the extent that it can be represented as a set of discrete, distinguishable states. Although in theory the number of states could be infinite (countably infinite, that is), physical limitations ensure that the number of states that are actually represented and processed never is or can be infinite.
(ii) *Knowledge is information compression*. Knowledge is manifested as an encoding of patterns that connect states, thereby permitting the anticipation of states not yet presented, based on states that are presented. All forms of biological adaptation consist in the encoding of patterns and regularities by means of natural selection. Human cognition and science are merely highly derived manifestations of this process.

Physical, biological and philosophical arguments in support of these two postulates are offered in appendix A.

The most general quantification of patterns between finite states is given by Shannon's mutual information function

$$I(Y; X) \equiv H(Y) + H(X) - H(Y, X) = H(Y) - H(Y|X) \tag{2.9}$$

in which $H(\cdot)$ is Shannon's entropy (equation (2.3)). The mutual information function is completely free from any assumption concerning the random variables involved (figure 1). In order to turn equation (2.9) into an operationalizable quantity of knowledge, we formalize the following properties:

(i) The pattern between $Y$ and $X$ is explicitly expressed by a conditioning. We therefore posit the existence of a third random variable, $T$, with alphabet $\mathcal{T} = \{\tau_a, \tau_b \dots\}$, such that $H(Y, X|T) = H(Y|T) + H(Y|X, T)$, or $H(Y, X|T) = H(Y) + H(X)$ if $\mathcal{T} = \emptyset$. Unlike $Y$ and $X$, $T$ is assumed to be uniformly distributed, and therefore the size of its alphabet is $z = |\mathcal{T}| = 2^n$, where $n$ is the minimum number of bits required to describe each $\tau$ in the set. The uniform distribution of $T$ also implies that $H(T) = -\log \Pr\{T = \tau\} = n$.
(ii) The mutual information expressing the pattern as described above is standardized (i.e. divided by the total information content of its own terms), in order to allow comparisons between different systems.

The two requirement above lead us to formulate knowledge as resulting from the contextual, system-specific connection of the quantities, defined by the following equation:

$$\frac{I(Y; X|T)}{H(Y) + H(X) + H(T)} \equiv \frac{H(Y) - H(Y|X, T)}{H(Y) + H(X) + H(T)} \tag{2.10}$$

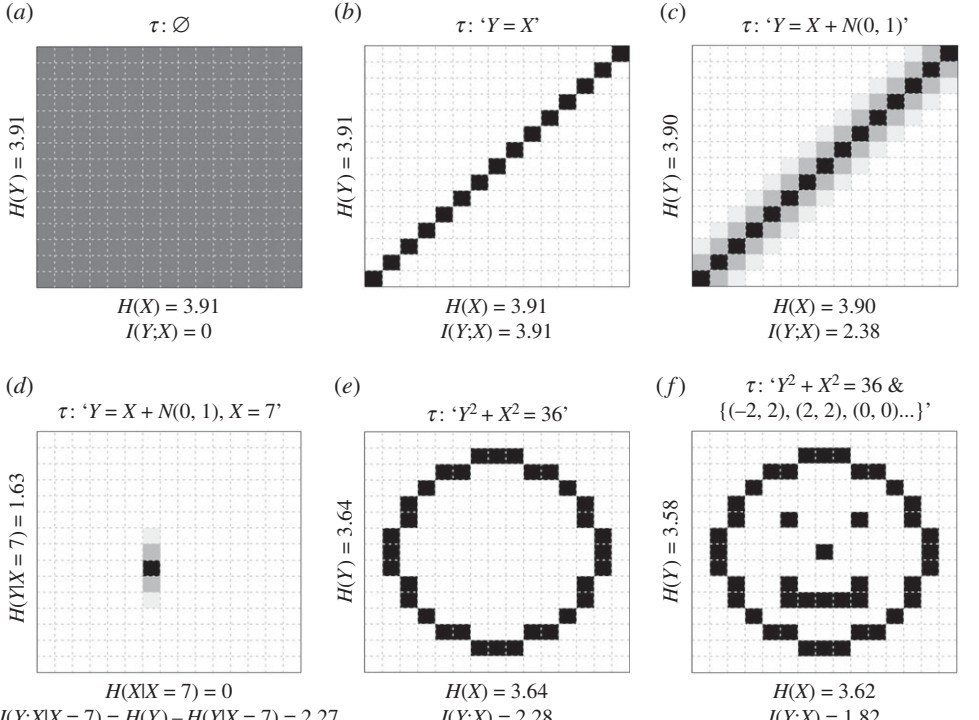

**Figure 1.** Pictorial representation of various patterns, with corresponding values of entropy and mutual information. The descriptions of the patterns $\tau$ are purely illustrative and not necessarily literal descriptions of what the pattern encodings would look like in practice. The intensity of grey in each cell represents the relative probability of occurrence of different cell values, with black $= 1$ and white $= 0$. The entropy and mutual information values were calculated by normalizing the cell values in the table or at the margins. For further details, see the source code in electronic supplementary material.

in which, to simplify the notation, we will typically use $H(Y)$ in place of $H(Y|T)$ and $H(X)$ in place of $H(X|T)$.

Note how, at this stage, the value computed by equation (2.10) is potentially very low, because $H(Y|X, T) = \sum_{\tau_i \in \mathcal{T}} P(T = \tau_i) H(Y|X, T = \tau_i)$ is the average value of the conditional entropy for every possible theory of description length $-\log p(\tau)$. The more complex is the average $\tau \in \mathcal{T}$, the larger is the number of possible theories of equivalent description length, and therefore the smaller is the proportion of theories $\tau_i$ that yield $H(Y|X, T = \tau_i) < H(Y)$ (because most realizable theories are likely to be nonsensical).

Knowledge is realized because, from *all* possible theories, only a specific theory (or possibly a subset of theories) is selected (figure 2). This selection is not merely a mathematical fiction, but is typically the result of Darwinian natural selection and/or other analogous neurological, memetic and computational processes. The details of how a $\tau$ is arrived at, however, need not concern us because, in mathematical terms, the result of a selection process is the same: the selection 'fixes' the random variable $T$ in equation (2.10) on a particular realization $\tau \in \mathcal{T}$, with two consequences. On the one hand, the entropy of $T$ goes to zero (because there is no longer any uncertainty about $T$), but on the other hand, the selection itself entails a non-zero amount of information.

Since $T$ has a uniform distribution, the information necessary to identify this realization of $T$ is simply $-\log P(T = \tau) = \log 2^{l(\tau)} = l(\tau)$, which is the shortest description length of $\tau$ (e.g. the minimum number of binary questions needed to identify $\tau$ in the alphabet of $T$). This quantity constitutes an informational cost that needs to be computed in the standardized equation (2.10). Therefore, we get

$$K(Y; X, \tau) = \frac{H(Y|T = \tau) - H(Y|X, T = \tau)}{H(Y|T = \tau) + H(X|T = \tau) + H(T|T = \tau) + l(\tau)} \equiv \frac{H(Y) - H(Y|X, \tau)}{H(Y) + H(X) - \log p(\tau)}. \tag{2.11}$$

Equation (2.1) is arrived at by generalizing (2.11) to the case in which the knowledge encoded by $\tau$ is applied to multiple *independent* realizations of explanandum and/or input, which are counted by the $n_Y$ and $n_X$ terms, respectively.

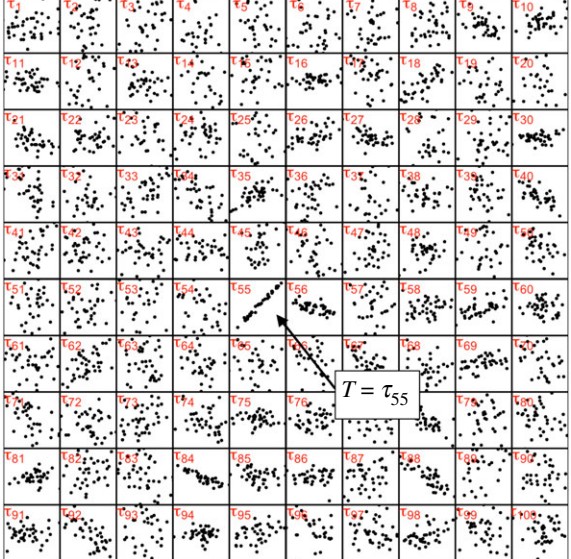

**Figure 2.** Pictorial representation of a set $\mathcal{T} = \{\tau_1, \tau_2, \ldots \tau_z\}$ of theories of a given description length that condition the relation between two variables. This set constitutes the alphabet of the uniformly distributed random variable $T$, from which a specific theory/methodology, in this case $\tau_{55}$, is selected. For further discussion, see text.

### 2.2.2. Statistical argument: $K$ as a universal measure of effect size

Despite having been derived theoretically and being potentially applicable to phenomena of any kind, i.e. not merely statistical ones, equation (2.1) bears structural similarities with ordinary measures of statistical effect size. Such similarities ought not to be surprising, in retrospect. Statistical measures of effect size are intended to quantify knowledge about patterns between variables, and so $K$ would be expected to reflect them. Indeed, structural analogies between the $K$ function and other measures of effect size offer further support for the theoretical argument made above that $K$ is a general quantifier of knowledge.

To illustrate such similarities, it is useful to point out that the value of the $K$ function can be approximated from the quantization of any continuous probability distribution. For information to be finite as required by the $K$ function, the entropy of a normally distributed quantized random variable $X^\Delta$ can be approximated by $H(X^\Delta) = \log \sqrt{2\pi e} \sigma$, in which $\sigma$ is the standard deviation rescaled to a lowest decimal (for example, from $\sigma = 0.123$ to $\sigma = 123$, further details in appendix B).

There is a clear structural similarity between the $k$ component of equation (2.6) and the coefficient of determination $R^2$. Since the entropy of a random variable is a monotonically increasing function of the variable's dispersion (e.g. its variance), this measure is directly related to $K$. For example, if $Y$ and $Y|X$ are continuous normally distributed RVs with variance $\sigma_Y$ and $\sigma_{Y|X}$, respectively, then $R^2$ is a function of $K$,

$$R^2 \equiv \frac{TSS - SSE}{TSS} \equiv \frac{n \times (\sigma_Y^2 - \sigma_{Y|X}^2)}{n \times \sigma_Y^2} = f\left(\frac{\log \sigma_Y - \log \sigma_{Y|X}}{\log \sigma_Y + C}\right) = f(K(Y; X, \tau)) \tag{2.12}$$

in which $TSS$ is the total sum of squares, $SSE$ is the sum of squared errors, $n$ is the sample size and $f(\cdot)$ represents an undefined function. The adjusted coefficient of determination $R^2_{adj}$ is also directly related to $K$ since

$$R^2_{adj} \equiv \frac{TSS/(n-1) - SSE/(n-k-1)}{TSS/(n-1)} = g\left(\frac{\log \sigma_y^2 - \log (\sigma_{y|x}^2 \times A)}{\log \sigma_y^2}\right) = f(K(Y; X, \tau)) \tag{2.13}$$

with $A = (n-1)/(n-k-1)$.

From this relation follows that multiple ordinary measures of statistical effects size used in meta-analysis are also functions of $K$. For example, for any two continuous random variables, $R^2 = r^2$, with $r$ the correlation coefficient. And since most popular measures of effect size used in meta-analysis,

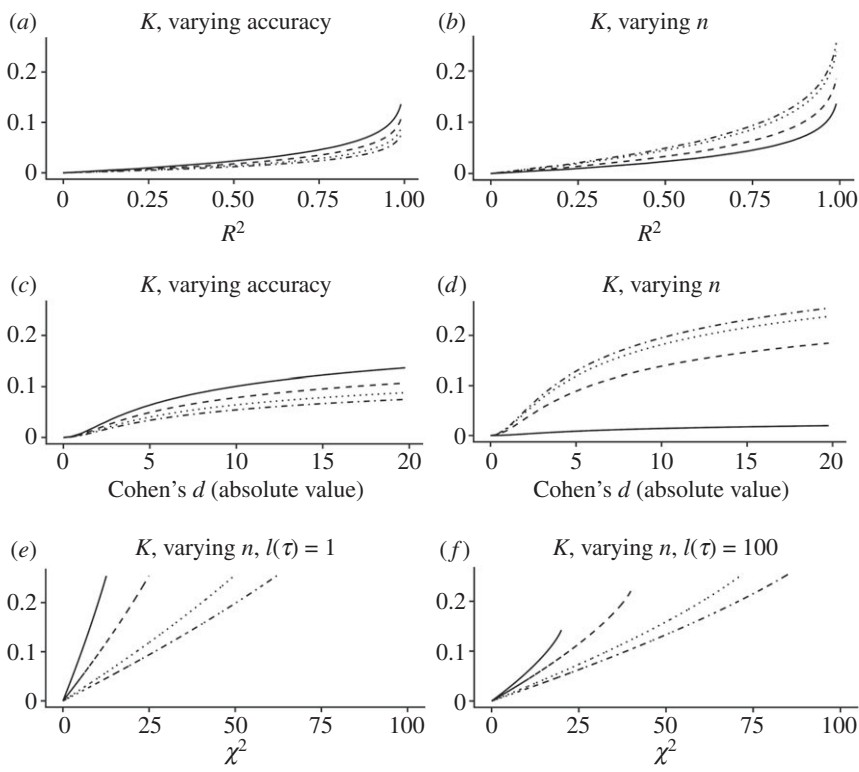

**Figure 3.** Relation between $K$ and common measures of effect size, with varying conditions of accuracy (i.e. of resolution, see §2.3.6), number of repetitions $n$ (i.e. the $n_Y$ in equation (2.1)) and size of $\tau$. The relation with $R^2$ and Cohen's $d$ was derived assuming a normal distribution of the explanandum. Increasing accuracy, in this case, corresponded to calculating entropies with a standard deviation measured with one additional significant digit, at each step, from solid line to dotted line. The values of $n$ for $R^2$ and Cohen's $d$ were, from dotted to solid line, 1, 2, 10, 100, respectively. The relation with $\chi^2$ was derived from the probability distribution of a $2 \times 2$ contingency table. From solid to dotted line, the value of $n$ was 20, 40, 80, 100, and the description length of $\tau$ was 1 bit for (a,c,e), and 100 bits for (b,d,f). The code used to generate these and all other figures is available in electronic supplementary material.

including Cohen's $d$ and odds ratios, are approximately convertible to and from $r$ [13], they are also convertible to $K$.

The direct connection between $K$ and measures of effect size like Cohen's $d$ implies that $K$ is also related to the $t$ and the $F$ distributions, which are constructed as ratios between the amount of what is explained and what remains to be explained, and are therefore constructed similarly to an 'odds' transformation of $K$

$$\frac{K(Y; X, \tau)}{1 - K(Y; X, \tau)} = \frac{n_Y(H(Y) - H(Y|X, \tau))}{n_Y H(Y|X, \tau) + n_X H(X) - \log p(\tau)}. \tag{2.14}$$

Other more general tests, such as the Chi-squared test, can be shown to be an approximation of the Kullback–Leibler distance between the probability distributions of observed and expected frequencies [12]. Therefore, they are a measure of the mutual information between two random variables, i.e. the same measure on which the $K$ function is built.

Figure 3 illustrates how these are not merely structural analogies, because $K$ can be approximately or exactly converted to ordinary measures of effect size. As the figure illustrates, $K$ stands in one-to-one correspondence with ordinary measures of effect sizes, but its specific value is modulated by additional variables that are critical to knowledge and that are ignored by ordinary measures of effect size. Such variables include the size of the theory or methodology describing the pattern, which is always non-zero, the number of repetitions (which, depending on analyses, may correspond to the sample size or to the intended total number of uses of a $\tau$); the resolution (e.g. accuracy of measurement, §2.3.6); distance in time and space and methods (§2.3.5) and Ockham's razor (§2.3.1).

The latter property also makes $K$ conceptually analogous to measures of minimum description length, discussed below.

*Minimum description length principle*. The minimum description length (MDL) principle is a formalization of the principle of inductive inference and of Ockham's razor that has many potential applications in statistical inference, particularly with regard to the problem of model selection [8]. In its most basic formulation, the MDL principle states that the best model to explain a dataset is the one that minimizes the quantity

$$L(H) + L(D|H) \tag{2.15}$$

in which $L(H)$ is the description length of the hypothesis (i.e. a candidate model for the data) and $L(D|H)$ is the description length of the data given the model. The $K$ equation has equivalent properties to equation (2.15), with $L(H) \equiv -\log p(\tau)$ and $L(D|H) \equiv n_Y H(Y|X, \tau)$. Therefore, the values that minimize equation (2.15) maximize the $K$ function.

The reader may question why, if $K$ is equivalent to existing statistical measures of effect size and MDL, we could not just use the latter to quantify knowledge. There are at least three reasons. The first reason is that only $K$ is a universal measure of effect size. The quantity measured by $K$ is completely free from any distributional assumptions about the subject matter being assessed. It can be applied not only to quantitative data with any distribution (e.g. figure 1), but also to any other explanandum that has a finite description length (although this potential application will not be examined in detail in this essay). In essence, $K$ can be applied to anything that is quantifiable in terms of information, which means any phenomenon that is the object of cognition—any phenomenon amenable to being 'known'.

The second reason is that, as illustrated above, $K$ takes into account factors that are overlooked by ordinary measures of effect size or model fit, and therefore is a more complete representation of knowledge phenomena (figure 3).

The third reason is that, unlike any of the statistical and algorithmic approaches mentioned above, $K$ has a straightforward physical interpretation, which is presented in the next section.

## 2.2.3. Physical argument: $K$ as a measure of negentropic efficiency

The physical interpretation of equation (2.1) follows from the physical interpretation of information, which was revealed by the solution to the famous paradox known as Maxwell's Demon. In the most general formulation of this *Gedankenexperiment*, the demon is an organism or a machine that is able to manipulate molecules of a gas, for example, by operating a trap door, and is thus able to segregate molecules that move at higher speed from those that move at lower speed, seemingly without dissipation. This created a theoretical paradox as it would contradict the second law of thermodynamics, according to which no process can have as its only result the transfer of heat from a cooler to a warmer body.

In one variant of this paradox, called the 'pressure demon', a cylinder is immersed in a heat bath and has a single 'gas' molecule moving randomly inside it. The demon inserts a partition right in the middle of the cylinder, thereby trapping the molecule in one half of the cylinder's volume. It then operates a measurement to assess in which half of the cylinder the molecule is, and pushes down, with a reversible process, a piston in the half that is empty. The demon could then remove the partition, allowing the gas molecule to push the piston up, and thus extract work from the system, apparently without dissipating any energy.

Objections to the paradox that involve the energetic costs of operating the machine or of measuring the position of the particle [5] were proven to be invalid, at least from a theoretical point of view [6,14]. The conclusive solution to the paradox was given in 1982 by Charles Bennett, who showed that dissipation in the process occurred as a byproduct of the demon's need to process information [15]. In order to know which piston to lower, the demon must memorize the position of the molecule, storing one bit of information, and it must eventually re-set its memory to prepare it for the next measurement. The recording of information can occur with no dissipation, but the *erasure* of it is an irreversible process that will produce heat that is at least equivalent to the work extracted from the system, i.e $kT\ln 2$ joules, in which $k$ is Boltzmann's constant. This solution to the paradox proved that information is a measurable physical quantity.

Figure 4 illustrates how the $K$ function relates to Maxwell's pressure demon. The explanandum $H(Y)$ (which is a shorthand for $H(Y|\tau)$, as explained previously) quantifies the entropy, i.e. the amount of uncertainty about the molecule's position relative to the partition in the cylinder. The input $H(X)$ is

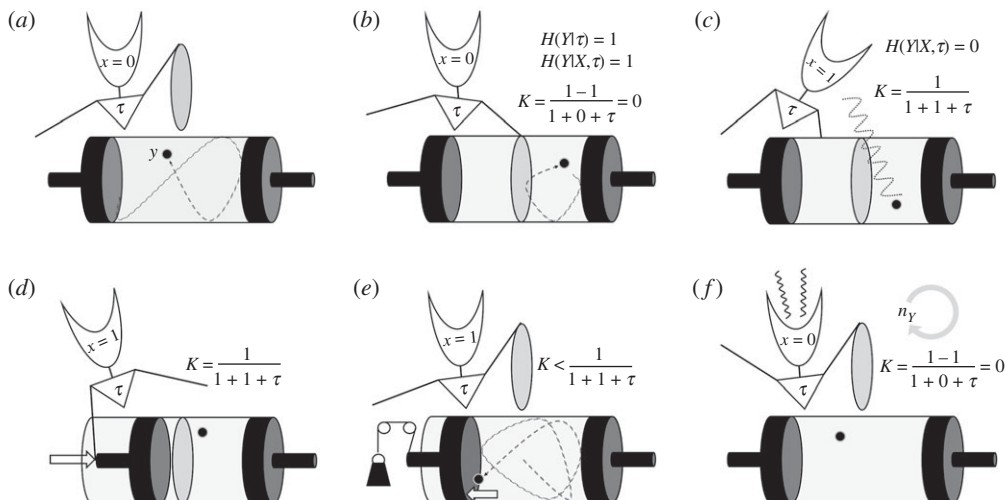

**Figure 4.** Illustration of Maxwell's 'pressure demon' paradox, and how it relates to $K$. ($a$) The system is set up, described by $\tau$, with a default memory state $X = 0$. ($b$) A partition is placed in the cylinder, generating one bit of information in the explanandum $Y$. The demon has zero knowledge about the molecule's position. ($c$) A measurement is made, allowing the position of the molecule to be stored in memory. An amount $K$ of knowledge is now possessed by the demon and put to use. ($d$) One of the pistons is pushed down allowing work to be extracted from the system. ($e$) Work is extracted at the expense of the demon's knowledge. ($f$) The demon's knowledge is now zero and its memory is re-set, dissipating entropy in the environment. The cycle will be repeated $n_Y$ times. See text for further explanations.

the external information obtained by a measurement. The input corresponds to the colloquial notion of 'information' as something that is acquired and 'gives form' (to subsequent choices, actions, etc.). Since this latter notion of information is a counterpart to the physical notion of information as entropy, it may be perhaps more correctly defined as *negentropy* [5].

The theory $\tau$ contains a description of the information-processing structure that allows the Pressure Demon to operate. The extent of this description will depend in part on how the system is defined. A minimal description will include at least an encoding of the identity relation between the state of $X$ and that of $Y$, i.e. '$X = Y$' as distinguished from its alternative, '$X \neq Y$'. This theory requires at least a binary alphabet and therefore one bit of memory storage. A more comprehensive description will include a description of the algorithm that enables the negentropy in $X$ to be exploited—something like 'if $X =$ left, press down right piston, else, press left piston'. Multiple other aspects of the system may be included in $\tau$. The amount of information contained in the explanandum, for example, is a function of where the partition is laid down, a variable that a truly complete algorithm would need to specify. The broadest possible physical description of the pressure demon ought to encode instructions to set up the entire system, i.e. the heat bath, the partition etc. In other words, a complete $\tau$ contains the genetic code to reproduce pressure demons.

The description length of $\tau$ will, intuitively, also depend on the language used to describe it. Moreover, some descriptions might be less succinct than others and contain redundancies, unnecessary complexities, etc. From a physical point of view, however, it is well understood that each $\tau$ would be characterized by its own specific minimum amount of information, a quantity known as Kolmogorov complexity [6]. This is defined as the shortest program that, if fed into a universal Turing machine, would output the $\tau$ and then halt. Mathematical theorems prove that this quantity cannot be computed directly—at least in the sense that one can never be sure to have found the shortest possible program. In practice, however, the Kolmogorov complexity of an object is approximated, by excess, by any information compression algorithm and is independent of the encoding language used, up to a constant. This means that, even though we cannot measure the Kolmogorov complexity in absolute terms, we can measure it rather reliably in relative terms. A $\tau$ that is more complex, and/or more redundant than another $\tau$ will necessarily have, all else being equal, a longer description length.

Whether we take $\tau$ to represent the theoretical shortest possible description length for the demon (in which case $-\log p(\tau)$ quantifies its Kolmogorov complexity), or whether we assume that it is a realistic, suboptimal description (in which case the description length $-\log p(\tau)$ is best interpreted in relative terms), the $K$ function expresses the efficiency with which the demon converts information into work.

At the start of the cycle, the demon's K is zero. After measuring the particle's position, the demon has stored one bit of information (or less, if the partition is not placed in the middle of the cylinder, but we will here assume that it is), and has knowledge $K > 0$, with the magnitude of K inversely related to the description length of $\tau$. By setting the piston and removing the partition, the demon puts its knowledge to use and extracts $k \ln 2$ of work from it. Once the piston is fully pushed out, the demon no longer knows where the molecule is ($K = 0$) and yet still has one bit stored in memory, a trace of its last experience. The demon has now two possible options. First, as in Bennett's solution to the paradox, it can simply erase that bit, re-setting X to the initial state $H(X) = 0$ and releasing $k \ln 2$ in the environment. At each cycle, the negentropy is renewed via a new measurement, whereas the fixed $\tau$ component remains unaltered. Since the position of the molecule at each cycle is independent of previous positions, the total cumulative explanandum (the total entropy that the demon has reduced) grows by one bit, whereas the theory component remains unaltered. For n cycles, the total K is therefore

$$K = \frac{nH(Y)}{nH(Y) + nH(X) - \log p(\tau)} = \frac{1}{1 + 1 - \dfrac{\log p(\tau)}{n}}, \tag{2.16}$$

which to the limit of infinite cycles is

$$\lim_{n \to \infty} K = \frac{1}{2}. \tag{2.17}$$

The value of $K = 1/2$ constitutes the absolute limit for knowledge that requires a direct measurement and/or a complete and direct description of the explanandum.

Alternatively, the demon could keep the value of X in memory and allocate new memory space for the information to be gathered in the next cycle ([6]). As Bennett also pointed out, in practice it could not do so forever. In any physical implementation of the experiment, the demon would eventually run out of memory space and would be forced to erase some of it, releasing the entropy locked in it. If, *ad absurdum*, the demon stored an infinite amount of information, then at each cycle the input would grow by one bit yielding

$$K = \frac{1}{1 + n - \dfrac{\log p(\tau)}{n}}, \tag{2.18}$$

which to the limit of infinite cycles is

$$\lim_{n \to \infty} K = 0 \tag{2.19}$$

again independent of $\tau$. This is a further argument to illustrate how information is necessarily finite, as we postulated (§2.2.1, see also §2.3.6 for another mathematical argument and appendix A for philosophical and scientific arguments).

More realistically, we can imagine that the number of physical bits available to the demon is finite. As cycles progress, the demon could try to allocate as many resources as possible to the memory X, for example, by reducing the space occupied by $\tau$. This is why knowledge entails compression and pattern encoding (see also §2.3.1).

Elaborations on the pressure demon experiment shed further light on the meaning of K and its implications for knowledge. First, let us imagine that the movement of the gas molecule is not actually random, but that, acted upon by some external force, the molecule periodically and regularly finds itself alternatively on the right and left side of the cylinder, and expands from there. If the demon kept a sufficiently long record of past measurements, say a number z of bits, it might be able to discover the pattern. Its $\tau$ could then store a new, slightly expanded algorithm, such as 'if last position was left, new position is right, else, new position is left'. With this new theory, and one bit of input to determine the initial position of the molecule, the demon could extract unlimited amounts of energy from the heat bath. In this case,

$$K = \frac{1}{1 + \dfrac{1}{n} - \dfrac{\log p(\tau)}{n}} \tag{2.20}$$

which to the limit of infinite cycles is

$$\lim_{n \to \infty} K = 1. \tag{2.21}$$

Therefore, the maximum amount of knowledge expressed in a system asymptotically approaches 1. As we would expect, it is higher than the maximum value of 1/2 attained by mere descriptions. Note, however, that $K$ can never actually be equal to 1, since $n$ is never actually infinite and $\tau$ cannot be 0.

Intermediate cases are also easy to imagine, in which the behaviour of the molecule is predictable only for a limited number of cycles, say $c$. In such case, $K$ would increase as the number of necessary measurements $n_X$ is reduced to $n_X/c$. At any rate, this example illustrated how the demon's ability to implement knowledge (in order to extract work, create order, etc.) is determined by the presence of regularities in the explanandum as well as the efficiency with which the demon can identify and encode patterns. Since this ability is higher when the explanans is minimized, the demon (the $\tau$) is selected to be as 'intelligent' and 'informed' as possible.

As a final case, let us imagine instead that the gas molecule moves at random and that its position is measurable only to limited accuracy. A single measurement yields the position of the molecule with an error $\eta$. However, each additional measurement reduces $\eta$ by a fraction $a$. The demon, in this case, could benefit from increasing the number of measurements. Indicating with $m$ the number of measurements and with $\tau_m$ the corresponding theory we have

$$K = \frac{1 - \eta \times a^{-m}}{1 + m - \dfrac{\log p(\tau_m)}{n}} \tag{2.22}$$

that to the limit of infinite cycles is

$$\lim_{n \to \infty} K = \frac{1 - \eta \times a^{-m}}{1 + m} < \frac{1}{2}. \tag{2.23}$$

The work extracted at each cycle will be $k \ln 2 \, (1 - \eta \times a^{-m})$. Therefore, $K$ expresses the efficiency with which work can be extracted from a system, given a certain error rate $a$ and number of measurements $m$.

## 2.3. Properties of knowledge

This section will illustrate how $K$ possesses properties that a measure of knowledge would be expected to possess. In addition to offering support for the three arguments given above, these properties underlie some of the results presented in §3.

### 2.3.1. Ockham's razor is relative.

As discussed in §2.2.2, the $K$ function encompasses the MDL principle, and therefore computes a quantification of Ockham's razor. However, the $K$ formulation of Ockham's razor highlights a property that other formulations overlook: that Ockham's razor is relative to the size of the explanandum and the number of times a given theory or explanation can be used. For a given $Y$ and $X$ and two alternative theories $\tau$ and $\tau'$ that have the same effect $H(Y|X, \tau) = H(Y|X, \tau')$ and that can be applied to a number of repetitions $n_Y$ and $n'_Y$, respectively, we have that

$$\frac{-\log p(\tau')}{n'_Y} < \frac{-\log p(\tau)}{n_Y} \Leftrightarrow K(Y^{n'_Y}; X, \tau') > K(Y^{n_Y}; X, \tau) \tag{2.24}$$

and similarly for the case in which $\tau = \tau'$ while $n_X H(X) \neq n'_X H(X')$,

$$\frac{n'_X H(X')}{n'_Y} < \frac{n_X H(X)}{n_Y} \Leftrightarrow K(Y^{n'_Y}; X^{n'_X}, \tau) > K(Y^{n_Y}; X^{n_X}, \tau). \tag{2.25}$$

Therefore, the relative epistemological value of the simplicity of an explanans, i.e. Ockham's razor, is modulated by the number of times that the explanans can be applied to the explanandum.

### 2.3.2. Prediction is more costly than explanation, but preferable to it.

The $K$ function can be used to quantify either explanatory or predictive efficiency. The expected (average) explanatory or predictive efficiency of an explanans with regard to an explanandum is measured when the terms of the $K$ function are entropies, i.e. expectation values of uncertainties. If instead the explanandum is an event that has already occurred and that carries information $-\log P(Y = y)$, $K$ quantifies the value of an explanation, whose information cost includes the surprisal of explanatory conditions $-\log P(X = x)$ and the complexity of the theory linking such conditions to the event,

$-\log P(T = \tau)$. Inference to the best explanation and/or model is, in both these cases, driven by the maximization of $K$.

If instead it is the explanans, that is pre-determined and fixed, then its predictive power is quantified by how divergent its predictions are relative to observations. To any extent that observations do not match predictions, the observed and predicted distributions will have a non-zero informational divergence, which quantifies the extra amount of information that would be needed to 'adjust' the predictions to make them match the observations. It follows that, indicating with the tilde sign the predictive theory, we can calculate an 'adjusted' $K$ as

$$K_{\text{adj}} = K_{\text{obs}} - D(Y|X, \tau \| Y|X, \tilde{\tau}) \frac{h}{H(Y)} \tag{2.26}$$

in which $K_{\text{obs}} = k_{\text{obs}} h = K(Y; X, \tau)$ is the $K$ observed, and $D(\cdot)$ is the Kullback–Leibler divergence between the observed and the predicted distribution (proof in appendix C). Since $D(Y|X, \tau \| Y|X, \tilde{\tau}) \geq 0$, $K_{\text{adj}} \leq K_{\text{obs}}$, with equality corresponding to perfect fit between observations and predictions. An analogous formula could be derived for the case in which the explanandum is a sequence, in which case the distance would be calculated following methods suggested in §3.3.3.

Now, note that the observed $K$ is the explanatory $K$, and therefore is always greater or equal to the predictive $K$ for individual observations. When evidence cumulates, then the explanans of an explanatory $K$ is likely to expand, reducing the cumulative $K$ (§3.3). Replacing a 'flexible' explanation with a fixed one avoids these latter cumulative costs, allowing a fixed explanans to be applied to a larger number of cases $n_Y$, with no cumulative increase in its complexity.

Therefore, predictive knowledge is simply a more generalized, unchanging form of explanatory knowledge. As intuition would suggests, prediction can never yield more knowledge than a post hoc explanation for a given event (e.g. an experimental outcome). However, predictive knowledge becomes cumulatively more valuable to the extent that it allows to explain, with no changes, a larger number of events, backwards or forwards in time.

### 2.3.3. Causation entails correlation and is preferable to it

Properties of the $K$ function also suggests why the knowledge we gain from uncovering a cause–effect relation is often, but not always, more valuable than that derived from a mere correlation. Definitions of causality have a long history of subtle philosophical controversies [16], but no definition of causality can dispense with counterfactuals and/or with assuming that manipulating present causes can change future effects [17]. The difference between a mere correlation and a causal relation can be formalized as the difference between two types of conditional probabilities, $P(Y = y | X = x)$ and $P(Y = y | do(X = x))$, where '$do(X = x)$' is a shorthand for '$X | do(X = x)$' and the '$do$' function indicates the manipulation of a variable. In general, correlation without causation entails $P(Y = y) \leq P(Y = y | X = x)$ and $P(Y = y) = P(Y = y | do(X = x))$ whereas causation entails $P(Y = y) \leq P(Y = y | X = x) \leq P(Y = y | do(X = x))$.

If knowledge is exclusively correlational, then $K(Y; X = x, \tau) > 0$ and $K(Y; do(X = x), \tau) = 0$, otherwise $K(Y; X = x, \tau) > 0$ and $K(Y; do(X = x), \tau) > 0$. Hence, all else being equal, the knowledge attainable via causation is larger under a broader set of conditions. Moreover, note that in the correlational case knowledge is only attained once an external input of information is obtained, which has an informational cost $n_Y H(X) > 0$. In the causal case, instead, the input has no informational cost, i.e. $H(X | do(X = x)) = 0$, because there is no uncertainty about the value of $X$, at least to the extent that the manipulation of the variable is successful. However, the explanans is expanded by an additional $\tau_{do(X=x)}$, which is the description length of the methodology to manipulate the value of $X$. Therefore, the value of causal knowledge is defined as

$$K(Y; \tau, \tau_{do(X=x)}) = \frac{n_Y H(Y) - n_Y H(Y|X, \tau)}{n_Y H(Y) - \log p(\tau) - \log p(\tau_{do(X=x)})} \equiv \frac{H(Y) - H(Y|X, \tau)}{H(Y) + \frac{-\log p(\tau_{do(X=x)}) - \log p(\tau)}{n_Y}} . \tag{2.27}$$

It follows that there is always an $n_Y^* \in \mathbb{N}$ such that $K(Y^{n_Y^*}; \tau, \tau_{do(X=x)}) > K(Y^{n_Y^*}; X^{n_Y^*}, \tau)$. Specifically, assuming $\tau$ to be constant, causal knowledge is superior to correlational knowledge when $n_Y^* > -\log p(\tau_{do(X=x)})/H(X)$.

### 2.3.4. Knowledge growth requires lossy information compression

Both theoretical and physical arguments suggest that $K$ is maximized when $\tau$ is minimized (§2.2). A simple calculation shows that such minimization must eventually consist in the encoding of

concisely described patterns, even if such patterns offer an incomplete account of the explanandum, because otherwise knowledge cannot grow indefinitely.

Let $\tau$ be a theory that is not encoding a relation between RVs $X$ and $Y$, but merely lists all possible $(x, y)$ pairs of elements from the respective alphabets, i.e. $x \in \mathcal{X}$ and $y \in \mathcal{Y}$. To take the simplest possible example, let each element $x \in \mathcal{X}$ correspond to one element of $y \in \mathcal{Y}$. Clearly, such $\tau$ would always yield $H(Y|X, \tau) = 0$, but its description length will grow with the factorial of the size of the alphabet. Indicating with $s$ the size of the two alphabets, which in our example have the same length, the size of $\tau$ would be proportional to $\log(s!)$. As the size of the alphabet grows, knowledge declines because

$$\lim_{s \to +\infty} K(Y; X, \tau) = \lim_{s \to +\infty} \frac{n_Y H(Y)}{n_Y H(Y) + n_X H(X) + \log(s!)} = 0 \tag{2.28}$$

independent of the probability distribution of $Y$ and $X$. Therefore, as the explanandum is expanded (i.e. its total information and/or complexity grows), knowledge rapidly decreases, unless $\tau$ is something other than a listing of $(x, y)$ pairs. In other words, knowledge cannot grow unless $\tau$ consists in a relatively short description of some pattern that exploits a redundancy. The knowledge cost of a finite level of error or missing information $H(Y|X, \tau) > 0$ will soon be preferable to an exceedingly complex $\tau$.

### 2.3.5. Decline with distance in time, space and/or explanans

Everyone's experience of the physical world suggests that our ability to predict future states of empirical phenomena tends to become less accurate the more 'distant' the phenomena are from us, in time or space. Perhaps less immediately obvious, the same applies to explanations: the further back we try to go in time, the harder it becomes to connect the present state of phenomena to past events. These experiences suggest that any spatio-temporal notion of 'distance' is closely connected to the information-theoretic notion of 'divergence'. In other words, our perception that a distance in time or space separates us from objects or events is cognitively intertwined, if not indeed equivalent, to our diminished ability to access and process information about those objects or events and, therefore, to our knowledge about them.

One of the most remarkable properties of $K$ is that it expresses how knowledge changes with informational distances between systems. It can be shown that, under most conditions in which a system contains knowledge, divergence in any component of the system will lead to a decline of $K$ that can be described by a simple exponential function of the form

$$K(Y'; X', \tau') = K(Y; X, \tau) \times A^{-\lambda \cdot d} \tag{2.29}$$

in which $A$ is an arbitrary basis, $Y'$, $X'$, $\tau'$ is a system having an overall distance (i.e. informational divergence) $d$ from $Y$, $X$, $\tau$, and $\lambda \cdot d = d_Y \lambda_Y + d_X \lambda_X + d_{\tau 1} \lambda_{\tau 1} + d_{\tau 2} \lambda_{\tau 2} + \cdots + d_{\tau l} \lambda_{\tau 1}$ defines the decline rate (proof in appendix D).

### 2.3.6. Knowledge has an optimal resolution

Accuracy of measurement is a special case of the general informational concept of resolution, quantifiable as the number of bits that are available to describe explanandum and explanans. It can be shown both analytically and empirically that any system $Y$, $X$, $\tau$ is characterized by a unique optimal resolution that maximizes $K$ (the full argument is offered in appendix E).

We may start by noticing how, even if empirical data is assumed to be measurable to infinite accuracy (against one of the postulates in §2.2.1), the resulting $K$ value will be inversely proportional to measurement accuracy, unless special conditions are met. When $K$ is measured on a continuous, normal and quantized random variable $Y^\Delta$ (§2.2.2), to the limit of infinite accuracy only one of two values is possible,

$$\lim_{n \to \infty} K(Y^\Delta; X, \tau) = \begin{cases} \lim_{n \to \infty} \frac{h(Y) + n - h(Y|X, \tau) - n}{h(Y) + n + H(X) - \log p(\tau)} = 0 \\ \lim_{n \to \infty} \frac{h(Y) + n}{h(Y) + n + H(X) - \log p(\tau)} = 1 \end{cases} \tag{2.30}$$

with $h(\cdot)$ representing Shannon's differential entropy function. The upper limit in equation (2.30) occurs if and when $h(Y|X, \tau) > 0$, i.e. by assumption there is a non-zero residual uncertainty that needs to be measured. When this is the case, then the two information terms $n$ brought about by the quantization cancel each other out in the numerator (because the explanandum and the residual error are necessarily measured at the same resolution). This is the typical case of empirical knowledge. The

lower limit in equation (2.30) presupposes *a priori* that $h(Y|X, \tau) = 0$, i.e. the explanandum is perfectly known via the explanans and there is no residual error to be quantized. This case is only represented by logico-deductive knowledge.

We can define empirical systems as intermediate cases, i.e. cases that have a non-zero conditional entropy and have a finite level of resolution. We can show (see appendix E) that all empirical systems have 'K-optimal' resolutions $\alpha_Y^*$ and $\alpha_X^*$, such that

$$K(Y^{\alpha_Y^*}; X^{\alpha_X^*}, \tau) > K(Y^{\alpha_Y}; X^{\alpha_X}, \tau) \forall \alpha_Y \neq \alpha_Y^*, \alpha_X \neq \alpha_X^*. \tag{2.31}$$

As the resolution increases, $K$ will increase up to a maximal value and then decline.

A system's optimal resolution is partially determined by the shape of the relation between explanandum and explanans in ways that are likely to be system-specific. Two simulations in figure 5 illustrate how both $K$ and $H(Y)K$ may vary depending on resolution.

The dependence of $K$ on resolution reflects its status as a measure of entropic efficiency (§2.2.3) and entails that, to compare systems for which the explanandum is measured to different levels of accuracy, the $K$ value needs to be rescaled. Such rescaling can be attained rather simply, by multiplying the value of $K$ by the entropy of the corresponding explanandum,

$$H(Y) \times K(Y; X, \tau). \tag{2.32}$$

The resulting product quantifies in absolute terms how many bits are extracted from the explanandum by the explanans.

# 3. Results

This section will illustrate, with practical examples, how the tools developed so far can be used to answer meta-scientific questions. Each of the questions is briefly introduced by a problem statement, followed by the answer, which comprises a mathematical equation, an explanation and one or more examples. Most of the examples are offered as suggestions of potential applications of the theory, and the specific results obtained should not be considered conclusive.

## 3.1. How much knowledge is contained in a theoretical system?

*Problem:* Unlike empirical knowledge, which is amenable to errors that can be verified against experiences, knowledge derived from logical and deductive processes conveys absolute certainty. It might therefore seem impossible to compare the knowledge yield of two different theories, such as two mathematical theorems. The problem is made even deeper by the fact that any logico-deductive system is effectively a tautology, i.e. a system that derives its own internal truths from a set of *a priori* axioms. How can we quantify the knowledge contained such a system?

*Answer:* The value of theoretical knowledge is quantified as

$$K = h \tag{3.1}$$

in which $K$ corresponds to equation (2.1) and $h$ to equation (2.8).

*Explanation:* Logico-deductive knowledge, like all other forms of knowledge, ultimately consists in the encoding of patterns. Mathematical knowledge, for example, is produced by revealing previously unnoticed logical connections between a statement with uncertainty $H(Y)$ and another statement, which may or may not have uncertainty $H(X)$ (depending on whether $X$ has been proven, postulated or conjectured), via a set of passages described in a proof $\tau$. The latter consists in the derivation of identities, creating an error-free chain of connections such that $P(Y|X, \tau) = 1$.

When the proof of the theorem is correct, the effect component $k$ in equation (2.6), is always equal to one, yielding equation (3.1). However, when the chain of connections $\tau$ is replaced with a $\tau'$ at a distance $d_\tau > 0$ from it, $k$ is likely to be zero, because even minor modifications of $\tau$ (for example, changing a passage in the proof of a theorem) break the chain of identities and invalidate the conclusion. This is equivalent to the case $\lambda_\tau \approx \infty$. Therefore, the reproducibility (§3.4) of mathematical knowledge, as it is embodied in a theorem, is either perfect or null,

$$K_r = K \quad \text{if } d_\tau = 0, \quad K_r = 0 \quad \text{otherwise.} \tag{3.2}$$

# relation between resolution and K

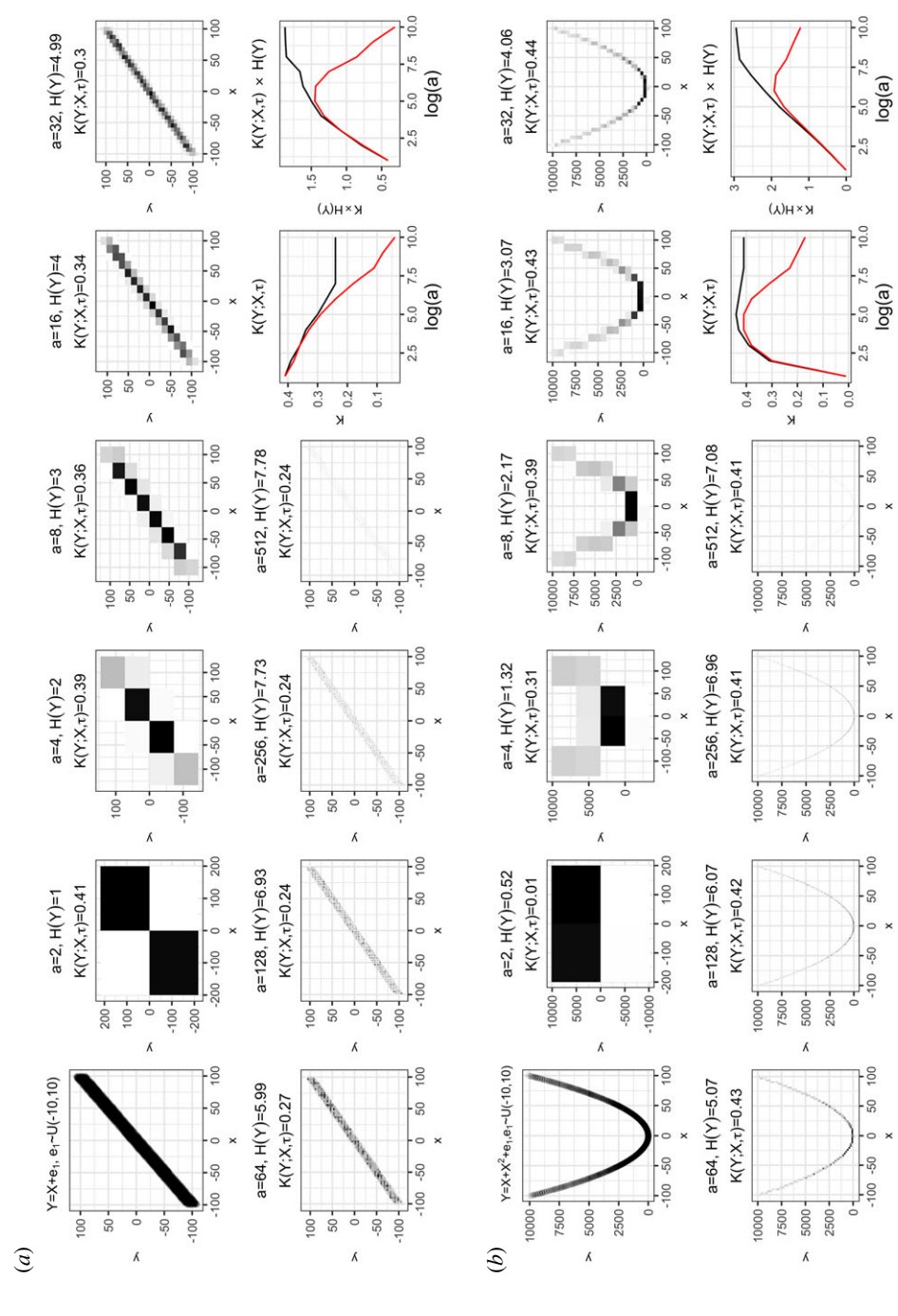

**Figure 5.** Illustrative example of how *K* varies in relation to the resolution measured for *Y* and *X*, depending on the shape of the pattern encoded. The figures and all the calculations were derived from a simulated dataset, in which the pattern linking explanandum to explanans was assumed to have noise with uniform distribution, as described in the top-left plot of each panel. Black line: entropies and *K* values calculated by maximum-likelihood method (i.e. counting frequencies in each bin). Red line: entropies and *K* values calculated using the 'shrink' method described in [18] (the R code used to generate the figures is provided in electronic supplementary material). Note how the value of *K* and its rescaled version *H*(*Y*)*K* have a unique maximum.

Alternative valid proofs, however, might also occur, and their $K$ value will be inversely proportional to their length, since a shorter proof yields a higher $h$.

Once a theorem is proven, its application will usually not require invoking the entire proof $\tau$. In $K$, we can formalize this fact by letting $\tau$ be replaced by a single symbol encoding the nature of the relationship itself. The entropy of $\tau$ will in this case be minimized to that of a small set of symbols, e.g. $\{=, \neq, >, < \cdots\}$. In such case, the value of the knowledge obtained will be primarily determined by $n_Y$, which is the number of times that the theorem will be invoked and used. This leads to the general conclusion that *the value of a theory is inversely related to its complexity and directly related to the frequency of its use*.

### 3.1.1. Example: The proof of Fermat's last theorem.

Fermat's last theorem (henceforth, FLT) states that there is no solution to the equation $a^n + b^n = c^n$ when all terms are positive integers and $n > 2$. The French mathematician Pierre de Fermat (1607–1665) claimed to have proven such statement, but his proof was never found. In 1995, Andrew Wiles published a proof of FLT, winning a challenge that had engaged mathematicians for three centuries [19]. How valuable was Wiles' contribution?

We can describe the explanandum of FLT as a binary question: 'does $a^n + b^n = c^n$ have a solution'? In absence of any proof $\tau$, the answer can only be obtained by calculating the result for any given set of integers $[a, b, c, n]$. Let $n_Y$ be the total plausible number of times that this result could be calculated. Of course, we cannot estimate this number exactly, but we are assured that this number is an integer (because a calculation is either made or not), and that it is finite (because the number of individuals, human or otherwise, who have, will, or might do calculations is finite). Therefore, the explanandum is $n_Y H(Y)$. For simplicity, we might assume that in absence of any proof, individuals making the calculations are genuinely agnostic about the result, such that $H(Y) = 1$.

Indicating with $\tau$ the maximally succinct (i.e. maximally compressed) description of this proof, the knowledge yielded by it is

$$K(Y^{n_Y}; \tau) = \frac{n_Y H(Y)}{n_Y H(Y) - \log p(\tau)} \equiv \frac{1}{1 - \dfrac{\log p(\tau)}{n_Y}}. \tag{3.3}$$

Here we assume that any input is contained in the proof $\tau$. The information size of the latter is certainly calculable in principle, since, in its most complete form, it will consist in an algorithm that derives the result from a small set of axioms and operations.

Wiles' proof of FLT is over 100 pages long and is based on highly advanced mathematical concepts that were unknown in Fermat's times. This suggests that Fermat's proof (assuming that it existed and was correct) was considerably simpler and shorter than Wiles'. Mathematicians are now engaged in the challenge of discovering such a simple proof.

How would a new, simpler proof compare to the one given by Wiles? Indicating this simpler proof with $\tau'$ and ignoring $n_Y$ because it is constant and independent of the proof, the maximal gain in knowledge is

$$K(Y; \tau') - K(Y; \tau) = \frac{1}{1 - \log p(\tau')} - \frac{1}{1 - \log p(\tau)} = \frac{-\log p(\tau) - (-\log p(\tau'))}{(1 - \log p(\tau'))(1 - \log p(\tau))} \approx \frac{\log p(\tau') - \log p(\tau)}{\log p(\tau') \times \log p(\tau)}. \tag{3.4}$$

Equation (3.4) reflects the maximal gain in knowledge obtained by devising a simpler, shorter proof of a previously proven theorem.

Given two theorems addressing different questions, in the more general case, the difference in knowledge yield will depend on the lengths of the respective proofs as well as the number of computations that each theorem allows to be spared. The general formula is, indicating with $Y'$ and $\tau'$ an explanandum and explanans different from $Y$ and $\tau$,

$$K(Y'; \tau') - K(Y; \tau) = \frac{n \log p(\tau') - n' \log p(\tau)}{(n' - \log p(\tau'))(n - \log p(\tau))}. \tag{3.5}$$

## 3.2. How much knowledge is contained in an empirical system?

*Problem:* Science is at once a unitary phenomenon and highly diversified and complex one. It is unitary in its fundamental objectives and in general aspects of its procedures, but it takes a myriad different forms when it is realized in individual research fields, whose diversity of theories, methodologies, practices,

sociologies and histories mirrors that of the phenomena being investigated. How can we compare the knowledge obtained in different fields, about different subject matters?

*Answer:* The knowledge produced by a study, a research field, and generally a methodology is quantified as

$$K = k \times h \tag{3.6}$$

in which $K$ is given by equation (2.1), $k$ by equation (2.7) and $h$ by equation (2.8).

*Explanation:* Knowledge entails a reduction of uncertainty, attained by the processing of stored information by means of an encoded procedure (an algorithm, a 'theory', a 'methodology'). Equation (3.6) quantifies the efficiency with which uncertainty is reduced. This is a scale-free, system-specific property. The system is uniquely defined by a combination of explanandum, explanans and theory, the information content of which is subject to physical constraints. Such physical constraints ensure that, among other properties, every system $Y$, $X$, $\tau$ has an optimal resolution, non-zero and non-infinite, and therefore a unique identifiable value $K$ (§2.3.6). As discussed in §2.3.6, this quantity can also be rescaled to $K \times H(Y)$, which gives the total net number of bits that are extracted from the explanandum by the explanans. Since $k \leq 1$, theoretical knowledge is typically, although not necessarily always, larger than empirical knowledge. Equation (3.6) applies to descriptive knowledge as well as correlational or causal knowledge, as examples below illustrate.

## 3.2.1. Example 1: The mass of the electron

Decades of progressively accurate measurements have led to a current estimate of the mass of the electron of $m_e = 9.10938356 \pm 11 \times 10^{-31}$ kg (based on the NIST recommended value [20]), with the error term representing the standard deviation of normally distributed errors. Since this is a fixed number of 39 significant digits, the explanandum is quantified by the amount of storage required to encode it, i.e. a string of information content $-\log P(Y = y) = 39 \times \log(10)$, and the residual uncertainty is quantified by the entropy of the normal distribution of errors with $\sigma = 11$. These measurements are obtained by complex methodologies that are in principle quantifiable as a string of inputs and algorithms, $-\log p(x) - \log p(\tau)$. However, the case of physical constants is similar to that of a mathematical theorem, in that the explanans becomes negligible to the extent that the value obtained can be used in a very large number of subsequent applications. Therefore, we estimate our current knowledge of the mass of the electron to be

$$K(m_e) = \frac{39 \log 10 - \log \sqrt{2\pi e} 11}{39 \log 10} \frac{1}{1 - \dfrac{\log p(x) - \log p(\tau)}{n_Y 39 \log 10}} \approx 0.957 \tag{3.7}$$

with the last approximation due to the case that the value can be stored and used for a very large $n_Y$ times, yielding $h \approx 1$. More accurate calculations would require estimating the $h$ component, too. In particular, to compare $K(m_e)$ to the $K$ value of another constant, the relative frequency of use would need to be taken into account. The corresponding rescaled value is $K(m_e) \times 39\log 10 \approx 124$ bits.

Note that the specific value of $K$ depends on the scale or unit in which $m_e$ is measured. If it is measured in grams ($10^{-3}$ kg), for example, then $K(m_e) = 0.954$. This reflects the fact that units of measurement are just another definable component of the system: there is no 'absolute' value of $K$, but solely one that is relative to how the system is defined. The relativity of $K$ may lead to difficulties when comparing systems that are widely different from each other (§3.8). However, results obtained comparing systems that are adequately similar to each other are coherent and consistent, as illustrated in the next paragraph.

We could be tempted to 'cheat' by rescaling the value of $m_e$ to a lower number of digits, in order to ignore the current measurement error. For example, we could quantify knowledge for the mass measured to 36 significant digits only (which is likely to cover over three standard deviations of errors, and therefore over 99% of possible values). By doing so, we would obtain $K(m_e) \approx 1$, suggesting that at that level of accuracy, we have virtually perfect knowledge of the mass of the electron. This is indeed the case: we have virtually no uncertainty about the value of $m_e$ in the first few dozen significant digits. However, note that the rescaled value of $K$ is $K(m_e) \times 36 \log 10 = 119.6$ bits. Therefore, by lowering the resolution, our knowledge increased in relative but not in absolute terms.

It should be emphasized that we are measuring here the knowledge value of the mass of the electron in the narrowest possible sense, i.e. by restricting the system to the mass itself. However, the knowledge we derive by measuring (describing) phenomena such as a physical constant has value also in a broader context, in its role *as an input* required to know other phenomena, as the next example illustrates.

### 3.2.2. Example 2: Predicting an eclipse

The total solar eclipse that occurred in North America on 21 August 2017 (henceforth, $E_{2017}$) was predicted with a spatial accuracy of 1–3 km, at least in publicly accessible calculations [21]. This error is mainly due to irregularities in the Moon's surface and, to a lesser extent, to irregularities of the shape of the Earth. Both sources of error can be reduced further with additional information and calculations (and thus a longer explanans), but we will limit our analysis to this estimate and therefore assume an average prediction error of $4 \, \text{km}^2$.

What is the value of the explanans for this knowledge? The theory component of the explanans consists in calculations based on the JPL DE405 solar system ephemeris, obtained via numerical integration of 33 equations of motion, derived from a total of 21 computations [22]. In the words of the authors, these equations are deemed to be 'correct and complete to the level of accuracy of the observational data' [22], which means that this $\tau$ can be used for an indefinite number $n_Y$ of computations, suggesting that we can assume $-\log p(\tau)/n_Y \approx 0$.

The input is in this case a defined object of information content $H(X) = -\log p(x)$. It contains 98 values of initial conditions, physical constants and parameters, measured to up to 20 significant digits, plus 21 auxiliary constants used to correct previous data, and the radii of 297 asteroids [22]. Assuming for simplicity that on average these inputs take five digits, we estimate the total information of the input to be at least $(98 + 21 + 297) \times 5 \times \log 10 \approx 6910$ bits. The accuracy of predictions is primarily determined by the accuracy of measurement of these parameters, which moreover are in many cases subject to revision. Therefore, in this case $n_X/n_Y > 0$, and the value of $H(X)$ is less appropriately neglected. Nonetheless, we will again assume for simplicity that $n_Y \gg n_X$ and thus $h \approx 1$.

Therefore, since the surface of the Earth is approximately $510\,072\,000 \, \text{km}^2$, we estimate our astronomical knowledge to be

$$K(E_{2017}; X, \tau) \approx \frac{\log(510\,072\,000) - \log(4)}{\log(510\,072\,000)} = 0.931 \tag{3.8}$$

and a rescaled value of $K(E_{2017}; X, \tau) \times \log(510\,072\,000) = 26.9261$.

Therefore, the value of $K$ for predicting eclipses is smaller than that obtained for physical constants (§3.2.1). However, our analysis is not complete and it still over-estimates the $K$ value of predicting an eclipse for at least two reasons. First, because the assumption of a negligible explanans for eclipse prediction is a coarser approximation than for physical constants, since physical constant are required to predict eclipses, and not vice versa. Secondly, and most importantly, our knowledge about eclipses is susceptible to declining with distance between explanans and explanandum. This is in stark contrast to the case of physical constants, which are, by definition, unchanging in time and space, such that $\lambda_y \approx 0$.

What is $\lambda$ in the case of eclipses? We will not examine here the possible effects of distance in methods, and we will only estimate the knowledge loss rate over time. We can do so by taking the most distant prediction made using the JPL DE405 ephemeris for a total solar eclipse: the one that will occur on 26 April AD 3000 [21]. The estimated error is approximately $7.8°$ of longitude, which at the predicted latitude of peak eclipse ($21.1°\,\text{N}$, $18.4°\,\text{W}$) corresponds to an error of approximately 815 km in either direction. Therefore, the estimated $K$ for predicting an eclipse 982 years from now is

$$K(E_{3000}; X, \tau) \approx \frac{\log(510\,072\,000) - 2\log(815)}{\log(510\,072\,000)} = 0.331. \tag{3.9}$$

Solving $K(E_{3000}; X, \tau) = K(E_{2017}; X, \tau) \times 2^{-\lambda \times 982}$ yields a knowledge loss rate of

$$\lambda_t = 0.0015 \tag{3.10}$$

per year. Which corresponds to a knowledge half life of $\lambda^{-1} \approx 667$ years. Therefore our knowledge about the position of eclipses, based on the JPL DE405 methodology, is halved for every 667 years of time-distance to predictions.

## 3.3. How much progress is a research field making?

*Problem:* Knowledge is a dynamic quantity. Research fields are known to be constantly evolving, splitting and merging [23]. As evidence cumulates, theories and methodologies are modified, enlarged or simplified, and may be extended to encompass new explananda and explanantia, or conversely may be re-defined to account more accurately for a narrower set of phenomena. To what extent do these dynamics determine scientific progress?

*Answer:* Progress occurs if and only if the following condition is met:

$$n_X H(X') - \log p(\tau') < n_Y H(Y) \frac{k' - k}{kh} \tag{3.11}$$

in which $H(X') \equiv \Delta H(X)$ and $-\log p(\tau') \equiv -\Delta \log p(\tau)$ are expansions or reductions of explanantia, and $k = (H(Y) - H(Y|X, \tau))/H(Y)$, $k' = (H(Y) - H(Y|X, X', \tau, \tau'))/H(Y)$, $h = n_Y H(Y)/(n_Y H(Y) + n_X H(X) - \log p(\tau))$ (see appendix F).

*Explanation:* Knowledge occurs when progressively larger explananda are accounted for by relatively smaller explanantia. This is the essence of the process of consilience, which has been recognized for a long time as the fundamental goal of the scientific enterprise [24]. Consilience drives progress at all levels of generality of scientific knowledge. At the research frontier, where new research fields are being created by identifying new explananda and/or new combinations of explanandum and explanans, $K$ grows by a process of 'micro-consilience'. A 'macro-consilience' may be said to occur when knowledge-containing systems are extended and unified across fields, disciplines and entire domains. Equation (3.11) quantifies the conditions for consilience to occur both at the micro- and macro-level.

The inequality (3.11) is satisfied under several conditions. First, when the explanantia $X'$ and/or $\tau'$ produce a sufficiently large improvement in the effect, from $k$ to $k'$. Second, equation (3.11) is satisfied even when explanatory power is lost, i.e. when $k' \leq k$, if $\Delta H(X) - \Delta \log p(\tau)$ is sufficiently negative. This entails that input, theory or methodology are being reduced or simplified. Finally, if $\Delta H(X) - \Delta \log p(\tau) = 0$, condition (3.11) is satisfied provided that $k' > k$, which would occur by expansion of the explanandum. In all cases, the conditions for consilience are modulated by the extent of application of the theories themselves, quantified by the $n_X$ and $n_Y$ indices.

### 3.3.1. Example 1: Evolutionary models of reproductive skew

Reproductive skew theory is an ambitious attempt to explain reproductive inequalities within animal societies according to simple principles derived from kin selection theory ([25] and references within). In its earliest formulation, reproductive skew was predicted to be determined by a 'transactional' dynamic between dominant and subordinate individuals, according to the condition,

$$p_{\min} = \frac{x_s - r(k - x_d)}{k(1 - r)} \tag{3.12}$$

in which $p_{\min}$ is the minimum proportion of reproduction required by the subordinate to stay, $x_s$ and $x_d$ are the number of offspring that the subordinate and dominant, respectively, would produce if breeding independently, $r$ is the genetic relatedness between subordinate and dominant and $k$ is the productivity of the group. The theory was later expanded to include an alternative 'compromise' model approach, in which skew was determined by direct intra-group conflict. Subsequent elaborations of this theory have extended its range of possible conditions and assumptions, leading to a proliferation of models whose overall explanatory value has been increasingly questioned [25].

We can use equation (3.11) to examine the conditions under which introducing a new parameter or a new model would constitute net progress within reproductive skew theory, using data from a comprehensive review [25]. In particular, we will focus on one of the earliest and most stringent predictions of transactional models, which concerns the correlation between skew and dominant-subordinate genetic relatedness. Contradicting earlier reported success [26], empirical tests in populations of 21 different species failed to support unambiguously transactional models in all but one case (data taken from table 2.2 in [25]).

Since this analysis is intended as a mere illustration, we will make several simplifying assumptions. First, we will assume that all parameters in the model are measurable to two significant digits, and that their prior expected distributions are uniform (in other words, any group from any species may exhibit a skew and relatedness ranging from 0.00 to 0.99, and individual and group productivities ranging from 0

to 99). Therefore, we assume that each of these parameters has an information content equal to $2\log 10 = 6.64$ bits. Second, we will assume that the data reported by [25] are a valid estimate of the average success rate of reproductive skew theory in any non-tested species. Third, we will assume that all of the parameters relevant to the theory are measured with no error. For example, we assume that for any organism in which a 'success' for the theory is reported, reproductive skew is explained or predicted exactly. Fourth, we will assume that the extent of applications of skew theory, i.e. $n_Y$, is sufficiently large to make the $\tau$ component (which contains a description of equation (3.12) as well as any other condition necessary to make reproductive skew predictions work) negligible. These assumptions make our analysis extremely conservative, leading to an over-estimation of $K$ values.

Indicating with $Y$, $X_s$, $X_d$, $X_r$, $X_k$ the values of $p_{\min}$, $x_s$, $x_d$, $r$, $k$ in equation (3.12), we obtain the value corresponding to the $K$ of transactional models

$$k = \frac{2\log 10 - \frac{20}{21}2\log 10}{2\log 10} = \frac{1}{21} = 0.048 \tag{3.13}$$

and

$$h = \frac{H(Y)}{H(Y) + H(X_s) + H(X_d) + H(X_r) + H(X_k) - \log p(\tau)} = \frac{1}{5 - \dfrac{\log p(\tau)}{n_Y 2\log 10}} \approx 0.2. \tag{3.14}$$

Plugging these values in equation (3.11) and re-arranging, we derive the minimal amount of increase in explanatory power that would justify adding a new parameter input $X'$,

$$k' > k\left(1 + \frac{n_X h H(X')}{n_Y H(Y)}\right) = 0.048\left(1 + 0.2\frac{H(X')}{6.64}\right). \tag{3.15}$$

This suggests, for example, that if $X'$ is a new parameter measured to two significant digits, with $H(X') = 2\log 10$, adding it to equation (3.12) would represent theoretical progress if $k' > 1.2k$, in other words if it increased the explanatory power of the theory by 20%. If instead $X'$ represented the choice between transactional theory and a new model then, assuming conservatively that $H(X') = 1$, we have $k' > 1.03k$, suggesting that any improvement above 3% would justify it.

Did the introduction of a single 'compromise' model represent a valuable extension of transactional theory? The informational cost of expanding transactional theory consists not only in the equations $\tau'$ that need to be added to the theory, but also in the additional binary variable $X'$ that determines the choice between the two models for each new species to which the theory is applied. We will assume conservatively that the choice equals one bit. According to Nonacs & Hager [25], compromise models were successfully tested in 2 out of the 21 species examined. Therefore, the $k = 3/21 = 0.14$ attained by adding a compromise model amply compensated for the corresponding increased complexity of reproductive skew theory.

The analysis above refers to results for tests of reproductive skew theory across groups within populations. When comparing the average skew of populations, conversely, transactional models were compatible with virtually all of the species tested, especially with regard to the association of relatedness with reproductive skew [25]. In this case, if we interpret these data as suggesting that $k \approx 1$, i.e. that transactional models are compatible with every species encountered, then progress within the field (the theory) could only be achieved by simplifying equation (3.12). This could be obtained by removing or recoding the parameters with the lowest predictive power, or by deriving the theory in question from more general theories. The latter is what the authors of the review did, by suggesting that the cross-population success of the theory is explainable more economically in terms of kin selection theory, from which these models are derived [25].

These results are merely preliminary and likely to over-estimate the benefits of expanding skew theory. In addition to the conservative assumptions made above, we have assumed that only one transactional model and one compromise model exist, whereas in reality several variants of these models have been produced, which entails that the choice $X'$ is not simply binary, and therefore $H(X')$ is likely to be larger than 1. Moreover, we have assumed that the choice between transactional and compromise models is made *a priori*, for example based on some measurable property of organisms that tells beforehand which type of model applies. If the choice is made *after* the variables are known then the costs of this choice have to be accounted for, with potentially disastrous consequences (§3.6).

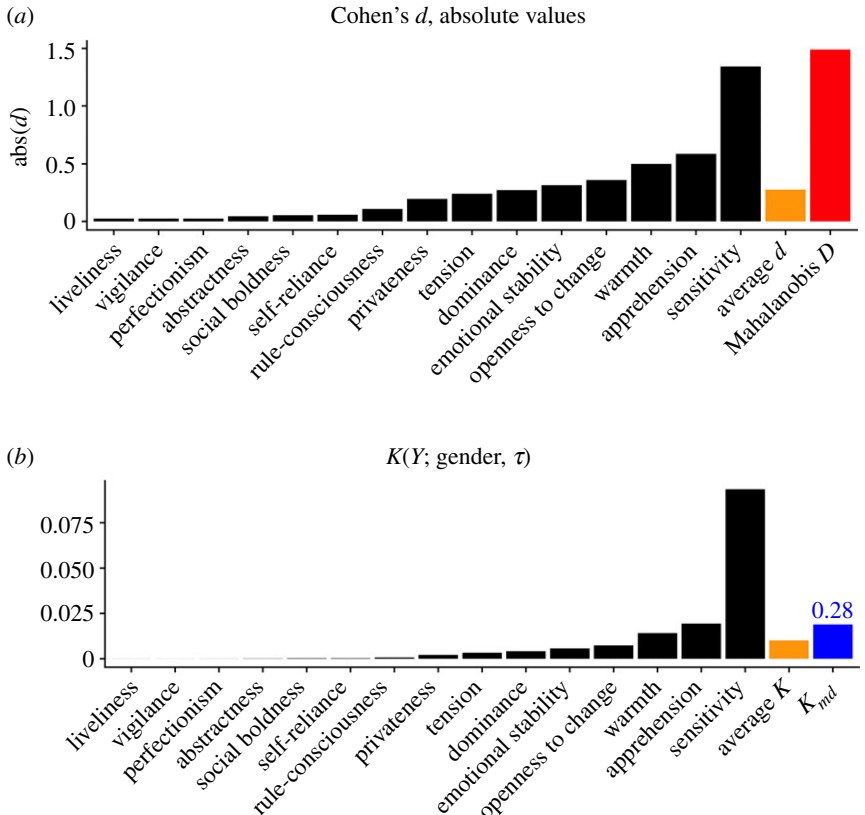

**Figure 6.** Uni- and multivariate analyses of gender differences in personality factors. (*a*) Cohen's *d* and Mahalanobis *D* calculated in [28]. (*b*) *K* values calculated on a dataset of one million individuals, reproduced using the covariance matrices for males and females estimated in [28]. Orange bar: average of unidimensional *K* values. Blue bar: $K_{md}$, calculated assuming that all factors are orthogonal, as in equation (3.16). The number above the blue bar represents the rescaled values $H(Y)K_{md}$. For further details, see text.

### 3.3.2. Example 2: gender differences in personality factors

In 2005, psychologist Janet Hyde proposed a 'gender similarity hypothesis', according to which men and women are more similar than different on most (but not all) psychological variables [27]. According to her review of the literature, human males and females exhibit average differences that, for most measured personality factors, are of small magnitude (i.e. Cohen's *d* less than or equal to 0.35). Assuming that these traits are normally distributed within each gender, this finding implies that the empirical distributions of male and female personality factors overlap by more than 85% in most cases.

The gender similarity hypothesis was challenged by Del Giudice *et al.* [28], on the basis that, even assuming that the distributions of individual personality factors do overlap substantially, the joint distribution of these factors might not. For example, if Mahalanobis distance *D*, which is the multivariate equivalent of Cohen's *d*, was applied to 15 psychological factors measured on a large sample of adult males and females, the resulting effect was large ($D = 1.49$), suggesting an overlap of 30% or less [28] (figure 6*a*).

The multivariate approach proposed by Del Giudice was criticized by Hyde primarily for being 'uninterpretable' [29], because it is based on a distance in 15-dimensional space, calculated from the discriminant function. This suggests that such a measure is intended to maximize the difference between groups. Indeed, Mahalanobis *D* will always be larger than the largest unidimensional Cohen's *d* included in its calculation (figure 6*a*).

The *K* function offers an alternative approach to examine the gender differences vs similarities controversy, using simple and intuitive calculations. With *K*, we can quantify directly the amount of knowledge that we gain, on average, about an individual's personality by knowing their gender. Since most people self-identify as male and female in roughly similar proportions, knowing the

gender of an individual corresponds to an input of one bit. In the most informative scenario, males and females would be entirely separated along any given personality factor, and knowing gender would return exactly one bit along any dimension. Therefore, we can test to what extent the gender factor is informative by setting up a one-bit information in each of the explananda: we divide the population in two groups, corresponding to values above and below the median for each dimension.

The resulting measure, which we will call 'multi-dimensional $K$' are psychologically realistic and intuitively interpretable and are calculated as

$$K_{md} \equiv \frac{\sum_{i=1}^{z} H(Y_i) - \sum_{i=1}^{z} H(Y_i|X, \tau_{Y_i|X})}{\sum_{i=1}^{z} H(Y_i) + H(X) - \sum_{i=1}^{z} \frac{\log p(\tau_{Y_i|X})}{n_Y}} \tag{3.16}$$

in which $z$ is the number of dimensions considered and $\tau_{Y_i|X}$ is the theory linking gender to each dimension $i$.

Note that, whereas the maximum value attainable by the unidimensional $K$ is 1/2, that of $K_{md}$ is 15/16 = 0.938. This value illustrates how, as the explanandum is expanded to new dimensions, $K_{md}$ could approach indefinitely the value of 1, value that would entail that input about gender yields complete information about personality. Whether it does so, and therefore the extent to which applying the concept of gender to multiple dimensions represents progress, is determined by conditions in (3.11).

To illustrate the potential applications of these measures, the values of $K$, average $K$, as well as $K_{md}$ were calculated from a dataset ($N=10^6$) simulated using the variance and covariance of personality factors estimated by [28,30]. All unidimensional personality measures were split in lower and upper 50% percentile, yielding one bit of potentially knowable information. In $K_{md}$, these were then recombined, yielding a 15-bit total explanandum.

Figure 6b reports results of this analysis. As expected, the unidimensional $K$ values are closely correlated with their corresponding Cohen's $d$ values (figure 6a,b, black bars). However, the multi-dimensional $K$ value offers a rather different picture from that of Mahalanobis $D$. $K_{md}$ is considerably smaller than the largest unidimensional effect measured, and is in the range of the second-largest effect. Indeed, unlike Mahalanobis $D$, $K_{md}$ is somewhat intermediate in magnitude, although larger than a simple average (given by the orange bar in figure 6b).

Therefore, we conclude that the overall knowledge conferred by gender about the 15 personality factors together is comparable to some of the larger, but not the largest, values obtained on individual factors. This is a more directly interpretable comparison of effects, which stems from the unique properties of $K$.

We can also calculate the absolute number of bits that are gained about an individual's personality by knowing a person's gender. For the unidimensional variables, where we assumed $H(Y) = 1$, this is equivalent to the $K$ values shown. For the multi-dimensional $K_{md}$, however, we have to multiply by 15, obtaining 0.28 (figure 6b). This value is larger than the largest unidimensional $K$ value of approximately 0.08, and suggests that, at least among the 15 dimensions considered, receiving one bit of input about an individual's gender allows to save at least one-quarter of a bit in predicting their personality.

These results are intended as mere illustrations of the potential utility of the methods proposed. Such potential was under-exploited in this particular case, because the original data were not available, and therefore the analysis was based on a re-simulation of data derived from estimated variances and co-variances. Therefore, this analysis inherited the assumptions of normality and linear covariance that are necessary but limiting components of traditional multivariate analyses, and were a source of criticism for data on gender differences too [29].

Unlike ordinary multivariate analyses, a $K$ analysis requires no distributional assumptions. If it were conducted on a real dataset about gender, the analysis might reveal nonlinear structures in personality factors, and/or identify the optimal level of resolution at which each dimension of personality ought to be measured (§2.3.6). This would yield a more accurate answer concerning how much knowledge about people's personality is gained by knowing their gender.

### 3.3.3. Example 3: Does cumulative evidence support a hypothesis?

The current tool of choice to assess whether the aggregate evidence of multiple studies supports an empirical hypothesis is meta-analysis, in which effect sizes of primary studies are standardized and pooled in a weighted summary [13]. The $K$ function may offer a complementary tool in the form of a cumulative $K$, $K_{cum}$. This is conceptually analogous to the $K_{md}$ described above but, instead of

assuming that the various composing explananda lie on orthogonal dimensions and the explanans is fixed, it assumes that both explanandum and explanans lie on single dimensions, and their entropy results from a mixture of different sources.

It can be shown that, for a set of RVs $Y_1$, $Y_2 \ldots Y_m$ with probability distributions $p_{Y_1}(\cdot), p_{Y_2}(\cdot) \ldots p_{Y_m}(\cdot)$, the entropy of their mixed distribution $\sum w_i p_{Y_i}$ is given by

$$H\left(\sum_{i \leq m} w_i p_{Y_i}\right) = \sum_{i \leq m} w_i H(Y_i) + \sum_{i \leq m} w_i D\left(p_{Y_i} \| \sum_{i \leq m} w_i p_{Y_i}\right) \equiv \overline{H(Y)} + \overline{d_Y}, \tag{3.17}$$

where the right-hand terms are a notation introduced for convenience, and $D(p_{Y_i} \| \sum_{i \leq m} w_i p_{Y_i})$ represents the Kullback–Leibler divergence between each RV and the mixed distribution.

For sequences, and particularly for those representing the theory $\tau$, the mixture operates on an element-by-element basis. For example, if $T_{i,p}$ and $T_{j,p}$ are the RVs representing choice $p$ in $\tau_i$ and $\tau_j$, respectively, a mixture of $\tau_i$ and $\tau_j$ will lead to choice $p$ now being represented by a RV $T_{ij,p}$, say, which has still uniform distribution and whose alphabet is the union set of the mixed alphabets, $\mathcal{T}_{ij,p} = \{\mathcal{T}_{i,p} \cup \mathcal{T}_{j,p}\}$.

Remembering that the minimum alphabet size of any element of a $\tau$ is 2, it can be shown that, if for example, $\tau_i = (\tau_{i,1}, \tau_{i,2} \ldots \tau_{i,l})$ and $\tau_j = (\tau_{j,1}, \tau_{j,2} \ldots \tau_{j,m})$ are two sequences of length $l$ and $m$ with $l > m$, their mixture will yield the quantity

$$\overline{\tau} + \overline{d_\tau} \equiv l + \sum_{u \leq l} \log \frac{|\mathcal{T}_u|}{2} \tag{3.18}$$

in which $|\mathcal{T}_u|$ is the size of the alphabet resulting from the mixture. For the mixing of $s$ theories $\{\tau_1, \tau_2 \ldots \tau_s\}$, $\overline{\tau}$ will be equal to the description length of the longest $\tau$ in the set. Indicating the latter with $l^*$, we have

$$0 \leq \overline{d_\tau} \leq l^* \log \frac{s+1}{2} \tag{3.19}$$

with the right-hand side equality occurring if the $s$ sequences have equal length and are all different from each other.

For example, if the methodology $\tau_i = (\text{'randomized'}, \text{'human'}, \text{'female'})$ is mixed with $\tau_j = (\text{'randomized'}, \text{'human'}, \text{'male} + \text{female'})$, the resulting mixture would have composing RVs $T_1 = \{\text{'randomized'}, \text{'not'}\}$, $T_2 = \{\text{'human'}, \text{'not'}\}$, $T_3 = \{\text{'female'}, \text{'male} + \text{female'}, \text{'not'}\}$, and its information content would equal $-\log(1/2) - \log(1/2) - \log(1/3) = 3.58$ or equivalently $\overline{\tau} + \overline{d_\tau} = 3 + \log(3/2) = 3 + 0.58$.

Therefore, the value of the cumulative $K$ is given by

$$K_{\text{cum}} \equiv \frac{n_Y(\overline{H(Y)} - \overline{H(Y|X, \tau)} + \overline{d_Y} - \overline{d_{Y|X,\tau}})}{n_Y \overline{H(Y)} + n_X \overline{H(X)} + \overline{\tau} + n_Y \overline{d_Y} + n_X \overline{d_X} + \overline{d_\tau}} \tag{3.20}$$

in which the $\overline{d}$ terms represent the average divergences from the mixed expananda or explanatia. Equation (3.20) is subject to the same conditions of equation (3.11), which will determine whether the cumulative knowledge (e.g. a cumulative literature) is overall leading to an increase or a decrease of knowledge.

The peculiarity of equation (3.20) lies in the presence of additional divergence terms, which allow knowledge to grow or decrease independently of the weighted averages of the measured effects. In particular, ignoring the repetition terms which are constant,

$$K_{\text{cum}} \geq \overline{K} \Leftrightarrow \overline{d_{Y|X,\tau}} \leq (1 - \overline{K})\overline{d_Y} - \overline{K}(\overline{d_X} + \overline{d_\tau}) \tag{3.21}$$

with $\overline{K} = (\overline{H(Y)} - \overline{H(Y|X, \tau)})/(\overline{H(Y)} + \overline{H(X)} + \overline{\tau})$ constituting the $K$ value obtained by the simple averages of each term. This property, combined with the presence of a cumulative theory/methodology component $\overline{\tau} + \overline{d_\tau}$ that penalizes the cumulation of diverse methodologies, makes $K_{\text{cum}}$ behave rather differently from ordinary meta-analytical estimates.

Figure 7 illustrates the differences between meta-analysis and $K_{\text{cum}}$. Like ordinary meta-analysis, $K_{\text{cum}}$ depends on the within- and between-study variance of effect sizes. Unlike meta-analysis, however, $K_{\text{cum}}$ decreases if the methodology of aggregated studies is heterogeneous, independent of the statistical heterogeneity that is observed in the effect sizes (that is, $K$ can decrease even if the effects are statistically homogeneous). Moreover, $K_{\text{cum}}$ can increase even when all included studies report null

# comparison of meta-analysis and K$_{cum}$

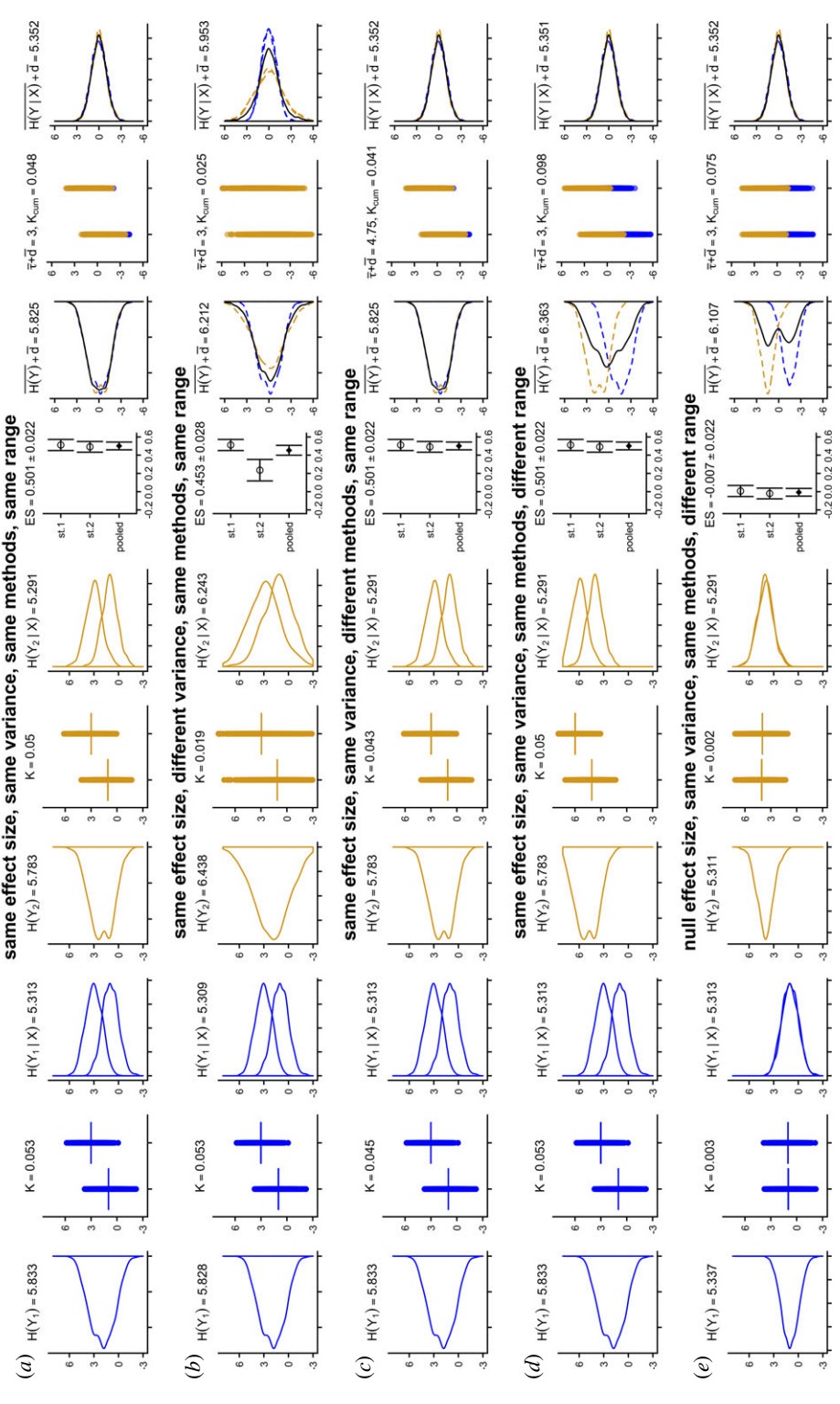

**Figure 7.** Comparison between meta-analysis and cumulative K analysis. From left to right, the graph shows the simulated data for two imagined studies (blue and golden, respectively) with different assumptions of variance and effect size, then the corresponding meta-analytical summary, and then the corresponding K analysis, with values calculated as in equation (3.20). The entropy value, meta-analytical summary effect size or K are indicated above each corresponding figure. See text for further discussion.

findings, if the aggregated studies cover different ranges of the explanandum, making the cumulative explanandum larger.

Note that we have not specified how the weights underlying the mixture are calculated. These may consist in an inverse-variance weighting, as in ordinary meta-analysis, or could be computed based on other epistemologically relevant variables, such as the relative divergence of studies' methodologies. The latter approach would offer an alternative to the practice of weighting studies by measures of quality, a practice that used to be common in meta-analysis and has now largely been abandoned due to its inherent subjectivity.

## 3.4. How reproducible is a research finding?

*Problem:* The concept of 'reproducibility' is the subject of growing concerns and expanding research programmes, both of which risk being misled by epistemological confusions of at least two kinds. The first source of confusion is the conflation of the reproducibility of methods and that of results [31]. The reproducibility of methods entails that identical results are reproduced if the same data is used, indicating that data and methods were reported completely and transparently. The reproducibility of results entails that identical results are obtained if the same methods are applied to new data. Whereas the former is a relatively straightforward issue to assess and to address, the latter is a complex phenomenon that has multiple causes that are hard to disentangle. When a study is reproduced using new data, i.e. sampling from a similar but possibly not identical population and using similar but not necessarily identical methods, results may differ for reasons that have nothing to do with flawed methods in the original studies. This is a very intuitive idea, which, however, struggles to be formally included in analyses of reproducibility. The latter typically follow the meta-analytical paradigm of assuming that, in absence of research and publication biases, results of two studies ought to be randomly distributed around a 'true' underlying effect.

The second source of confusion comes from treating the concept of reproducibility as a dichotomy—either a study is reproducible/reproduced or it is not—even though this is obviously a simplification. A scientific finding may be reproduced to varying degrees, depending on the nature of what is being reproduced (e.g. is it an empirical datum? A relation between two operationalized concepts? A generalized theory?) and contingent upon innumerable characteristics of a research which include not just how the research was conducted and reported, but also by characteristics of the research's subject matter and general methodology.

How can we distinguish the reproducibility of methods and results and define them in a single, continuous measure?

*Answer:* The relation between a scientific study and one that reproduces it is described by the relation

$$K_r = KA^{-\lambda \cdot d} \tag{3.22}$$

in which $K_r$ is the result of a replication study conducted at a study-specific 'distance' (information divergence) given by the inner-product of a vector $d : [d_Y, d_X, d_{\tau_1}, d_{\tau_2} \cdots]$ of distances and a vector $\lambda : [\lambda_Y, \lambda_X, \lambda_{\tau_1}, \lambda_{\tau_2} \ldots]$ of corresponding loss rates.

*Explanation:* A study that attempts to reproduce another study is best understood as a new system that is at a certain 'distance' from the previous one. An identical replication is guaranteed to occur only if the exact same methods and exact same data are used, in which case the divergence between the two systems is likely to be zero on all dimensions, and the resulting $K$ (and corresponding measure of effect size produced by the study's results) is expected to be identical. Note that even this is an approximation, since the instruments (e.g. hardware and software) used to repeat the analyses may be different, and this could in principle generate some discrepancies.

If attainable at all, a divergence of zero is only really likely to characterize the reproducibility of methods and is unlikely to occur in the reproducibility of results (in which new data are being collected). In the latter, different characteristics in the population being sampled ($d_Y$), the measurements or interventions made ($d_X$) and/or other critical choices made in the conduction of the study ($d_\tau$) may affect the outcome. Contrary to what is normally assumed in reproducibility studies, these differences cannot be assumed to exert random and symmetric influences on the result. The more likely direction of change is one of reduction: divergences in any element of the system, particularly if not dictated by the objective to increase $K$, are likely to introduce noise in the system, thus obfuscating the pattern encoded in the original study.

Section 2.3.5 showed how the exponential function (3.22) described the decline of a system's $K$ due to divergences in subject matter or methodology. In practical terms, a divergence vector will consist in classifiable, countable differences in components of the methods used and/or characteristics of subject matter that, based on theory and prior data, are deemed likely to reduce the level of $K$ by some proportional factor.

Applications of equation (3.22) to individual cases require measuring study-specific divergences in explanandum and explanans and their corresponding loss rates. However, the universality of the function in equation (3.22) allows us to derive general, population-level predictions about reproducibility, as the example below illustrates.

### 3.4.1. Example: How reproducible is Psychological Science?

The Reproducibility Initiative in Psychology (RIP) was a monumental project in which a consortium of laboratories attempted to replicate 100 studies taken from recent issues of three main psychology journals. Results were widely reported in the literature and mass media as suggesting that less than 40% of studies had been replicated, a figure deemed to be disappointingly low and indicative of significant research and publication biases in the original studies [32]. This conclusion, however, was questioned on various grounds, including: limitations in current statistical approaches used to predict and estimate reproducibility (e.g. [33–35]), methodological differences between original and replication studies [36], variable expertise of the replicators [37] and variable contextual sensitivity of the phenomena studied [38,39]. The common element behind all these concerns is that the replication study was not actually identical to the original but diverged in details that affected the results unidirectionally. This is the phenomenon that equation (3.22) can help to formalize, predict and estimate empirically.

In theory, each replication study in the RIP could be examined individually using equation (3.22), but doing so would require field-specific information on the impact that various divergences may have on the results. This fine-grained analysis is not achievable, at least presently, because the necessary data are not available. However, we can use equation (3.22) to formulate a general prediction about *the shape of the distribution* of results of a reproducibility study, under varying frequencies and impacts of errors.

Figure 8 simulated the distribution of effect sizes (here shown as correlation coefficients derived from the corresponding $K$) that would be observed in a set of replication studies, depending on their average distances $d$ and impacts $\lambda$ from an original or ideal study. Distances were assumed to follow a Poisson distribution, with a mean of 1, 5 and 20, respectively. The impact of these distances was increased moving from the top to the bottom row, by assuming the values of $\lambda$ illustrated in the top-most panel. The dotted vertical line in each plot reports the initial value of $K$ (i.e. the left-hand side of equation (3.22)), whereas the solid vertical line shows the mean of the distribution of results.

The figure can be given different interpretations. The distances simulated in figure 8 may be interpreted as between-study differences in the explanandum or input (e.g. cultural differences in the studied populations), between-study differences in methodological choices, or as study-specific methodological errors and omissions, or a combination of all three. The dotted line may represent either the result of the original study or the effect that would be obtained by an idealized study for which the $K$ is maximal and from which all observed studies are at some distance.

Irrespective of what we assume these distances to consist in, and to the extent that they represent a loss of information, their frequency and impact profoundly affect the expected distribution of replication results. The distribution is compact and right-skewed when distances are few and of minor impact (top-left). As the number of such minor-impact distances grows, the distribution tends to be symmetrical and bell-shaped (top-right). Indeed, if the number of distances was increased further, the shape would resemble that of a Gaussian curve (mirroring the behaviour of a Poisson distribution). In such a (special) case, the distribution of replication results would meet the assumption of symmetrical and normally distributed errors that current statistical models of reproducibility make. This condition, however, is a rather extreme case and by no means the most plausible. As the impact of distances increases in magnitude, the distribution tends to become left-skewed, if distances are numerous, or bimodal if they are few (bottom-right and bottom-left, respectively).

This suggests that the conditions typically postulated in analyses of reproducibility (i.e. a normal distribution around the 'true' or the 'average' effect in a population of studies) are only realized under the special condition in which between-studies differences, errors or omissions in methodologies are numerous and of minor impact. However, when important divergences in explanandum or explanans occur (presumably in the form of major discrepancies in methods used, populations examined etc.),

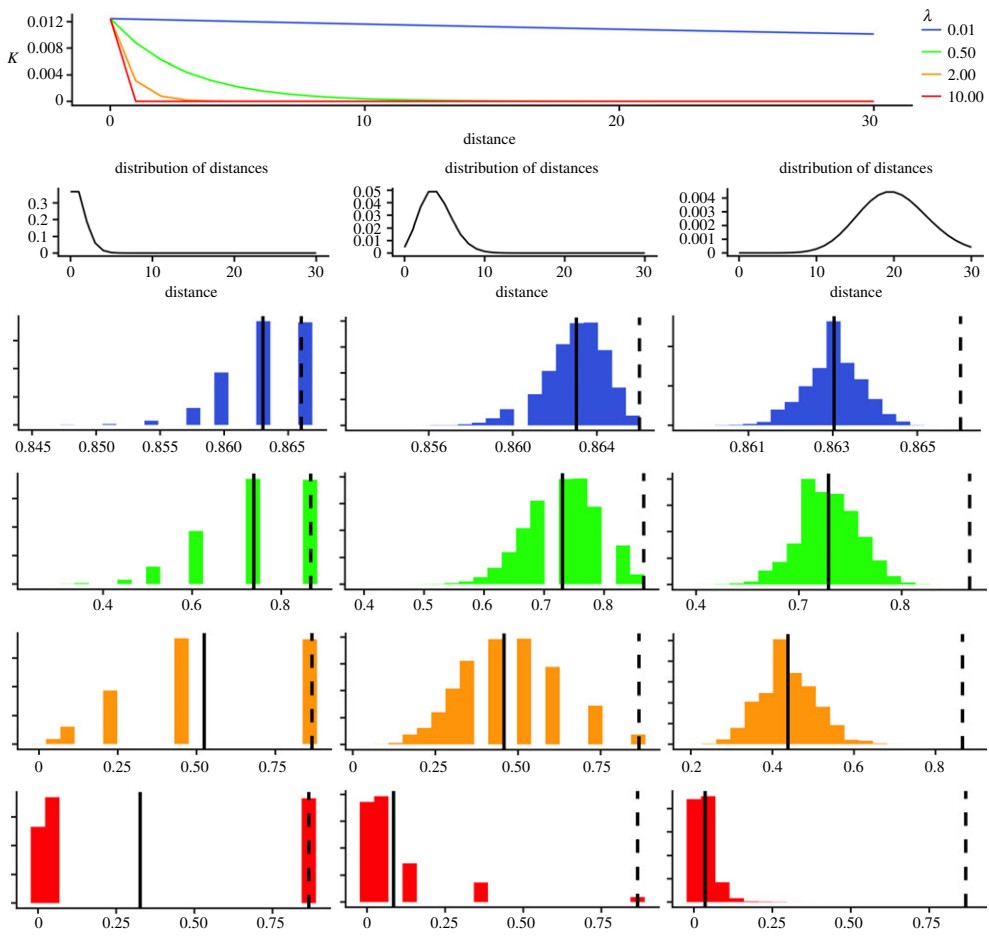

**Figure 8.** Simulated distribution of results of reproducibility studies, under varying conditions of distances (e.g. number of differences in methodologies), *d*, and average impact per distance λ. The top panel shows how *K* declines as the number of divergences increases, depending on different values of λ. Panels in the second row show the probability distribution of the simulated distances (Poisson distributions, with mean 1, 5 and 20, respectively). The nine panels below show the distribution of correlation coefficients of reproducibility studies under each combination of number of distances and their impact. The impact is colour coded as in the top panel. For further discussion see text.

the distribution becomes increasingly asymmetrical and concentrated around null results and may either be left-skewed or bimodal, depending on whether the number of elements subject to divergence is large or small.

Data from the RIP support these predictions. Before undertaking the replication tests, the authors of the RIP had classified the studies by level of expertise required to replicate them. As figure 9 illustrates, replication results of studies that were deemed to require moderate or higher expertise are highly concentrated around zero, with a small subset of studies exhibiting medium to large effects. This distribution is markedly different from that of studies that required null or minimal expertise, which was unimodal instead. Note how the distribution of original results reported by both categories of studies are, instead, undistinguishable in shape. Additional differences between distributions might be explained by a classification of the stability of the explanandum or explanans (e.g. the contextual sensitivity suggested by Van Bavel *et al.* [39]).

Although preliminary, these results suggest that a significant cause of reproducibility 'failures' in the RIP may have been high-impact divergences in the systems or methodologies employed by the replicating studies. These divergences may have occurred despite the fact that many authors of the original studies had contributed to the design of the replication attempts. A significant component of a scientists' expertise consists in 'tacit knowledge' [40], manifested in correct methodological choices that are not codified or described in textbooks and research articles, and that are unconsciously acquired by researchers through practice. Therefore, authors of the original studies might have taken

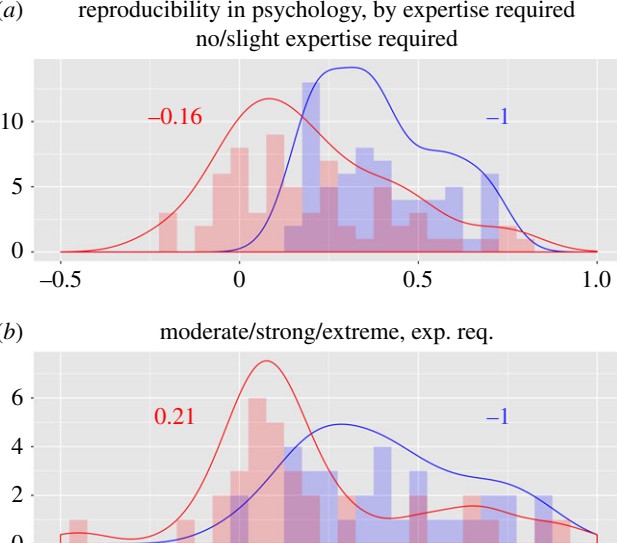

**Figure 9.** Distributions of correlation coefficients reported by the studies examined in the Reproducibility Initiative in Psychology [32]. Blue: effect sizes originally reported. Red: results of replications. Numbers report the kurtosis of each distribution.

for granted, or unwittingly overlooked, important aspects of their own research design when instructing the RIP replicators. The latter, even if professionally prepared, might have lacked sufficient expertise about the systems that are the object of the replication attempt, and may therefore have made 'tacit errors' that neither they or the authors of the original studies were able to document.

It may still be the case that *p*-hacking and selective publication had affected some of the studies examined by RIP. However, if research biases were the sole factor leading to low reproducibility, then the two distributions in figure 9 should look similar. The fact that studies requiring higher level of expertise are harder to reproduce ought, in retrospect, not to surprise us. It simply suggests the very intuitive idea that many scientific experiments cannot be successfully conducted by anyone who simply follows the recipe, but need to be conducted by individuals with high levels of expertise about the methodology and the phenomena being studied. This fact still raises important questions about the generalizability of published results and how to improve it, but such questions should be disentangled as much as possible from questions about the integrity and objectivity of researchers.

### 3.5. What is the value of a null or negative result?

*Problem:* How scientists should handle 'null' and 'negative' results is the subject of considerable ambiguity and debate. On the one hand, and contrary to what their names might suggest, 'null' and 'negative' results undoubtedly play an important role in scientific progress, because it is by cumulation of such results that hypotheses and theories are refuted, allowing progress to be made by 'theory falsification', rather than verification, as Karl Popper famously argued [41]. Null and negative results are especially important in contexts in which multiple independent results are aggregated to test a single hypothesis, as is done in meta-analysis [42].

On the other hand, as Popper himself had noticed, the falsifiability of a hypothesis is typically suboptimal, because multiple 'auxiliary' assumptions (or, equivalently, auxiliary hypotheses) may not be controlled for. Moreover, it is intuitively clear that a scientific *discovery* that leads to useful knowledge is made when a new pattern is *identified*, and not merely when a pattern is proved not to subsist.

This is why, if on the one hand there are increasing efforts to counter the 'file-drawer problem', on the other hand there are legitimate concerns that these efforts might generate a 'cluttered office' problem, in which valuable knowledge is drowned in a chaotic sea of uninformative publications of null results [43]. The problem is that the value of null and negative results is context-specific. How can we estimate it?

*Answer:* The knowledge value of a null or negative result is given by

$$K_{\text{null}} \leq \frac{h}{H(Y)} \log \frac{|\mathcal{T}|}{|\mathcal{T}| - 1} \tag{3.23}$$

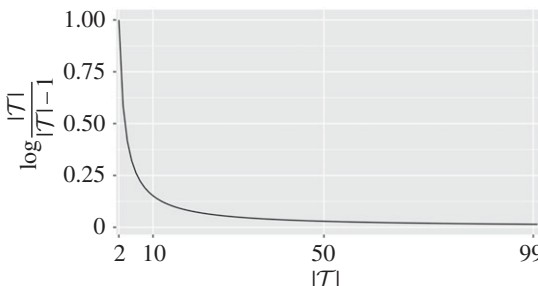

**Figure 10.** Relation between $|\mathcal{T}|$, the total number of hypotheses/assumptions entering a study and the main multiplicative factor that determines the upper limit to $K_{\mathrm{null}}$ in equation (3.23).

in which $h = H(Y)/(H(T) + H(Y) + H(T))$, $K_{\mathrm{null}}$ is the knowledge gained by the conclusive refutation of a hypothesis, and $|\mathcal{T}|$ is the size of the set of hypotheses being potentially tested (including all unchecked assumptions) in the study. All else equal, the maximum value of $K_{\mathrm{null}}$ declines rapidly as $|\mathcal{T}|$ increases (figure 10).

*Explanation:* Section 2.2.1 described knowledge as resulting from the selection of a $\tau \in \mathcal{T}$, where $\mathcal{T}$ is the a set of possible theories (methodologies) determining a pattern between explanandum and input. These theories can, as usual, be described by a uniform random variable $T$. It can be shown that, because of the symmetry property of the mutual information function,

$$K(Y; X, T) = K(T; Y, X), \tag{3.24}$$

i.e. the information that the set of theories contains about the data is equivalent to the information that the data contains about the theories (see appendix G).

This is indeed how knowledge is attained. A theory $\tau$ is selected among available alternatives because it best fits a data $Y^{n_Y}$, $X^{n_X}$, and ideally maximizes $k_{\mathrm{adj}} - k_{\mathrm{obs}}$ (§2.3.2). The data are obtained by experiment (or experiences) and the process is what we call learning, as it is embodied in the logic of Bayes' theorem, the MDL principle and generally the objective of any statistical inference method. Since no knowledge (including knowledge about a theory) can be obtained in the absence of a 'background' conditioning theory and methodology, a more accurate representation of an experiment entails the specification of an unvarying component which we will indicate as $m$, which quantifies the aspects of the theory and methodology of an experiment that are not subject to uncertainty, and the component for which knowledge is sought, the random variable $T$, which therefore represents the hypothesis or hypotheses being tested by the experiment. The knowledge attained by the experiment is then given by

$$K(T; Y^{n_Y}, X^{n_X}, m) = \frac{h}{H(Y)}(H(T) - H(T|Y, X, m)). \tag{3.25}$$

It follows that the experiment is maximally informative when $H(T)$ is as large as possible and $H(T|Y, X, m) = 0$, that is, when multiple candidate hypotheses are examined and each of them is in one-to-one correspondence with each of possible states of $Y$, $X$.

Real-life experiments depart from this ideal condition in two ways. First, they usually retain uncertainty about the result, $H(T|Y, X, m) > 0$, because multiple alternative hypotheses are compatible with the same experimental outcome. Second, real experiments usually test no more than one hypothesis at a time. This entails that $H(T|Y, X, m)$ rapidly approaches $H(T)$, as the size of the alphabet of $T$ increases (see appendix H). These limitations suggest that, assuming maximally informative conditions in which all tested hypotheses are equally likely and one hypothesis is conclusively ruled out by the experiment, we have $H(T) - H(T|Y = y, X = x, m) = \log|\mathcal{T}| - \log(|\mathcal{T}| - 1)$, which gives equation (3.23).

As intuition would suggest, even if perfectly conclusive, a null finding is intrinsically less valuable than its corresponding 'positive' one. This occurs because a tested hypothesis is ruled out when the result is positive as well as when it is negative, and therefore the value quantified in equation (3.23) is obtained with positive as well as negative results, a condition that we can express formally as $K(T; Y, X, m, T = \tau_1) = K(T; Y, X, m, T = \tau_0)$. Positive results, however, also yield knowledge about a pattern. Therefore, whereas a conclusive rejection of a non-null hypothesis yields at most $K(T; Y, X, m, T = \tau_0) = h/H(Y)$, a conclusive rejection of the null hypothesis in favour of the alternative yields $K(T; Y, X,$

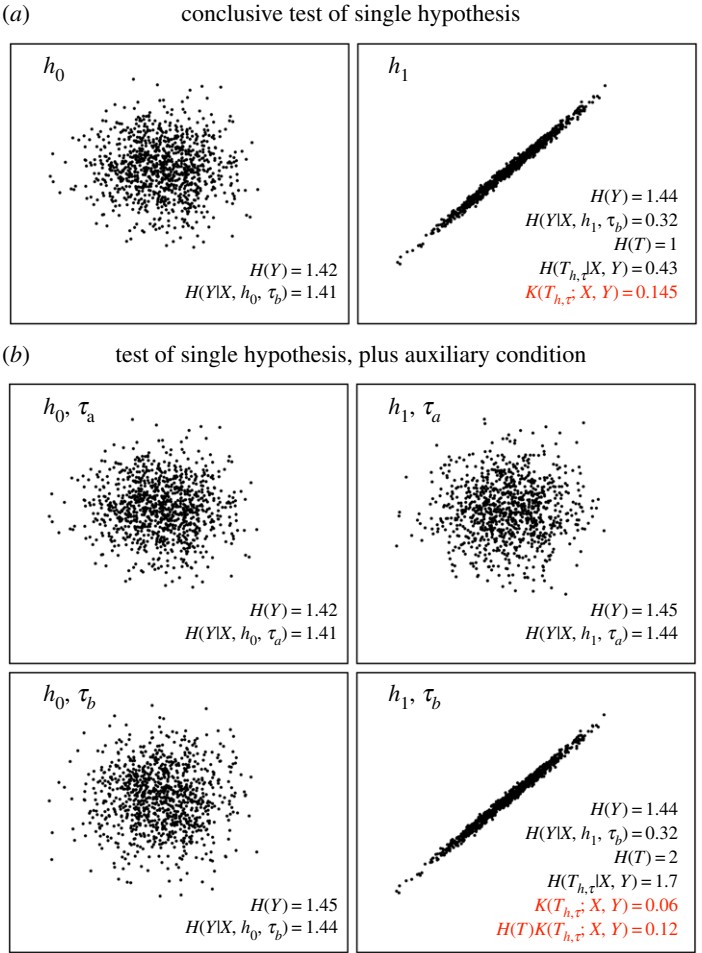

**Figure 11.** *K* analysis of the informativeness of data with regard to a hypothesis *h*, in absence (*a*) or presence (*b*) of a second condition *τ* that modulates results of the test. Numbers report all parameters calculated from the analysis. The R code to generate the data and figure is in electronic supplementary material. See text for further details and discussion.

$m$, $T = \tau_1$) + $K(Y; X, \tau_1) > h/H(Y)$. Perfect symmetry between 'negative' and 'positive' results is only attained in the ideal conditions mentioned above, in which $H(T|Y, X, m) = 0$ and $H(T) = H(Y)$, and therefore each experimental outcome identifies a theory with empirical value and at the same time refutes other theories. This is the scenario in which 'perfect' Popperian falsificationism can operate, and real-life experiments depart from this ideal in proportion to the number $\log(|\mathcal{T}| - 1)$ of auxiliary hypotheses that are not addressed by the experiment.

The departure from ideal conditions is especially problematic in biological and social studies that are testing a fixed 'null' hypothesis $\tau_0$ that predicts $K(Y; X, \tau_0) = 0$ against a non-specified alternative $\tau_1$ for which $K(Y; X, \tau_1) > 0$. First of all, due to noise and limited sample size, $K(Y;X,\tau_0) > 0$. This problem can be substantially reduced by increasing statistical power but can never be fully eliminated, especially in fields in which large sample sizes and high accuracy (resolution) are difficult or impossible to obtain. Moreover, and *regardless of statistical power*, a null result is inherently more likely to be compatible with multiple 'auxiliary' hypotheses/assumptions, which real-life experiments may be unable to control.

### 3.5.1. Example 1: A simulation

To offer a practical example of the theoretical argument made above, figure 11 reports a simulation. The value of $K(T; X, Y)$, i.e. how much we know about a hypothesis given data, was first calculated when a single hypothesis $h_1$ is at stake, and all other conditions are fixed (figure 11*a*). Subsequently, the alphabet of $T$ (the set of hypotheses in the experiment) was expanded to include a second condition, with two

possible states $\tau_a$ or $\tau_b$, the former of which produces a null finding regardless of $h_1$. The state of this latter condition (hypothesis/assumption) is not determined in the experiment. The corresponding value of $K(T; X, Y)$ is measurably lower, even if rescaled to account for the greater complexity of the explanandum (i.e. the number of tested hypotheses, figure 11b).

This is a simple illustration of how the value of negative results depends on the number of uncontrolled conditions and/or possible hypotheses. If field-specific methods to estimate the number of auxiliary hypotheses are developed, the field-specific and study-specific informativeness of a null result could be estimated and compared.

The conclusions reached in this section, combined with the limitations of replication studies discussed in §3.4, may offer new insights into debates over the problem of publication bias and how to solve it. This aspect is briefly discussed in the example below.

### 3.5.2. Example 2: Should we publish all negative results?

Debates on whether publication bias is a bane or boon in disguise recur in the literature of the biological and social sciences. A vivid example was offered by two recent studies that used virtually identical methods and arguments but reached opposite conclusions concerning whether 'publishing everything is more effective than selective publishing of statistically significant results' [44,45].

Who is right? Both perspectives may be right or wrong, depending on specific conditions of a field, i.e. of a research question and a methodology. An explicit but rarely discussed assumption made by most analyses of publication bias is that the primary studies subjected to bias are of 'similar quality'. What this quality specifically consists in is never defined concretely. Nonetheless, it seems plausible to assume that quality, like any other property of studies, will be unequally distributed within a literature, and the level of heterogeneity will vary across fields. This field-specific heterogeneity, however, cannot be overlooked, because it determines the value of $H(T|Y, X, m)$ and $|\mathcal{T}|$, i.e. the falsifiability of the main hypothesis being tested. Therefore, to properly estimate the true prevalence and impact of publication bias and determine cost-effective solutions, the falsifiability of hypotheses needs to be estimated on a case-by-case (i.e. field-specific or methodology-specific) basis.

In general, the analysis above suggests that current concerns for publication bias and investments to counter it are most justified in fields in which methodologies are well codified and hypotheses to be tested are simple and clearly defined. This is likely to be the condition of most physical sciences, in which not coincidentally negative results appear to be valued as much or more than positive results [46,47]. It may also reflect the condition of research in clinical medicine, in which clearly identified hypotheses (treatments) are tested with relatively well-codified methods (randomized controlled trials). This would explain why concerns for publication bias have been widespread and most proactively addressed in clinical medicine [42]). However, the value of negative results is likely to be lower in other research fields, and therefore the cost–benefit ratio of interventions to counter publication bias need to be assessed on a case-by-case basis.

Methods proposed in this article might help us determine relevant field-specific and study-specific conditions. In particular, the statistical relevance of a null result produced by a study with regard to a specified hypothesis is likely to be inversely related to the expected divergence of the study from a standard (or an ideal) methodology and explanandum $\lambda \cdot d$ (§3.4). This effect is in turn modulated by the complexity and flexibility of a field's methodological choices and magnitude of effect sizes, both quantifiable in terms of the $K$ function proposed in this study.

### 3.6. How much knowledge do we lose from questionable research practices?

*Problem:* In addition to relatively well-defined forms of scientific misconduct, studies and policies about research integrity typically address a broader category of 'questionable research practices' (QRP). This is a class of rather loosely defined behaviours such as 'dropping outliers based on a feeling that they were inaccurate', or 'failing to publish results that contradicted one's previous findings'. Behaviours that, by definition, may or may not be improper, depending on the context [48].

Since QRP are likely to be more frequent than outright fraud, it has long been argued that their impact on the reliability of the literature may be very high—indeed, even higher than that of data fabrication or falsification (e.g. [49]). However, besides obvious difficulties in quantifying the relative frequency of proper versus improper QRP, there is little epistemological or methodological basis for grouping together an extremely heterogeneous set of practices and branding them as equally worrying [50]. Setting aside ethical breaches that do not affect the validity of data or results—which

will not be considered here—it is obvious that our concerns for QRP ought to be proportional not simply to the frequency of their use but to the frequency of their use multiplied by the distorting effect on the literature. How can we quantify the impact of misconduct and QRP?

*Answer:* The impact on knowledge of a Questionable Research Practice is given by a 'bias-corrected' $K$ value

$$K_{\mathrm{corr}} \equiv K_u - \frac{h_u}{h_b} B, \tag{3.26}$$

in which $K_u = K(Y; X, \tau)$ is the 'unbiased' $K$, $h_u = n_Y H(Y)/(n_Y H(Y) + n_X H(X) - \log p(\tau))$ and $h_b = n_Y H(Y)/(n_Y H(Y) + n_X H(X) - \log p(\tau) - n_\beta \log p(\beta))$ are the the hardness terms for the study, without and with bias, respectively, and

$$B = \frac{D(Y|X, \tau \| Y|X, \tau, \beta)}{n_Y H(Y) + n_X H(X) - \log p(\tau)} \tag{3.27}$$

is the bias caused by the practice.

*Explanation:* Equation (3.26) is derived by a similar logic to that of predictive success, discussed in §2.3.2. If a research practice is deemed epistemologically improper, that is because it must introduce a bias in the result. This implies that the claim made using the biased practice $\beta$ is different from the claim that is declared or intended: $K(Y; X, \tau, \beta) \neq K(Y; X, \tau)$. Just as in the case of prediction costs, therefore, we can adjust the $K$ value by subtracting from it the costs required to derive the claimed result from the observed one, costs that are here quantified by $B$ (equation (2.26)).

Differently from the case of prediction, however, in the presence of bias the methods employed are of different size. In particular, the bias introduced in the results has required an additional methodology $\beta$. Following our standard approach, we posit that $\beta$ is an element of the alphabet of a uniform random variable $B$. Similarly to $\tau$, $-\log p(\beta)$ is the description length of a sequence of choices and $n_\beta$ will be the number of times these choices have to be made. For example, a biased research design (that is, an ante hoc bias) will have $n_\beta = 1$, and therefore a cost $-\log p(\beta)$ corresponding to the description length of the additional components to be added to $\tau$. Conversely, if the bias is a post hoc manipulation of some data or variables, then $\beta$ may be as simple as a binary choice between dropping and retaining data (see example below), and $n_\beta$ may be as high as $n_Y$ or higher. The term $h_u/h_b$ quantifies the relative costs of the biased methodology.

An important property of equation (3.26) is that the condition holds regardless of the direction of the bias. The term $B$ is always non-negative, independent of how results are shifted. Therefore, a QRP that nullified an otherwise large effect (in other words, a bias against a positive result) would require a downwards correction just as one that magnified it.

### 3.6.1. Example 1: Knowledge cost of data fabrication

The act of fabricating an entire study, its dataset, methods, analysis and results can be considered an extreme form of ante hoc bias, in which the claimed effect was generated entirely by the methods.

Let $\beta$ represent the method that fabricated the entire study. By assumption, the effect observed without that method is zero, yielding

$$K_{\mathrm{corr}} = -\frac{h_u}{h_b} B < 0. \tag{3.28}$$

Hence, an entirely fabricated study yields no positive knowledge and yields indeed *negative* knowledge. This result suggests a solution to an interesting epistemological conundrum raised by the scenario in which a fabricated study reports a true fact: if independent, genuine studies confirm the made-up finding, then technically the fabricated study did no damage to knowledge. Shall we therefore conclude that data fabrication can help scientific progress?

Equation (3.26) may shed new light on this conundrum. We can let $K$ represent the amount of genuine knowledge attained within a field. The fabricated study's $K_{\mathrm{corr}}$ is then $K - (h_u/h_b)B \leq 0$, because $B = K$ and $h_u > h_b$. The extra information costs of fabricating the entire study generate a net loss of information, even if the underlying claim is correct.

### 3.6.2. Example 2: Knowledge cost of arbitrarily dropping data points

Let's imagine a researcher who collected a sample of $n$ data points and made a claim $K(Y^n; X^n, \tau) > 0$ without explicitly declaring that during the analysis she had dropped a certain number $n_\beta$ of data points which made her results look 'better'—i.e. her $K$ appear larger than it is. How egregious was this behaviour?

From equation (3.26), we derive the minimum conditions under which a bias is tolerable ($K_{corr} > 0$) as

$$K(Y; X, \tau) > \frac{h_u}{h_b} B. \tag{3.29}$$

The choice to drop or not a data point is binary, and therefore $-\log p(\beta) = 1$. In the best-case scenario, the researcher identified possible outliers based on a conventional threshold of $3\sigma$, and was therefore confronted with the choice of dropping only 0.3% of her data points, i.e. $n_\beta = 0.003n$. This leads to $h_u/h_b \approx 1$ and the simplified condition, $K > B$, in which the bias has to be smaller than the total effect reported. For $B \geq K$ to occur under these conditions (in other words, to generate the full reported effect by dropping no more than 0.3% of data points), it has to be the case that either the reported effect $K$ was extremely small, and therefore unlikely to be substantively significant, or that the dropped outliers were extremely deviant from the normal range of data. In the latter case, the outliers ought to have been removed and, if naively retained in the dataset, their presence and influence would not go unnoticed to the reader. Therefore, arbitrariness in dropping statistical outliers has a minor impact on knowledge.

In the worst-case scenario, however, the researcher has inspected each of the $n$ data points and decided whether to drop them or not based on their values. In this case, $n_\beta = n$, and $-\log p(\beta) \gg 1$ because the bias consists in a highly complex procedure in which each value of the data is assessed for its impact on the results, and then retained or dropped accordingly. For the purposes of illustration, we will assume that $\beta$ is as complex as the dataset, in which case

$$\frac{h_u}{h_b} = 1 + \frac{-\log p(\beta)}{H(Y) + H(X) - \frac{\log p(\tau)}{n}} \approx 2 \tag{3.30}$$

with the latter approximation derived from assuming that $n$ is large. In this case, therefore the QRP would be tolerable only if $K > 2B$, i.e. the result obtained without the QRP is twice as large as that produced with the QRP. However, if the $K$ was very large to begin with, then the researcher would have little improper reasons to drop data points, unless she was biased against producing a result (in which case $K = B$ and therefore $K_{corr} < 0$). Therefore, under the most likely conditions in which it occurs, selecting data points indiscriminately would be an extremely damaging practice, leading to $K_{corr} < 0$.

The two examples above illustrate how the generic and very ambiguous concept of QRP can be defined more precisely. A similar logic could be applied to all kinds of QRP, to assess their context-specific impact, to distinguish the instances that are innocuous or even positive from the ones of concern, and to rank the latter according to the actual damage they might do to knowledge in different research fields. This logic may also aid in assessing the egregiousness of investigated cases of scientific misconduct.

### 3.7. What characterizes a pseudoscience?

*Problem:* Philosophers have proposed a vast and articulated panorama of criteria to demarcate genuine scientific activity from metaphysics or pseudoscience (table 2).

However, none of these criteria are accepted as universally valid, and prominent contemporary philosophers of science tend to endorse a 'multi-criteria' approach, in which the sciences share a 'family resemblance' to each other but no single universal trait is common to all of them (e.g. [51–53]).

The multi-criterial solution to the demarcation problem is appealing but has limited theoretical and practical utility. In particular, it shifts the question from identifying a single property common to all the sciences to identifying many properties common to some. Proposed lists of criteria typically include normative principles or behavioural standards such as 'rigorously assessing evidence', 'openness to criticism', etc. These standards are unobjectionable but are hard to assess rigorously. Furthermore, since the minimum number of characteristics that a legitimate science should possess is somewhat arbitrary, virtually any practice may be considered a 'science' according to one scheme or another (e.g. intelligent design [60]).

**Table 2.** Previously proposed demarcation criteria.

| principle | science | non-/pseudoscience | author year [ref] |
|---|---|---|---|
| positivism | reached the positive stage: builds knowledge on empirical data | still in theological or meta-physical stages: phenomena are explained by recurring to deities or non-observables entities | Comte 1830 [2] |
| methodologism | follows rigorous methods for selecting hypotheses, acquiring data and drawing conclusions | fails to follow the scientific method | e.g. Pearson 1900 [54], Poincaré 1914 [55] |
| verificationism | builds upon verified statements | relies on non-verifiable statements | Wittgenstein 1922 [56] |
| falsificationism | builds upon falsifiable, non-falsified statements | produces explanations devoid of verifiable counterfactuals | Popper 1959 [41] |
| methodological falsificationism | generates theories of increasing empirical content, which are accepted when surprising predictions are confirmed | protects its theories with a growing belt of auxiliary hypotheses, giving rise to 'degenerate' research programmes | Lakatos 1970 [57] |
| norms | follows four fundamental norms, namely: universalism, communism, disinterestedness, organized scepticism | operates on different, if not the opposite, sets of norms | Merton 1942 [58] |
| paradigm | is post-paradigmatic, meaning it solves puzzles defined and delimited by the rules of an accepted paradigm | is pre-paradigmatic: lacks a unique and unifying intellectual framework or is fragmented into multiple competing paradigms | Kuhn 1974 [59] |
| multi-criterial approaches | bears a sufficient 'family resemblance' to other activities we call 'science' | shares too few characteristics with activities that we consider scientific | e.g. Laudan 1983 [51], Dupre 1993 [52], Pigliucci 2013 [53] |

Is there a single distinctive characteristic of pseudosciences and, if so, how can we measure it?
*Answer:* A pseudoscientific field is characterized by $K_{\text{corr}} < 0$, because

$$K < \frac{h_u}{h_b} B, \tag{3.31}$$

where the terms $K$, $B$, $h_u$, $h_b$ are the cumulative equivalent of the terms in equation (3.26).

*Explanation:* Activities such as palmistry, astrology, homeopathy or psychoanalysis are characterized by having a defined methodology, which contains its own laws, rules and procedures, let us call it $\psi$. This $\psi$ is what makes these practices appear scientific, and it is believed by its practitioners to produce a $K(Y; X, \psi) > 0$. However, such activities are deemed epistemically worthless (and have been so, in many cases, for centuries before the concept of science was formalized), because they typically manifest three conditions: (1) they (appear to) produce large amounts of explanatory knowledge but typically little predictive or causal knowledge; (2) any predictive success or causal power that their practitioners attribute to the explanans is more economically explained by well-understood and unrelated

phenomena and methodologies; and/or (3) their theories and methodologies are independent from, and often incompatible with, those of well-established and successful theories and methodologies ([53]).

All three properties are contained and quantified in equation (3.26).

— Condition 1 implies that a field's observed, as opposed to predicted, $K$ is zero, leading to the condition $K_{adj} < 0$ (§2.3.2) and therefore also to $K_{corr} < 0$ (§3.6).
— Condition 2 entails that, to any extent that a pseudoscientific methodology (appears to) successfully explain, influence or predict an outcome, the same effect can be obtained with a $\tau$ that lacks the specific component $\psi$. Conscious and unconscious biases in study design (e.g. failure to account for the placebo effect) and post hoc biases (e.g. second-guessing one's interpretation) fall into this category of explainable effects. We could also interpret $K$ as being the effect produced by standard methods $\tau$, and $B$ as the (identical) effect produced by the pseudoscience, which, however, has a methodology that is more complex than necessary (the sum $-(\log p(\tau) + \log p(\psi))$), leading to $h_u/h_b > 1$ in equation (3.31).
— Condition 3 can be quantitatively understood as a cost of combining incompatible theories. Let $\upsilon$ be a third theory, which represents the combination of the pseudoscientific theory $\psi$ with other standard theories $\tau$. When the two theories are simply used opportunistically and not unified in a single, coherent theory, then $\log p(\upsilon) = \log p(\tau) + \log p(\psi)$. When the two theories are entirely compatible with each other, indeed one is partially or entirely accounted for by the other, then $-\log p(\upsilon) \ll -\log p(\tau) - \log p(\psi)$. Conversely, to the extent that the two theories are not directly compatible, such that additional theory needs to be added and formulated to attain a coherent and unified account $-\log p(\upsilon) \gg -\log p(\tau) - \log p(\psi)$, leading to $h_u/h_b \gg 1$ in equation (3.31). Formal methods to quantify theoretical discrepancies may be developed in future work.

### 3.7.1. Example: How pseudoscientific is Astrology?

Many studies have been conducted to test the predictions of Astrology, but their results were typically rejected by practising astrologers on various methodological grounds. A notable exception is represented by [61], a study that was designed and conducted with the collaboration and approval of the National Council for Geocosmic Research, a highly prominent organization of astrologers.

In the part of the experiment that was deemed most informative, practising astrologers were asked to match an astrological natal chart with one of three personality profiles produced using the California Personality Inventory. If the natal chart contains no useful information about an individual's personality, the success rate is expected to be 33%, giving $H(Y) = 1.58$. The astrologers predicted that their success rate would be at least 50%, suggesting $H(Y|X, \psi) = 1.58/2 = 0.79$. The astrologer's explanans includes the production of a natal chart, which requires the input of the subject's birth time (hh:mm), date (dd/mm/yyyy) and location (latitude and longitude, four digits each) for a total information of approximately 50 bits. The theory $\psi$ includes the algorithm to compute the star and planet's position, and the relation between these and the personality of the individual. The size of $\psi$ could be estimated, but we will leave this task to future analyses. This omission may have a significant or a negligible impact on the calculations, in proportion to how large the $n_Y$ is, i.e. in proportion to how unchanging the methods of astrology are. The alternative, scientific hypothesis according to which there is no effect to be observed, has $h_u = 1$.

Results of the experiment showed that the astrologers did not guess an individual's personality above chance [61]. Therefore, $K = 0$ and equation (3.31) is satisfied. The $K$ value of astrology from this study is estimated to be

$$K(Y; X, \psi) = -\frac{h_u}{h_b} B = -\frac{1.58 + 50 - \log(\psi)\frac{n_\psi}{n_Y}}{1.58}\frac{1.58 - 0.79}{1.58} < -16.32 \tag{3.32}$$

in which the inequality is due to the unspecified size of $\psi$ and $n_Y$. This analysis is still likely to over-estimate the $K$ of Astrology, because the experiment offered a conservative choice between only three alternatives, whereas astrology's claimed explanandum is likely to be much larger, as it includes multiple personality dimensions (§3.3.3).

### 3.8. What makes a science 'soft'?

*Problem:* There is extensive evidence that many aspects of scientific practices and literatures vary gradually and almost linearly if disciplines are arranged according to the complexity of their subject

**Table 3.** Properties previously proposed to vary across the sciences.

| principle | property or properties | author year [ref] |
|---|---|---|
| hierarchy of the sciences | simplicity, generality, quantifiability, recency, human relevance | Comte 1830 [2] |
| consilience | ability to subsume disparate phenomena under general principles | Whewell 1840 [67] |
| lawfulness | nomoteticity, i.e. interest in finding general laws, as opposed to idioteticity, i.e. interest in characterizing individuality | Windelband 1894 [68] |
| data hardness | data that resist the solvent influence of critical reflection | Russell 1914 [69] |
| empiricism | ability to calculate in advance the results of an experiment | Conant 1951 [70] |
| rigour | rigour in relating data to theory | Storer 1967 [71] |
| maturity | ability to produce and test mechanistic hypotheses, as opposed to mere fact collection | Bunge 1967 [72] |
| cumulativity | cumulation of knowledge in tightly integrated structures | Price 1970 [73] |
| codification | consolidation of empirical knowledge into succinct and interdependent theoretical formulations | Zuckerman & Merton 1973 [66] |
| consensus | level of consensus on the significance of new knowledge and the continuing relevance of old | Zuckerman & Merton 1973 [66] |
| core cumulativity | rapidly growing core of unquestioned general knowledge | Cole 1983 [74] |
| invariance | contextual invariance of phenomena | Humphreys 1990 [65] |

matters (i.e. broadly speaking, mathematics, physical, biological, social sciences and humanities) [46,62–64]. This order reflects what people intuitively would consider an order of increasing scientific 'softness', yet this concept has no precise definition and the adjective 'soft science' is mostly considered denigrative. This may be why the notion of a hierarchy of the sciences is nowadays disregarded in favour of a partial or complete epistemological pluralism (e.g. [52]). How can we define and measure scientific softness?

*Answer:* Given two fields studying systems $Y_A$, $X_A$, $\tau_A$ and $Y_B$, $X_B$, $\tau_B$, field $A$ is harder than $B$ if

$$\frac{k_A}{k_B} > \frac{h_B}{h_A} \tag{3.33}$$

in which $k_A$, $k_B$ and $h_A$, $h_B$ are representatively valid estimates of the fields' bias-adjusted cumulative effects and hardness component, given by properties of their systems as well as the field's average level of accuracy, reproducibility and bias.

*Explanation:* equation (3.33) is a re-arrangement of the condition $K(Y_A; X_A, \tau_A) > K(Y_B; X_B, \tau_B)$, i.e. the condition that field $A$ is more negentropically efficient than field $B$. As argued below, this condition reflects the intuitive concept of scientific hardness.

The various criteria proposed to distinguish stereotypically 'hard' sciences like physics from stereotypically 'soft' ones like sociology cluster along two relevant dimensions:

— Complexity: moving across research fields from the physical to the social sciences, subject matters go from being simple and general to being complex and particular. This increase in complexity corresponds, intuitively, to an increase in the systems' number of relevant variables and the intricacy of their interactions [65].
— Consensus: moving across research fields from the physical to the social sciences, there is a decline in the ability of scientists to reach agreement on the relevance of findings, on the correct methodologies to use, even on the relevant research questions to ask, and therefore ultimately on the validity of any particular theory [66].

(see table 3, and [64] for further references).

Both concepts have a straightforward mathematical interpretation, which points to the same underlying characteristic: having a relatively complex explanans and therefore a low $K$. A system with many interacting variables is a system for which $H(X)$ and/or $H(Y|X, \tau)$ are large. Consequently, progress is slow (§3.3). A system in which consensus is low is one in which the cumulative

methodology $\bar{\tau} + \bar{d}_\tau$ expands rapidly as the literature grows. Moreover, higher complexity and particularity of subject matter entails that a given knowledge is applicable to a limited number of phenomena, entailing smaller $n_Y$. Therefore, all the typical traits associated with a 'soft' science lead to predict a lower value of $K$.

### 3.8.1. Example: mapping a hierarchy of the sciences

The idea that the sciences can be ordered by a hierarchy, which reflects the growing complexity of subject matter and, in reverse order, the speed of scientific progress, can be traced back at least to the ideas of Auguste Comte (1798–1857). The $K$ values estimated in previous sections for various disciplines approximately reflect the order expected based on equation (3.33), particularly if the rescaled $K$ values are compared instead, i.e.

$$H(Y_A)k_A h_A > H(Y_B)k_B h_B. \tag{3.34}$$

Mathematics is a partial exception, in that its $K$ value is likely to be in most cases higher than that of any empirical field, but its rescaled $K$ is not (at least, not if we quantify the explanandum as a binary question). Intriguingly, mathematics were considered an exception also in August Comte's scheme, due to their non-empirical nature. Therefore, the $K$ account of the hierarchy of the sciences mirrors Comte's original hierarchy rather accurately.

However, the hierarchy depicted by results in this essay is merely suggestive, because the examples we used are preliminary. In addition to making frequent simplifying assumptions, the estimates of $K$ derived in this essay were usually based on individual cases (not on cumulative evidence coming from a body of literature) and have overlooked characteristics of a field that may be relevant to determine the hierarchy (for example, the average reproducibility of a literature). Moreover, there may be yet unresolved problems of scaling that impede a direct comparison between widely different systems. Therefore, at present, equation (3.34) can at best be used to rank fields that are relatively similar to each other, whereas methods to compare widely different systems may require further methodological developments.

If produced, a $K$-based hierarchy of the sciences would considerably extend Comte's vision in at least two respects. Firstly, it would rank not quite 'the sciences' but rather scientific 'fields', i.e. literatures and/or research communities identified by a common explanandum and/or explanans. Although the average $K$ values of research fields in the physical, biological and social sciences are predicted to reflect Comte's hierarchy, the variance within each science is likely to be great. It is entirely possible that some fields within the physical sciences may turn out to have lower $K$ values (and therefore to be 'softer') than some fields in the biological and social sciences and vice versa. Secondly, as illustrated in §3.7, a $K$-based hierarchy would encompass not just sciences but also pseudosciences. Whereas the former extend in the positive range of $K$ values, the latter extend in the negative direction. The more negative the value, the more pseudoscientific the field.

## 4. Discussion

This article proposed that $K$, a quantity derived from a simple function, is a general quantifier of knowledge that could find useful applications in meta-research and beyond. It was shown that, in addition to providing a universal measure of effect size, $K$ theory yields concise and memorable equations that answer meta-scientific questions and may help understand and forecast phenomena of great interest, including reproducibility, bias and misconduct, and scientific progress (table 1). This section will first discuss how $K$ theory may solve limitations of current meta-science (§4.1 and 4.2), then address the most likely sources of criticisms (§4.3), and finally it will suggest how the theory can be tested (§4.4).

### 4.1. Limitations of current meta-science

The growing success and importance of meta-research have made the need for a meta-theory ever more salient and pressing. Growing resources are invested, for example, in ensuring reproducibility [1], but there is little agreement on how reproducibility ought to be predicted, measured and understood in different fields [31,75]. Graduate students are trained in courses to avoid scientific misconduct and questionable research practices, and yet the definition, prevalence and impact of questionable

behaviours across science are far from well established [50]. Increasing efforts are devoted to measuring and countering well-documented problems such as publication bias, even though inconclusive empirical evidence [42] and past failures of similar initiatives (e.g. the withering and closure of most journals of negative results [76]) suggest that the causes of these problems are incompletely understood.

At present, meta-scientific questions are addressed using theoretical models derived from very specific fields. As a consequence, their results are not easily extrapolated to other contexts. The most prominent example is offered by the famous claim that most published research findings are false [77]. This landmark analysis has deservedly inspired meta-studies in all disciplines. However, its predictions are based on an extrapolation of statistical techniques used in genetic epidemiology that have several limiting assumptions. These assumptions include that all findings are generated by stable underlying phenomena, independently of one another, with no information on their individual plausibility or posterior odds, and with low prior odds of any one effect being true. These assumptions are unlikely to be fully met even within genetic studies [78], and the extent to which they apply to any given research field remains to be determined.

Similar limiting assumptions are increasingly noted in the application of meta-research methodologies. Reproducibility and bias, for example, are measured using meta-analytical techniques that treat sources of variation between studies as either fixed or random [13,79]. This assumption may be valid when aggregating results of randomized control trials [80], but may be inadequate when comparing results of fields that use varying and evolving methods (e.g. ecology [81]) and that study complex systems that are subject to non-random variation (expressed, for example, in reaction norms [82]).

Statistical models can be used to explore the effects of different theoretical assumptions (e.g. [83–86]) as well as other conditions that are believed to conduce to bias and irreproducibility (e.g. [87,88]). However, the plural of 'model' is not 'theory'. A genuine 'theory of meta-science' ought to offer a general framework that, from maximally simple and universal assumptions, explains how and why scientific knowledge is shaped by local conditions.

## 4.2. *K* theory as a meta-theory of science

Why does *K* theory offer the needed framework? First and foremost, this theory provides a quantitative language to discuss meta-scientific concepts in terms that are general and abstract and yet specific enough to avoid confusing over-simplifications. For example, the concept of bias is often operationalized in meta-research as an excess of statistically significant findings [77] or as an exaggeration of findings due to QRP [89]. Depending on the meta-research question, however, these definitions may be too narrow, because they exclude biases against positive findings and only apply to studies that use null-hypothesis significance testing, or they may be too generic, because they aggregate research practices that differ in relevant ways from each other. Similar difficulties in how reproducibility, negative results and other concepts are used have emerged in the literature as discussed in the Results section. As illustrated by the examples offered throughout this essay, *K* theory avoids these limitations by proposing concepts and measures that are extremely abstract and yet adaptable to reflect field-specific contexts.

Beyond the conceptual level, *K* theory contextualizes meta-research results at an appropriate level of generalization. Current meta-research models and empirical studies face a conundrum: they usually aim to draw general conclusions about phenomena that may occur anywhere in science, but these phenomena find contextual expression in fields that vary widely in characteristics of subject matter, theory, methodology and other aspects. As a result, meta-research studies are forced to choose between under-generalizing their conclusions by restricting them to a specific field or literature and over-generalizing them to an entire field or discipline, or even to the whole of science. One of the unfortunate consequences of this over-generalization of results has been the diffusion of a narrative that 'science is in crisis', narrative that has no empirical or pragmatic justification [75]. Excessive under- and over-generalizations may be avoided by systematizing meta-research results with *K* theory, which offers a mid-level understanding of meta-scientific phenomena that is independent of subject matter and yet measurable in context.

An example of the mid-level generalizations permitted by *K* theory is the hierarchy of sciences and pseudosciences proposed in §3.8. A classification based on this approach, for example, could lead us to abandon traditional disciplinary categories (e.g. 'physics' or 'social psychology') in favour of epistemologically relevant categories such as 'high-h' fields, or 'low-λ' systems.

Other classifications and theories about science may be derived from *K* theory. An alternative to the rather ill-defined 'hard−soft' dimension, for example, could be a continuum between two strategies. At

one end of the spectrum, is what we might call a '$\tau$-strategy', which invests more resources in identifying and encoding regularities and laws that allow general explanations and long-term predictions, at the cost of contingent details. At the other end, is an '$X$-strategy', which invests greater resources in acquiring large amounts of contingent, descriptive information that enables accurate but proximate explanations and predictions. Depending on characteristics of the explananda and the amount of resources available for the storage and processing of information, each scientific field expresses an optimal balance between $\tau$-strategy and $X$-strategy.

## 4.3. Foreseeable criticisms and limitations

At least five criticisms of this essay may be expected. The first is a philosophical concern with the notion of knowledge, which in this article is defined as information compression by pattern encoding. Critics might argue that this definition does not correspond to the epistemological notion of knowledge as 'true, justified belief' [90]. Even Fred Dretske, whose work extensively explored the connection between knowledge and information [10], maintained that 'false information' was not genuine information and that knowledge required the latter [91]. The notion of knowledge proposed in this text, however, is only apparently unorthodox. In the $K$ formalism, a true justified belief corresponds to a system for which $K > 0$. It can be shown that a 'false, unjustified' belief is one in which $K \leq 0$. Therefore, far from contradicting information-theoretic epistemologies, $K$ theory may give quantitative answers to open epistemological questions such as 'how much information is enough'? [91].

The second criticism may be that the ideas proposed in this essay are too simple and general not to have been proposed before. The claim made by this essay, however, is not that every concept in it is new. Rather to the contrary, the claim is that $K$ theory unifies and synthesizes innumerable previous approaches to combining cognition, philosophy and information theory, and it does so in a formulation that, to the best of the author's knowledge, is entirely new and original. Earlier ideas that have inspired the $K$ function are found, for example, in Brillouin's book *Science and information theory*, which discussed the information value of experiments and calculated the information content of a physical law [5]. Brillouin's analysis, however, did not include factors that are key to the $K$ function, including the standardization on logarithm space, the decline rate of knowledge, the number $n_Y$ of potential applications of knowledge and the inclusion of the information costs of the theory $\tau$. The description length of theories (or, at least, of statistical models) is a key component of the minimum description length principle, which was first proposed by Rissanen [7] and is finding growing applications in problems of statistical inference and computation (e.g. [6,8]). The methods developed by MDL proponents and by algorithmic information theory are entirely compatible with the $K$ function (and could be used to quantify $\tau$) but differ from it in important theoretical and mathematical aspects (§2.2.2). Within philosophy, Paul Thagard's *Computational philosophy of science* [11] offers numerous insights into the nature of scientific theories and methodologies. Thagard's ideas may be relevant to $K$ theory because, among other things, they illustrate what the $\tau$ of a scientific theory might actually contain. However, Thagard's theory differs from $K$ theory in substantive conceptual and mathematical aspects, and it does not offer a general quantifier of knowledge nor does it produce a meta-scientific methodology. Finally, $K$ theory was developed independently from other recent attempts to give informational accounts of cognitive phenomena, for example, the free-energy principle (e.g. [92]) and the integrated information theory of consciousness (e.g. [93]). Whereas these theories bear little resemblance to that proposed in this essay, they obviously share a common objective with it, and possible connections may be explored in future research.

The third criticism might be methodological, because entropy is a difficult quantity to measure. Estimates of entropy based on empirical frequencies can be biased when sample sizes are small, and they can be computationally demanding when data is large and multi-dimensional. Neither of these limitations, however, is critical. With regard to the former problem, as demonstrated in §2.3.6, powerful computational methods to estimate entropy with limited sample size are already available [18]. With regard to the latter problem, we may note that the 'multi-dimensional' $K_{md}$ used in §3.3 is the most complex measure proposed and yet it is not computationally demanding, because it is derived from computing unidimensional entropies. The 'cumulative' $K_{cum}$ may also be computationally demanding, as it requires estimating the entropy of mixed distributions. However, analytical approaches to estimate the entropy of mixed distributions and other complex data structures are already available and are likely to be developed further (e.g. [94,95]).

The fourth criticism may regard the empirical validity of the measures proposed. As it was emphasized throughout the text, all the practical examples offered were merely illustrative and

preliminary, because they generally relied on incomplete data and simplifying assumptions. In particular, it appears to be difficult to quantify exactly the information content of $\tau$, especially for what concerns the description of a methodology. This limitation, however, is often avoidable. In most contexts of interests, it will suffice to estimate $\tau$ with some approximation and/or in relative terms. It may be a common objective within studies using $K$ theory, for example, to estimate the divergence between two methodologies. Even if complete information about a methodology in unavailable (if anything, because it is likely to include 'tacit' components that are by definition hidden) relative differences documented in the methods' description are simple to identify and therefore to quantify by $K$ methods. These relative quantifications could become remarkably accurate and extend across research fields, if they were based on a reliable taxonomy of methods that provided a fixed 'alphabet' $\mathcal{T}$ of methodological choices characterizing scientific studies. Taxonomies for research methods are already being developed in many fields to improve reporting standards (e.g. [96]) and could be extended by meta-scientists for meta-research purposes.

The fifth criticism that may be moved to $K$ theory is that it is naively reductionist, because it appears to overlook the preponderant role that historical, economic, sociological and psychological conditions play in shaping scientific practices. Quite to the contrary, $K$ theory is not proposed as an alternative to historical and social analyses of science, but as a useful complement to them, which is necessary to fully understand the history and sociology of a research field. A parallel may be drawn with evolutionary biology: to explain why a particular species evolved a certain phenotype or to forecast its risk of extinction, we need to combine contingent facts about the species' natural history with general theories about fitness dynamics; similarly, to better understand and forecast the trajectory taken by a field we need to combine contingent and historical information with general principles about knowledge dynamics.

## 4.4. Testable predictions and conclusion

We can summarize the overall prediction of $K$ theory in a generalized rule: *An activity will exhibit the epistemological, historical, sociological and psychological properties associated with a science if and to the extent that:*

$$K > \frac{h}{h_b}B, \tag{4.1}$$

in which $K$ is the knowledge, corrected for biases, and $h_b$ and $B$ are the costs and impacts of biases internal or external to the system. If biases are absent or not easily separable from the system, and indicating with $K$ the overall knowledge yield of the activity, the rule simplifies to

$$K > 0. \tag{4.2}$$

This overall prediction finds specific expression in the relations reported in table 1, each of which leads to predict observable phenomena in the history and sociology of science. These predictions include:

— Scientific theories and fields fail or thrive in proportion to the their rate of consilience, measured at all levels—from the micro ($K_{cum}$) to the macro ($K_{md}$, and see inequality (3.11)). For example, we predict that discredited theories, such as that of phlogiston or phrenology, were characterized by a $K$ that was steadily declining and were abandoned when $K \leq 0$. Conversely, fields and theories that grow in size and importance are predicted to exhibit a positive growth rate of $K$. When the rate of growth of $K$ slows down and/or when it reaches a plateau, $K$ is 're-set' to zero by the splitting in sub-fields and/or the expansion to new explananda or explanantia.

— The expected reproducibility of published results is less than 100% for most if not all fields, and is inversely related to the average informational divergence, of explanandum and/or explanans, between the original study and its replications. In some instances, the divergence of methods might reflect the differential presence of bias. However, the prediction is independent of the presence of bias.

— The value of null and contradictory findings is smaller or equal to that of 'positive' findings, and is directly related to the level of a field's theoretical and methodological codification ($|\mathcal{T}|$) and explanatory power ($k$). This value may be reflected, for example, in the rate of citations to null results, their rate of publication and the space such results are given in articles with multiple results.

— In functional sciences, the prevalence of questionable, problematic and openly egregious research practices is inversely related to their knowledge cost. Therefore, their prevalence will vary

depending on details of the practice (e.g. how it is defined) as well as the level of codification and explanatory power of the field.

— The relative prestige and influence of a field is directly related to its $K$ (scaled and/or not scaled). All else being equal, activities that can account for greater explananda with smaller explanantia are granted a higher status, reflected in symbolic and/or material investments (e.g. societal recognition and/or public research funds).

— The relative popularity and influence of a pseudoscience is inversely related to its $K$. An activity that (pretends) to yield knowledge will acquire relatively more prestige to the extent that it promises to explain a wider range of phenomena using methods that appear to be highly codified and very complex.

The testability of these predictions is limited by the need to keep 'all else equal'. As discussed above, there is no denying that contingent and idiosyncratic factors shape the observable phenomena of science to a significant, possibly preponderant extent. Indeed, if empirical studies using $K$ theory cumulate, we may eventually be able to apply $K$ theory to itself, and it may turn out that the empirical $K$ value of $K$ theory is relatively small and that, to any extent that external confounding effects are not accounted for, the $|\mathcal{T}|$ of $K$ theory is large, leading to low falsifiability.

The testability of $K$ theory, however, extends beyond the cases examined in this essay. On the one hand, within meta-science, more contextualized analyses about a field or a theory will lead to more specific and localized predictions. These localized predictions will be more accurately testable, because most irrelevant factors will be controlled for more easily. On the other hand, and most importantly, the theory can in principle apply to phenomena outside the contexts of science.

The focus of this article has been quantitative scientific research, mainly because this is the subject matter that inspired the theory and that represents the manifestation of knowledge that is easier to conceptualize and quantify. However, the theory and methods proposed in this essay could be adapted to measure qualitative research and other forms of knowledge. Indeed, with further development, the $K$ function could be used to quantify any expression of cognition and learning, including humour, art, biological evolution or artificial intelligence (see appendix A), generating new explanations and predictions that may be explored in future analyses.

Ethics. This research does not involve the use of animal or human subject, nor the handling of sensitive information. No ethical approval and no permission to carry out fieldwork was required.

Data accessibility. The R code and datasets used to generate all analyses and figures are included as electronic supplementary material. Any other empirical dataset used in the analyses was obtained from publications and repositories that are publicly accessible and indicated in the text.

Competing interests. I declare I have no competing interests.

Funding. I received no funding for this study.

Acknowledgements. Marco del Giudice gave helpful comments about the analysis of gender differences in personality.

# Appendix A

## A.1. Postulates underlying $K$ theory

### A.1.1. Postulate 1: information is finite

The first postulate appears to reflect a simple but easily overlooked fact of nature. The universe—at least, the portion of it that we can see and have causal connection to—contains finite amounts of matter and energy, and therefore cannot contain infinite amounts of information. If each quantum state represents a bit, and each transition between (orthogonal) states represents an operation, then the universe has performed circa $10^{120}$ operations on $10^{90}$ bits since the Big Bang [97].

Advances in quantum information theory suggest that our universe may have access to unlimited amounts of information, or at least of information processing capabilities [98] (but see [99] for a critique). However, even if this were the case, there would still be little doubt that information is finite as it pertains to knowledge attainable by organisms. Sensory organs, brains, genomes and all other pattern-encoding structures that underlie learning are finite. The sense of vision is constructed from a limited number of cone and rod cells; the sense of hearing uses information from a limited number of hair cells, each of which responds to a narrow band of acoustic frequencies; brains contain a limited number of connections; genomes a countable number of bases, etc. The finitude of all biological structures is one of the considerations that has led cognitive scientists and biologists to assume

information is finite when attempting, for example, to model the evolution of animal cognitive abilities [100]. Even mathematicians have been looking with suspicion to the notion of infinity for a long time [101]. For example, it has been repeatedly and independently shown that, if rational numbers were actually infinite, then infinite information could be stored in them and this would lead to insurmountable contradictions [102].

Independent of physical, biological and mathematical considerations, the postulate that information is finite is justifiable on instrumentalist grounds, because it is the most realistic assumption to make when analysing scientific knowledge. Quantitative empirical knowledge is based on measurements, which are technically defined as partitionings of attributes in sets of mutually exclusive categories [103]. In principle, this partitioning could recur an infinite number of times, but in practice it never does. Measurement scales used by researchers to quantify empirical phenomena might be idealized as extending to infinity, but in practice they always consist in a range of plausible values that is delimited at one or both ends. Values beyond these ends can be imagined as constituting a single set of extreme values that may occur with very small but finite probability.

Therefore, following either theoretical or instrumentalist arguments, we are compelled to postulate that information, i.e. the source of knowledge, is a finite quantity. Its fundamental unit of measurement is discrete and is called the bit, i.e. the 'difference that makes a difference', according to Gregory Bateson's famous definition [104]. For this difference to make any difference, it must be perceivable. Hence, information presupposes the capacity to dichotomize signals into 'same' and 'not same'. This dichotomization can occur recursively and we can picture the process by which information is generated as a progressive subdivision (quantization) of a unidimensional attribute. This quantization operates 'from the inside out', so to speak, and by necessity always entails two 'open ends' of finite probability.

### A.1.2. Postulate 2: knowledge is information compression

The second postulate claims that the essence of any manifestation of what we call 'knowledge' consists in the encoding of a pattern, which reduces the amount of information required to navigate the world successfully. By 'pattern' we intend here simply a dependency between attributes—in other words, a relationship that makes one event more or less likely, from the point of view of an organism, depending on another event. By encoding patterns, an organism reduces the uncertainty it confronts about its environment—in other words, it *adapts*. Therefore, postulate 2, just like postulate 1, is likely to reflect an elementary fact of nature; a fact that arguably underlies not just human knowledge but all manifestations of life.

The idea that knowledge, or at least scientific knowledge, is information compression is far from new. For example, in the late 1800s, physicist and philosopher Ernst Mach argued that the value of physical laws lay in the 'economy of thought' that they permitted [3]. Other prominent scientists and philosophers of the time, such as mathematician Henri Poincaré, expressed similar ideas [55]. Following the development of information theory, scientific knowledge and other cognitive activities have been examined in quantitative terms (e.g. [5,105]). Nonetheless, the equivalence between scientific knowledge and information compression has been presented as a principle of secondary importance by later philosophers (including for example Popper [41]), and today does not appear to occupy the foundational role that it arguably deserves [106].

The reluctance to equate science with information compression might be partially explained by two common misconceptions. The first one is an apparent conflation of lossless compression, which allows data to be reconstructed exactly, with lossy compression, in which instead information from the original source is partially lost. Some proponents of the compression hypothesis adopt exclusively a lossless compression model, and therefore debate whether empirical data are truly compressible in this sense (e.g. [107]). However, science is clearly a lossy form of compression: the laws and relations that scientists discover typically include error terms and tolerate large portions of unexplained variance.

The second, and most important, source of scepticism seems to lie in an insufficient appreciation for the fundamental role that information compression plays not only in science, but also knowledge and all other manifestations of biological adaptation. Even scientists who equate information compression with learning appear to under-estimate the fundamental role that pattern-encoding and information compression play in all manifestations of life. In their seminal introductory text to Kolmogorov complexity [6], for example, Li and Vitányi unhesitatingly claim that 'science may be regarded as the

art of data compression' [6, p. 713], that 'learning, in general, appears to involve compression of observed data or the results of experiments', and that 'in everyday life, we continuously compress information that is presented to us by the environment', but then appear cautious and conservative in extending this principle to non-human species, by merely suggesting that 'perhaps animals do this as well', and citing results of studies on tactile information transmission in ants [6, p. 711]. It seems that even the most prominent experts and proponents of information compression methodologies can be disinclined to apply their favoured principle beyond the realm of human cognition and animal behaviour.

This essay takes instead the view that information compression by pattern encoding is the quintessence of biological adaptation, in all of its manifestations. Changes in a population's genetic frequencies in response to environmental pressures can be seen as a form of adaptive learning, in which natural selection reinforces a certain phenotypic response to a certain environment and weakens other responses, thereby allowing a population's genetic codes to 'remember' fruitful responses and 'forget' erroneous (i.e. non-adaptive) ones. For these reinforcement processes to occur at all, environmental conditions must be heterogeneous and yet partially predictable. Natural selection, in other words, allows regularities in the environment to be genetically encoded. This process gives rise to biodiversity that may mirror environmental heterogeneity at multiple levels (populations, varieties, species, etc.). Such environmental heterogeneity is not exclusively spatial (geographical). Temporal heterogeneity in environmental conditions gives rise to various forms of phenotypic plasticity, in which identical genomes express different phenotypes depending on cues and signals received from the environment [108]. Whether genetic or phenotypic, adaptation will be measurable as a correlation between possible environmental conditions and alternative genotypes or phenotypes. This correlation is in itself a measurable pattern.

As environments are increasingly shaped by biological processes, they become more complex and heterogeneous, and they therefore select for ever more efficient adaptive capabilities—ever more rapid and accurate ways to detect and process environmental cues and signals. Immune systems, for example, allow large multicellular plants and animals to protect themselves from infective agents and other biological threats whose rate of change far out-competes their own speed of genetic adaptation; endocrine systems allow the various parts of an organism to communicate or coordinate their internal activities in order to respond more rapidly to changes in external conditions. Similar selective pressures have favoured organisms with nervous systems of increasing size and complexity. Animal behaviour and cognition, in other words, are simply higher-order manifestations of phenotypic plasticity, which allow an organism to respond to environmental challenges on shorter temporal scales. Behavioural responses may be hard-wired in a genome or acquired during an organism's lifetime, but in either case they entail 'learning' in the more conventional sense of encoding, processing and storing memories of patterns and regularities abstracted from environmental cues and signals.

Human cognition, therefore, may be best understood as just another manifestation of biological adaptation by pattern encoding. At the core of human cognition, as with all other forms of biological adaptation, lies the ability to anticipate events and thus minimize error. When we say that we 'know' something, we are claiming that we have fewer uncertainties about it because, given an input, we can predict above chance what will come next. We 'know a city', for example, in proportion to how well we are able to find our way around it, by going purposely from one street to the next and/or navigating it by means of a simplified representation of it (i.e. a mental map). This ability embodies the kind of information we may communicate to a stranger when asked for directions: if we 'know the place', we can provide them with a series of 'if-then' statements about what direction to take once identifiable points are reached. In another example, we 'know a song' in proportion to how accurately we can reproduce its specific sequence of words and intonations with no error or hesitation, or in proportion to how readily we can recognize it when we hear a few notes from it. Similarly, we 'know a person' in proportion to how many patterns about them we have encoded: at first, we might only be able to recognize their facial features; after making superficial acquaintance with them, we will be able to connect these features to their name; when we know them better, we can tell how they will respond to simple questions such as 'where are you from?'; eventually we might 'know them well' enough to predict their behaviour rather accurately and foretell, for example, the conditions that will make them feel happy, interested, angry, etc.

The examples above aim to illustrate how the concept of 'prediction' underlies all forms of knowledge, not just scientific knowledge, and applies to both time (e.g. knowing a song) and space (e.g. knowing a city). Memory and recognition, too, can be qualified as forms of prediction and therefore as manifestations of information compression, whereby sequences of sensory impressions are

encoded and recalled (i.e. memorized) or matched to new experiences (i.e. recognized) in response to endogenous or exogenous signals. Language is also a pattern-encoding, information compression tool. A typical sentence, which constitutes the fundamental structure of human language and thought, expresses the connection between one entity, the subject, and another entity or property, via a relation condition encoded in a verb. It is not a coincidence that the most elementary verb of all—one that is fundamental to all human languages—is the verb 'to be'. This verb conveys a direct relation between two entities, and thus represents the simplest pattern that can be encoded: 'same' versus 'not same', as discussed in relation to Postulate 1. Even a seemingly abstract process like logical deduction and inference can be understood as resulting from pattern encoding. According to some analyses, computing itself and all other manifestations of artificial and biological intelligence may result from a simple process of pattern matching [109].

Scientific knowledge, therefore, is most naturally characterized as just one manifestation of human cognition among many and, therefore, as nothing more than a pattern-encoding activity that reduces uncertainty about one phenomenon by relating it to information about other phenomena. The knowledge produced by all fields of scientific research is structured in this way:

— Mathematical theorems uncover logical connections between two seemingly unrelated theoretical constructs, proving that the two are one and the same.
— Research in the physical sciences typically aims at uncovering mathematical laws, which are rather explicitly encoding patterns (i.e. relationships between quantities). Even when purely descriptive, however, physical research actually consists in the encoding of pattern and relations between phenomena—for example, measuring the atomic weight of a known substance might appear to be a purely descriptive activity, but the substance itself is identified by its reactive properties. Therefore, such research is about drawing connections between properties.
— Most biological and biomedical research consists in identifying correlations or causes and/or in describing properties of natural phenomena, all of which are pattern-encoding activities. Research in taxonomy and systematics might appear to be an exception, but it is not: organizing the traits of a multitude of species into a succinct taxonomical tree is the most elementary form of data compression.
— Quantitative social and behavioural sciences operate in a similar manner to the biological sciences. Even qualitative, ethnographic, purely descriptive social and historical research consists in data compression, because it presupposes that there are general facts about human experiences, individuals, or groups that can be communicated, entailing that they can be described, connected to each other and/or summarized in a finite amount of text.
— The humanities aim to improve our understanding of complex and often unique human experiences, and might therefore appear to have fundamentally different objectives from the natural and social sciences. To any extent that they offer knowledge and understanding, however, these come in the form of information compression. Research in History, for example, is guided by the reconstruction and succinct description of events, which is based on logic, inference and drawing connections to other events, and therefore it follows the principles of economy of thought and compression. The study of literary works, to make another example, produces knowledge by drawing connections and similarities between texts, identifying general schemata and/or uncovering new meaning in texts by recurring to similes and metaphors [110]. Similarities, connections, schemata, similes and metaphors arguably constitute the basis of human cognition [110] and are all manifestations of information compression by pattern encoding.

Other non-academic manifestations of human cognition, creativity and communication can be understood as stemming from a process of information compression, too. The sensual and intellectual pleasure that humans gain from music and art, for example, seems to derive from an optimal balance between perception of structure (pattern that generates predictions and expectations) and perception of novelty (which stimulates interest by presenting new and knowable information) [111]. The sense of humour similarly seems to arise from the sudden and unexpected overturning of the predicted pattern, which occurs when an initially plausible explanation of a condition is suddenly replaced by an alternative, unusual and yet equally valid one [112]. The intellectual and artistic value of a work of art lies in its ability to reveal previously unnoticed connections between events or phenomena in the world (thereby revealing a pattern) and/or in its capacity to synthesize and communicate effectively what are otherwise highly individual, complex and ineffable human experiences—thereby lossy-compressing and transmitting the experience.

# Appendix B

## B.1. Relation with continuous distribution

Indicating with $f(x)$ a probability density function and with $h(X) = -\int f(x) \log f(x)\, d(x)$ the corresponding differential entropy, we have

$$H(X^{\Delta}) \approx h(X) + \log \frac{1}{\Delta} = h(X) + n \tag{B 1}$$

in which $\Delta = 2^{-n}$ is the size of the length of the bin in which $f(x)$ is quantized, and $n$ corresponds to the number of bits required to describe the function to $n-bit$ accuracy. Evidently, we can always rescale $X$ in order to have $\Delta = 1$.

Equation (B 1) applies to any probability density function. Here we will consider in particular the case of the normal distribution, the differential entropy of which is simply $h(x) = \log \sqrt{2\pi e}\sigma_y$. Therefore, if $y$ is a continuous RV, quantized to $n$ bits, for a given $x$ and $\tau$ we have

$$
\begin{aligned}
K(y; x, \tau) &= \frac{\log(\sqrt{2\pi e}\sigma_y) + n - \log(\sqrt{2\pi e}\sigma_{y|x,\tau}) - n}{\log(\sqrt{2\pi e}\sigma_y) + n + x + \tau} = \frac{\log\sqrt{2\pi e} + \log \sigma_y - \log \sqrt{2\pi e} - \log \sigma_{y|x,\tau}}{\log \sqrt{2\pi e} + \log \sigma_y^2 + n + x + \tau} \\
&= \frac{\log \sigma_y - \log \sigma_{y|x,\tau}}{\log \sigma_y + x + \tau + \log \sqrt{2\pi e} + n} \to \frac{\log \sigma'_y - \log \sigma'_{y|x,\tau}}{\log \sigma'_y + x + \tau + \log \sqrt{2\pi e}} = \frac{\log \sigma'_y - \log \sigma'_{y|x,\tau}}{\log \sigma'_y + x + \tau + C},
\end{aligned} \tag{B 2}
$$

in which $C = \log \sqrt{2\pi e}$ and $\sigma'$ corresponds to $\sigma$ rescaled to a common lowest significant digit (e.g. from $\sigma = 0.123$ to $\sigma' = 123$).

# Appendix C

*Proof.*

$$
\begin{aligned}
K_{\text{adj}} &\equiv h\left( \frac{H(Y) - \sum p(y, x|\tau) \log \frac{1}{p(y|x, \hat{\tau})}}{H(Y)} \right) \\
&= h\left( \frac{H(Y) - \sum p(y, x|\tau) \log \frac{1}{p(y|x, \tau)} - \sum p(y, x|\tau) \log \frac{p(y|x, \tau)}{p(y|x, \hat{\tau})}}{H(Y)} \right) \\
&= K(Y; X, \tau) - D(Y|X, \tau \| Y|X, \hat{\tau}) \frac{h}{H(Y)} \equiv K_{\text{obs}} - D(Y|X, \tau \| Y|X, \hat{\tau}) \frac{h}{H(Y)}
\end{aligned} \tag{C 1}
$$

$\square$

# Appendix D

Firstly note that, independently of the size of the vectors $\lambda$ and $d$ in equation (2.29), their inner product yields a number. Therefore, for the purposes of our discussion we can assume $\lambda$ and $d$ to be single numbers. Equation (2.29) claims that there exists a $\lambda \in \mathcal{R}$ such that

$$\lambda = \frac{1}{d} \log_A \frac{K(Y; X, \tau)}{K(Y'; X', \tau')} \tag{D 1}$$

in which $d > 0$ expresses the divergence between systems, and $A$ is an arbitrary basis. This statement is self-evidently true, as long as $K(Y'; X', \tau') \neq 0$ and $K(Y; X, \tau) \neq 0$ or, equivalently, if we allow $\lambda$ to be approximately infinite in the case that $K$ goes to zero in one step $d = 1$. However, two rather useful conclusions can be derived about this equation:

(i) Under most conditions, $K$ is a non-increasing function of divergence. That is, $K(Y'; X', \tau') \leq K(Y; X, \tau)$ and therefore $\lambda \geq 0$.

(ii) The larger the divergence, the larger the decline of $K$, such that under typical conditions we have $K(Y_{d+1}; X, \tau) = K(Y_d; X, \tau) A^{-\lambda} = K(Y; X, \tau) A^{-\lambda(d+1)}$ for distances in the explanandum, and similarly for distances in the explanans.

We will review each argument separately.

## D.1. Statement (i)

From equation (D 1), if $\lambda d \geq 0$, and regardless of the base $A$ chosen for the logarithm, we have

$$\log\frac{H(Y') + H(X') - \log p(\tau')}{H(Y) + H(X) - \log p(\tau)} \geq \log\frac{H(Y') - H(Y'|X', \tau')}{H(Y) - H(Y|X, \tau)} \equiv \log\frac{I(Y'; X', \tau')}{I(Y; X, \tau)} \tag{D 2}$$

in which $I(Y; X) = H(Y) - H(Y|X)$ is the mutual information function.

Claiming that the explanandum $Y_d$ is at a divergence $d$ from $Y$ implies that not all information about $Y_d$ may be contained in $Y$. This condition is typically described mathematically as a Markov chain (MC). An MC is said to be formed by random variables (RVs) $X$, $Y$, $Z$ in that order, and is indicated as $X \to Y \to Z$, when the distribution of $Z$ is conditionally independent of $X$. In other words, the best predictor of $Z$ is $Y$, and if $Y$ is known, $X$ adds nothing. In entropy terms, this entails that $H(Z|Y, X) = H(Z|Y)$, and it formalizes our intuition that information transmitted along a noisy channel tends to be lost.

Markov chains are used to model a variety of systems in the physical, biological and social sciences. An isolated physical system, for example, would be represented as an MC, in which the transition probabilities from one state of the system to the next are determined by the laws of physics. In the $K$ formalism, the laws of physics would be encoded in a $\tau$, whereas a Markov chain may consist in the input $X$ and subsequent states of $Y$, i.e. $X \to Y \to Y_d \to Y_{d+1} \ldots$. Other representations are possible. For example, if no input is present, then the MC would consist in $Y \to Y_d \to Y_{d+1} \ldots$ or, if the state of both input and explanandum is allowed to change, then the MC is $(X, Y) \to (X_d, Y_d) \to (X_{d+1}, Y_{d+1}) \ldots$.

Regardless of how it is formalized in $K$, a system describable by an MC is subject to a central result of information theory, the data processing inequality (DPI), which states that the mutual information between explanandum and explanans will be non-increasing. We will repeat here the proof of the DPI assuming a constant $\tau$ and a Markov chain $X \to Y \to Y_d$. We consider the mutual information between input and two states of the explanandum, and note that it can be expressed in two different ways:

$$I(Y, Y_d; X) = I(Y; X) + I(Y_d; X|Y) = I(Y_d; X) + I(Y; X|Y_d) \tag{D 3}$$

since by Markovity, $I(Y_d; X|Y) = 0$, and remembering that the mutual information is always non-negative we re-arrange and conclude that

$$I(Y_d; X) = I(Y; X) - I(Y; X|Y_d) \leq I(Y; X), \tag{D 4}$$

which proves the DPI. Applying this result to inequality (D 2), we obtain

$$\log\frac{H(Y_d) + H(X) - \log p(\tau)}{H(Y) + H(X) - \log p(\tau)} \geq \log\frac{I(Y_d; X, \tau)}{I(Y; X, \tau)} \leq 0. \tag{D 5}$$

Therefore, inequality (D 2) is always satisfied when $H(Y_d) \geq H(Y)$ (which makes the left-hand side of the inequality larger or equal to 0). In other words, $K$ will always be non-increasing, as long as the entropy in the explanandum is stable or increasing. A stable or increasing entropy is the most probable condition of physical phenomena.

Although a less likely occurrence, it may be the case that the entropy of the explanandum actually declines with divergence, in which case inequality (D 1) may or may not be satisfied. To examine this case, let $H(Y_d) < H(Y) = H(Y_d) + d_Y$, with $d_Y > 0$ quantifying the divergence. And, similarly, let $H(Y|X, \tau) = H(Y_d|X, \tau) + d_{Y|X}$. Then inequality (D 1) can be arranged as

$$\frac{H(Y_d) + d_Y + H(X) - \log p(\tau)}{H(Y_d) + H(X) - \log p(\tau)} \leq \frac{H(Y_d) + d_Y - H(Y_d|X, \tau) - d_{Y|X}}{H(Y_d) - H(Y_d|X, \tau)}, \tag{D 6}$$

which with a few re-arrangements leads to the condition

$$K(Y_d; X, \tau) \leq \frac{d_Y - d_{Y|X}}{d_Y}, \tag{D 7}$$

which is not guaranteed to be true, but can in principle always be met. This follows because by definition either $d_{Y|X} > 0$ and $d_Y \geq d_{Y|X}$ (otherwise we would have $I(Y_d; X, \tau) > I(Y; X, \tau)$, contradicting the DPI), or $d_{Y|X} < 0$, and again $d_Y \geq d_{Y|X}$, because $d_Y \geq 0$. Therefore, the right-hand side is always non-negative, so it could in principle be larger than the $K$ value on the left-hand side. However, if $d_Y = d_{Y|X}$, then the

inequality is certainly false because in that case $K(Y_d; X, \tau) > 0$. Therefore, we conclude that $K$ may increase with divergence, when the information in (the uncertainty, complexity of) the explanandum decreases, which, however, is a less likely occurrence.

For the case of a theory/methodology $\tau' = \tau_d$ at a divergence $d$ from another $\tau$, the argument is only slightly different. Crucial, in this case, is the assumption that the divergence $d$ represents a random deviation from $\tau$, i.e. one that is independent of $\tau$ itself and is not determined by the value of $K(Y; X, \tau_d)$. This assumption is equivalent to that of made for a Markov chain, in which the $\tau$ is subjected to a level of noise proportional to $d$. However, the effects on $K$ require a different analysis.

Firstly, note that the two components may have the same description length, $\log p(\tau_d) = \log p(\tau)$, or not. In the former case, $\tau$ and $\tau'$ differ solely in some of the symbols that compose them—in other words, they encode the same number and types of choices, but differ in some of the specific choices made. In the latter case, the distance $d$ quantifies the information that is missing—in other words, the choices encoded in $\tau$ that are not specified in $\tau_d$—and $\log p(\tau) = \log p(\tau_d) + d$.

Starting with the case that $\tau_d$ is not shorter than $\tau$, the consequences of a divergence $d$ can be understood by defining a set $\mathcal{T}_d : \{\tau_1, \tau_2 \ldots \tau_d\}$ of all possible (components of) theories of description length $-\log p(\tau)$, that are at an information distance $d$ from the 'original' theory/methodology $\tau$. To avoid confusion, we will henceforth indicate the latter with $\tau^*$. Now, let $T_d$ be the uniform RV corresponding to this set, and let $\mathcal{K}_d : \{K(Y; X, \tau_i) : \tau_i \in \mathcal{T}_d\}$ be the set of $K$ values corresponding to each instantiation of $T_d$. Clearly, $\mathcal{K}_d$ has one maximum, except for the special case in which $K(Y; X, \tau_i) = K(Y; X, \tau_j) \quad \forall \tau_i, \tau_j \in \mathcal{T}_d$, and all $K$ have exactly the same value irrespective of the theory. If the latter were the case, then $\tau_i$ would be a redundant element of the theory/methodology, in other words an unnecessary specification. However, such redundancies should not be a common occurrence, if $\tau$ is fixed to maximize $K$.

Therefore, excluding the improbable case in which $\tau_i$ is redundant, then $\mathcal{K}_d$ has a maximum. If $\tau^*$ is the theory corresponding to the maximum value $K(Y; X, \tau^*)$ in $\mathcal{K}_d$, then for all the remaining $\tau_i \neq \tau^*$, $0 \leq K(Y; X, \tau_i) < K(Y; X, \tau^*)$ and therefore $K(Y; X, \tau_d) < K(Y; X, \tau^*)$ or equivalently $H(Y) - H(Y|X, \tau_d) < H(Y) - H(Y|X, \tau^*)$, which satisfies inequality (D 1).

Lastly, if $\tau^*$ and $\tau_d$ are both elements drawn at random from $\mathcal{T}_d$ (in other words, neither was fixed because of its resulting value of $K$), then their respective effects will both correspond, on the average, to the expected value of the set:

$$H(Y) - H(Y|X, \tau_d) = H(Y) - H(Y|X, \tau^*) = H(Y) - \sum_{\tau_i \in \mathcal{T}_d} \Pr\{T_d = \tau_i\} H(Y|X, T_d = \tau_i), \qquad (D\,8)$$

which, on the average, would meet condition (D 1) as it entails equality (no decline in $K$). In practice, the difference in $K$ between two randomly chosen $\tau^*$ and $\tau_d$ would be randomly distributed around the value of zero.

The case of $\tau^*$ and $\tau_d$ being random elements, however, is again generally implausible and unrealistic. In the most probable scenario, a $\tau$ was selected because it optimized the value of $K$ in specific conditions. If those conditions remain and the $\tau$ is altered, then the default assumption must be that the corresponding $K$ will be lower.

This assumption of random differences is a rarely questioned standard in statistical modelling. In meta-analysis, for example, between-study heterogeneity is assumed to be random and normally distributed, which translates into assuming that the variance of effects produced by methodologically heterogeneous studies is symmetrically distributed around a true underlying effect [79]. However, examined from the perspective of how methods are developed to produce knowledge, a random distribution of between-study differences does not appear to be the most likely, indeed the most realistic, assumption.

The logic above can be extended to the case in which the two $\tau$ components do not have the same description length. In particular, let $\tau_d$ represent a theory/methodology of shorter description length, $-\log p(\tau^*) = -\log p(\tau_d) + d$, and let $T_d$ be an RV with alphabet $\mathcal{T}_d : \{\tau_1, \tau_2 \ldots \tau_d\}$ representing the set of all possible theories that have distance $d$ from $\tau^*$. Then inequality (D 1) can be re-arranged as

$$\frac{H(Y) + H(X) - \log p(\tau_d) + d}{H(Y) + H(X) - \log p(\tau_d)} \leq \frac{H(Y) - H(Y|X, T_d = \tau^*)}{H(Y) - \sum_{\tau_i \in \mathcal{T}_d} \Pr\{T_d = \tau_i\} H(Y|X, T_d = \tau_i)}, \qquad (D\,9)$$

which leads to the condition

$$d \leq \frac{E[H(Y|X, T_d)] - H(Y|X, T_d = \tau^*)}{K(Y; X, T_d)} \qquad (D\,10)$$

in which $E[H(Y|X, T_d)] = \sum_{\tau_i \in \mathcal{T}_d} \Pr\{T_d = \tau_i\} H(Y|X, T_d = \tau_i)$ is the expected value of the residual entropy across every possible specification of the $\tau$. Since $d > 0$, the inequality will not be satisfied if $E[H(Y|X, T_d)] \leq H(Y|X, T_d = \tau^*)$, i.e. $\tau^*$ yields a larger residual entropy than the average element in $\mathcal{T}_d$. However, as argued above, this is the least likely scenario, as it would presuppose that the original and longer theory/methodology $\tau^*$ had not been selected because it generated a relatively large $K$ value.

## D.2. Statement (ii)

With regard to divergences in the explanandum, the statement follows from the recursive validity of the DPI. The statement entails that

$$
\begin{aligned}
\lambda &= \log \frac{H(Y_{d+1}) + H(X) - \log p(\tau)}{H(Y_d + H(X) - \log p(\tau)} - \log \frac{H(Y_{d+1}) - H(Y_{d+1}|X, \tau)}{H(Y_d) - H(Y_d|X, \tau)} \\
&= \log \frac{H(Y_{d+1}) + H(X) - \log p(\tau)}{H(Y_d + H(X) - \log p(\tau)} - \log \frac{H(Y_d) - H(Y_d|X, \tau) - (H(Y_d|Y_{d+1}) - H(Y_d|Y_{d+1}, X, \tau))}{H(Y_d) - H(Y_d|X, \tau)} \\
&= \log \frac{H(Y_{d+1}) + H(X) - \log p(\tau)}{H(Y_d + H(X) - \log p(\tau)} - \log \left(1 - \frac{I(Y_d; X, \tau|Y_{d+1})}{I(Y_d; X, \tau)}\right).
\end{aligned}
\tag{D 11}
$$

Therefore, $\lambda$ is a constant as long as the proportional loss of mutual information and/or the increase in entropy of $Y$ is constant. As before, whereas there may be peculiar circumstances in which this is not the case, in general a proportional change follows from assuming that the loss is due to genuine noise.

Indeed, exponential curves describe how a Markov chain reaches a steady state [113]. Exponential curves are also used to model the evolution of chaotic systems. A system is said to be chaotic when it is highly sensitive to initial conditions. Since accuracy of measurement of initial states is limited, future states of the system become rapidly unpredictable even when the system is seemingly simple and deterministic. Paradigmatic chaotic systems, such as the three-body problem or the Lorenz weather equations, share the characteristics of being strikingly simple and yet are extremely sensitive to initial conditions, which make their instability particularly notable [114,115].

In standard chaos theory, the rapidity with which a system diverges from the predicted trajectory is measured by an exponential function in the form:

$$
\frac{d_N}{d_0} \approx e^{\lambda N},
\tag{D 12}
$$

in which $d_N/d_0$ is the relative offset after N steps (i.e. recalculations of the state of the system), and $\lambda$ is known as Liapunov exponent, a parameter that quantifies sensitivity of the system to initial conditions. Positive Liapunov exponents correspond to a chaotic system, and negative values correspond to stable systems, i.e. systems that are resilient to perturbation. There is a clear analogy between Liapunov exponents and $\lambda$ in equation (2.29), but the two are not equivalent, and the relationship between chaos theory and $K$ theory remains to be explored in future research.

The argument for a proportionality between the divergence $d$ in $\tau$ and the corresponding decline of $K$ is weaker, although rather intuitive. As already argued when formulating the theoretical argument for $K$, the larger the set $\mathcal{T}_b$ of possible theories, the lower the expected value of $K$ in the set, $K(Y, X, T_b)$, because most of the theories/methodologies in the set are likely to be nonsensical and yield $K \approx 0$. Therefore, at least in very general terms, the relation of equation (2.29) holds for divergences in $\tau$ as well.

The argument in this case is weaker because the relation between the divergence of $\tau$, $d_\tau$ and $K(Y; X, \tau_d)$ is likely to be complex and idiosyncratic. For any given $d$, multiple different $\tau_d$ are possible. For example, if one binary choice in $\tau$ is missing from $\tau_d$, then $d = 1$ but the values of $K(Y; X, \tau_d)$ can vary greatly, from being approximately identical to $K$ to being approximately zero, depending on what element of the methodology is missing. Mathematically, this fact can be expressed by allowing different values of $\lambda$ for any given distance. These values may be specific to a system and may need to be estimated on a case-by-case basis.

Therefore, to allow practical applications, the relationship between $K$ and $d_\tau$ is best modelled as the inner product of two vectors, e.g. $\lambda \cdot d = d_Y \lambda_Y + d_{\tau 1} \lambda_{\tau 1} + d_{\tau 2} \lambda_{\tau 2} + \cdots + d_{\tau l} \lambda_{\tau l}$, in which $\lambda = \lambda_{\tau 1} + \lambda_{\tau 2} + \cdots + \lambda_{\tau l}$ contains empirically derived estimates of the impact that distances of specific elements of the theory/methodology have on $K$. Extending this model to divergences in explanandum and input leads to the general formulation of equation (2.29).

# Appendix E

Let $X^\alpha$ be an RV quantized to resolution (i.e. bin size, or accuracy) $\alpha$, and let $a \in \mathbb{N}$ be the size of the alphabet of $X$, such that $\alpha = 1/a$. At no cost to generality, let an increase of resolution consist in the progressive sub-partitioning of $\alpha$, such that $\alpha' = \alpha/q$ with $q \in \mathbb{N}$, $q \geq 2$. Then

$$0 < H(X^{\alpha'}) - H(X^\alpha) \leq \log(q). \tag{E 1}$$

*Proof.* If $H(X^\alpha) = -\sum_1^a p(x) \log p(x)$, with $x$ representing any one of the $a$ partitions, then $H(X^{\alpha'}) = -\sum_1^{a \times q} p(x') \log p(x') = -\sum_1^a \sum_1^q p(a)p(q|a) \log[p(a)p(q|a)] \equiv H(A) + H(Q|A)$, where $Q$ and $A$ are the random variables resulting from the partitions. Known properties of entropy tell us that the entropy produced by the $q$-partition of $\alpha$ is smaller or equal to the logarithm of the number $q$ of partitions with equality if and only if the $q$-partitions of $\alpha$ have all the same probability, i.e. $H(Q|A) \leq \log q$. $\qquad\square$

## E.1. Definition: maximal resolution

Let $X^\alpha$ be a generic quantized random variable with resolution $\alpha$, and let $\alpha' = \alpha/q$ represent a higher resolution. The measurement error of $X^\alpha$ is a quantity $e > 0$, $e \in \mathbb{Q}$ such that:

$$H(X^{\alpha'}) - H(X^\alpha) = \log(q), \quad \forall \alpha \leq e \tag{E 2}$$

## E.2. Definition: empirical system

A system is said to be empirical if the quantization of explanandum and input has a maximal resolution. Equivalently, a non-empirical, (i.e. logico-deductive) system is a system for which $e = 0$.

The effect that a change in resolution has on $K$ depends on the characteristic of the system, and in particular on the speed with which the entropy of the explanandum and/or explanans increase relative to their joint distribution.

For every empirical system for which there is a $\tau \neq \emptyset$ such that $K(Y; X, \tau) > 0$, the system's quantization $Y^{\alpha_Y}$, $X^{\alpha_X}$ has optimal values of resolution $\alpha_y^*$ and $\alpha_x^*$ such that:

$$K(Y^{\alpha_Y^*}; X^{\alpha_X^*}, \tau) > K(Y^{\alpha_Y}; X^{\alpha_X}, \tau), \quad \forall \alpha_Y \neq \alpha_Y^*, \ \alpha_X \neq \alpha_X^* \tag{E 3}$$

*Proof.* If $\alpha$ is the resolution of $Y$ and $\alpha' = \alpha/q$ is a higher resolution then, assuming for simplicity that $\tau$ is constant:

$$K(Y^{\alpha'}; X, \tau) > K(Y^\alpha; X, \tau) \Leftrightarrow \frac{H(Y^{\alpha'}) - H(Y^{\alpha'}|X, \tau)}{H(Y^\alpha) - H(Y^\alpha|X, \tau)} > \frac{H(Y^{\alpha'}) + H(X) + \tau}{H(Y^\alpha) + H(X) + \tau}. \tag{E 4}$$

From equation (E 1), we know that $H(Y^{\alpha'}) \leq H(Y^\alpha) + \log(q)$, assuming equality and re-arranging equation (E 4) we get the condition:

$$H(Y^{\alpha'}|X, \tau) - H(Y^\alpha|X\tau) < (1 - K(Y^\alpha; X, \tau)) \log(q), \tag{E 5}$$

which is only satisfied when $K(Y^\alpha; X, \tau)$ is small and $H(Y^{\alpha'}|X, \tau) - H(Y^\alpha|X, \tau) \ll \log(q)$.

The corresponding condition for $X$ is

$$H(Y|X^{\alpha'}, \tau) - H(Y|X^\alpha, \tau) < -\log(q)K(Y; X^\alpha, \tau), \tag{E 6}$$

where the left-hand side has a lower bound in $-H(Y|X^\alpha, \tau)$, whereas the right-hand side can be arbitrarily negative.

Combining equations (E 4) and (E 6) yields the general condition:

$$K(Y^{\alpha_Y/q_Y}; X^{\alpha_X/q_X}, \tau) > K(Y^{\alpha_Y}; X^{\alpha_X}, \tau)$$
$$\Leftrightarrow H(Y^{\alpha_Y/q_Y}|X^{\alpha_X/q_X}, \tau) - H(Y^{\alpha_Y}|X^{\alpha_X}, \tau) < (1 - K) \log q_Y - K \log q_X, \tag{E 7}$$

in which $K \equiv K(Y^{\alpha_Y}; X^{\alpha_X}, \tau)$. The left-hand side of equation (E 7) is bounded between $H(Y^{\alpha_Y})$ and $-H(Y^{\alpha_Y}|X^{\alpha_X}, \tau)$, whereas the right-hand side is bounded between $\log q_Y$ when $K = 0$ and $-\log q_X$ when $K = 1$.

The only scenario in which $K$ never ceases to grow with increasing resolution entails $e = 0$ and thus a non-empirical system (definition (E 2)). $\square$

## Appendix F

*Proof.* To simplify the notation, we will posit that the explanans is expanded by two positive elements $H(X')$ and $-\log p(\tau')$.

$$
K(Y^{n_Y}; X^{n_X}, X'^{n_X}, \tau, \tau') > K(Y^{n_Y}; X^{n_X}, \tau)
$$
$$
\rightarrow \frac{n_Y H(Y) - n_Y H(Y|X, X', \tau, \tau')}{n_Y H(Y) + n_X H(X) + n_X H(X') - \log p(\tau) - \log p(\tau')}
$$
$$
> \frac{n_Y H(Y) - n_Y H(Y|X, \tau)}{n_Y H(Y) + n_X H(X) - \log p(\tau)}
$$
$$
\rightarrow (n_Y H(Y) + n_X H(X) - \log p(\tau))(n_Y H(Y) - n_Y H(Y|X, X', \tau, \tau'))
$$
$$
- (n_Y H(Y) + n_X H(X) + n_X H(X') - \log p(\tau) - \log p(\tau'))
$$
$$
\times (n_Y H(Y) - n_Y H(Y|X, \tau)) > 0
$$
$$
\rightarrow (n_Y H(Y) + n_X H(X) - \log p(\tau))(n_Y H(Y|X, \tau) - n_Y H(Y|X, X', \tau, \tau'))
$$
$$
> (n_X H(X') - \log p(\tau'))(n_Y H(Y) - n_Y H(Y|X, \tau))
$$
$$
\rightarrow n_Y H(Y|X, \tau) - n_Y H(Y|X, X', \tau, \tau') > (n_X H(X') - \log p(\tau'))K(Y; X, \tau)
$$
$$
\rightarrow (n_Y H(Y) - n_Y H(Y|X, X', \tau, \tau')) - (n_Y H(Y) - n_Y H(Y|X, \tau))
$$
$$
> (n_X H(X') - \log p(\tau'))K(Y; X, \tau)
$$
$$
\rightarrow k' - k > \frac{n_X H(X') - \log p(\tau')}{n_Y H(Y)} kh. \tag{F 1}
$$

$\square$

The same result would be derived if $H(X') \equiv \Delta H(X)$ and $-\log p(\tau') \equiv -\Delta \log p(\tau)$ represented any difference in size, positive or negative, between two explanantia.

## Appendix G

*Proof.*

$$
\begin{aligned}
K(Y; X,T) &= \frac{h}{H(Y)}(H(Y) - H(Y|X,T)) = \frac{h}{H(Y)}(H(Y) - H(Y,X,T) + H(X,T)) \\
&= \frac{h}{H(Y)}(H(Y) - H(Y,X,T) + H(X) + H(T)) = \frac{h}{H(Y)}(H(T) - (H(Y,X,T) - H(Y) - H(X)) \\
&= \frac{h}{H(Y)}(H(T) - H(T|Y,X)) = K(T; Y, X).
\end{aligned} \tag{G 1}
$$

$\square$

## Appendix H

*Proof.* Let $T$ be a random variable (RV) of alphabet $\mathcal{T} = \{\tau_1, \tau_2 \ldots \tau_z\}$, probability distribution $p(\tau)$ and entropy $H(T) = -\sum_i p(\tau_i) \log p(\tau_i)$. Let $T'$ be an RV derived from $T$ by removing from its alphabet the element $\tau_j \in \mathcal{T}$ of probability $p(\tau_j)$. Then

$$
H(T') = \frac{1}{1 - p(\tau_j)} \sum_{i \neq j} p(\tau_i) \log \frac{1}{p(\tau_i)} - \log \frac{1}{1 - p(\tau_j)}. \tag{H 1}
$$

When $|\mathcal{T}| = 2$, $H(T') = 0$ regardless of the probability distribution of $T$. Otherwise, the value rapidly approaches $H(T)$ as $p(\tau_j)$ decreases (e.g. as the alphabet of $T$ increases in size). Note that under specific conditions $H(T') > H(T)$—for example, if $T$ equals $p(\tau_j) = 0.9$, $p(\tau_k) = 0.05$, $P(\tau_k) = 0.05$. This entails that the uncertainty about a condition might momentarily increase if the most probable case is excluded. However, the effect is circumscribed since, as more elements are removed from the alphabet, $H(T')$ tends to 0. $\square$

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
