## [Reviewer comments · Royal Society Open Science]

Review History

RSOS-181055.R0 (Original submission)

Review form: Reviewer 1 (David Grimes)

Is the manuscript scientifically sound in its present form?

Yes

Are the interpretations and conclusions justified by the results?

Yes

Is the language acceptable?

Yes

Is it clear how to access all supporting data?

Yes

Do you have any ethical concerns with this paper?

No

Have you any concerns about statistical analyses in this paper?

Yes

Recommendation?

Major revision is needed (please make suggestions in comments)

Comments to the Author(s)

Comments are attached in PDF file with this review (Appendix A).

Review form: Reviewer 2 (Jonathan Schooler)

Is the manuscript scientifically sound in its present form?

Yes

Are the interpretations and conclusions justified by the results?

Yes

Is the language acceptable?

Yes

Is it clear how to access all supporting data?

Not Applicable

Do you have any ethical concerns with this paper?

No

Have you any concerns about statistical analyses in this paper?

I do not feel qualified to assess the statistics

Recommendation?

Accept with minor revision (please list in comments)

Comments to the Author(s)

See attached file (Appendix B).

Decision letter (RSOS-181055.R0)

12-Oct-2018

Dear Dr Fanelli,

The editors assigned to your paper ("A theory and methodology to quantify knowledge") have now received comments from reviewers. We would like you to revise your paper in accordance with the referee and Associate Editor suggestions which can be found below (not including confidential reports to the Editor). Please note this decision does not guarantee eventual acceptance.

Please submit a copy of your revised paper before 04-Nov-2018. Please note that the revision deadline will expire at 00.00am on this date. If we do not hear from you within this time then it will be assumed that the paper has been withdrawn. In exceptional circumstances, extensions may be possible if agreed with the Editorial Office in advance. We do not allow multiple rounds of revision so we urge you to make every effort to fully address all of the comments at this stage. If deemed necessary by the Editors, your manuscript will be sent back to one or more of the original reviewers for assessment. If the original reviewers are not available, we may invite new reviewers.

- Data accessibility

<http://datadryad.org/submit?journalID=RSOS&manu=RSOS-181055>

- Competing interests

- Authors' contributions

All submissions, other than those with a single author, must include an Authors' Contributions section which individually lists the specific contribution of each author. The list of Authors

should meet all of the following criteria; 1) substantial contributions to conception and design, or acquisition of data, or analysis and interpretation of data; 2) drafting the article or revising it critically for important intellectual content; and 3) final approval of the version to be published.

- Acknowledgements

- Funding statement

Please note that Royal Society Open Science charge article processing charges for all new submissions that are accepted for publication. Charges will also apply to papers transferred to Royal Society Open Science from other Royal Society Publishing journals, as well as papers submitted as part of our collaboration with the Royal Society of Chemistry (<http://rsos.royalsocietypublishing.org/chemistry>). If your manuscript is newly submitted and subsequently accepted for publication, you will be asked to pay the article processing charge, unless you request a waiver and this is approved by Royal Society Publishing. You can find out more about the charges at <http://rsos.royalsocietypublishing.org/page/charges>. Should you have any queries, please contact openscience@royalsociety.org.

on behalf of Prof. Mark Chaplain (Subject Editor)
openscience@royalsociety.org

Comments to Author:

Reviewers' Comments to Author:

Reviewer: 1

Comments to the Author(s)

Comments are attached in PDF file with this review.

Reviewer: 2

Comments to the Author(s)
See attached file

Author's Response to Decision Letter for (RSOS-181055.R0)

See Appendix C.

RSOS-181055.R1 (Revision)

Review form: Reviewer 1 (David Grimes)

Is the manuscript scientifically sound in its present form?

Yes

Are the interpretations and conclusions justified by the results?

Yes

Is the language acceptable?

Yes

Is it clear how to access all supporting data?

Yes

Do you have any ethical concerns with this paper?

No

Have you any concerns about statistical analyses in this paper?

No

Recommendation?

Accept as is

Comments to the Author(s)

I have very little to add since my last review - apologies for my tardiness. The author has kindly addressed my major questions, and I am satisfied with the response. The examples are very useful, and I wonder if there's merit in applying these more early on, but this is a minor point given the manuscript is well laid out, but highly information dense. I am satisfied the paper will make a valuable addition to literature.

Review form: Reviewer 2 (Jonathan Schooler)

Is the manuscript scientifically sound in its present form?

Yes

Are the interpretations and conclusions justified by the results?

Yes

Is the language acceptable?

Yes

Is it clear how to access all supporting data?

Yes

Do you have any ethical concerns with this paper?

No

Have you any concerns about statistical analyses in this paper?

I do not feel qualified to assess the statistics

Recommendation?

Accept with minor revision (please list in comments)

Comments to the Author(s)

See attached (Appendix D).

Decision letter (RSOS-181055.R1)

18-Jan-2019

Dear Dr Fanelli:

On behalf of the Editors, I am pleased to inform you that your Manuscript RSOS-181055.R1 entitled "A theory and methodology to quantify knowledge" has been accepted for publication in Royal Society Open Science subject to minor revision in accordance with the referee suggestions. Please find the referees' comments at the end of this email.

The reviewers and Subject Editor have recommended publication, but also suggest some minor revisions to your manuscript. Therefore, I invite you to respond to the comments and revise your manuscript.

- Ethics statement

- Data accessibility

It is a condition of publication that all supporting data are made available either as supplementary information or preferably in a suitable permanent repository. The data

accessibility section should state where the article's supporting data can be accessed. This section should also include details, where possible of where to access other relevant research materials such as statistical tools, protocols, software etc can be accessed. If the data has been deposited in an external repository this section should list the database, accession number and link to the DOI for all data from the article that has been made publicly available. Data sets that have been deposited in an external repository and have a DOI should also be appropriately cited in the manuscript and included in the reference list.

If you wish to submit your supporting data or code to Dryad (<http://datadryad.org/>), or modify your current submission to dryad, please use the following link:
<http://datadryad.org/submit?journalID=RSOS&manu=RSOS-181055.R1>

- **Competing interests**

- **Authors' contributions**

- **Acknowledgements**

- **Funding statement**

Because the schedule for publication is very tight, it is a condition of publication that you submit the revised version of your manuscript before 27-Jan-2019. Please note that the revision deadline will expire at 00.00am on this date. If you do not think you will be able to meet this date please let me know immediately.

To revise your manuscript, log into <https://mc.manuscriptcentral.com/rsos> and enter your Author Centre, where you will find your manuscript title listed under "Manuscripts with Decisions". Under "Actions," click on "Create a Revision." You will be unable to make your

revisions on the originally submitted version of the manuscript. Instead, revise your manuscript and upload a new version through your Author Centre.

on behalf of Prof Mark Chaplain (Subject Editor)
openscience@royalsociety.org

Editor comments:

The referees have recommended acceptance, though one has suggested a few minor tweaks. Please incorporate and respond to these changes where possible, and submit your revision for consideration. Thanks for the support.

Reviewer comments to Author:

Reviewer: 1

Comments to the Author(s)

I have very little to add since my last review - apologies for my tardiness. The author has kindly addressed my major questions, and I am satisfied with the response. The examples are very useful, and I wonder if there's merit in applying these more early on, but this is a minor point given the manuscript is well laid out, but highly information dense. I am satisfied the paper will make a valuable addition to literature.

Reviewer: 2

Comments to the Author(s)

See attached.

Author's Response to Decision Letter for (RSOS-181055.R1)

See Appendix E.

Decision letter (RSOS-181055.R2)

06-Feb-2019

Dear Dr Fanelli,

I am pleased to inform you that your manuscript entitled "A theory and methodology to quantify knowledge" is now accepted for publication in Royal Society Open Science.

Kind regards,
Royal Society Open Science Editorial Office
Royal Society Open Science

on behalf of Professor Mark Chaplain (Subject Editor)
openscience@royalsociety.org

Appendix A

Review notes

October 2nd 2018

General comments

This work is fascinating, but perhaps the most difficult paper I've ever had to review. If I have understood correctly, the scope of Dr. Fanelli's ideas are wide-ranging and of fundamental importance. However, this is a massive work, and I admit there are parts I found difficult to follow. I think if clarity could be improved, the manuscript would be much more beneficial. It is entirely possible too I have misunderstood aspects of the work at hand – while I am familiar with Shannon's theorem and AIT, I am certainly no expert and so I offer my comments in this vein, and could be very grateful if these could be addressed.

Abstract

The abstract is a powerful statement of intent, and relatively clear – but QRP should be defined use.

Introduction section (a)

The initial introduction section is very clearly written, and I am happy this could be broadly followed by any reader. I will break down minor criticisms of the subsequent sections here

1. Shannon's entropy function is introduced quite rapidly in 2.1, but I think it requires a little more elaboration. I think if I understand correctly, the formulation being invoked is the Shannon entropy, typically given by

$$H(X) = - \sum_{i=1}^N P(x_i) \log_b P(x_i)$$

rewritten here in terms of Y , with the log base not specified (but presumably dependent on number of possible states). It might be worth explaining what this means to the reader first, potentially with a simple coin flip example, for as the author himself points out, AIT and information theory are not commonly used in meta-research, and to encourage use requires demystification so that others may see the merit of this approach.

2. RV is random variable presumably, but this should be defined.
3. My major issue with this section is that I do not see clearly how the pivotal equation 2.1 was arrived at, and from whence it came. I do not dispute it's rectitude, but I think as it's so fundamental to the work, it's origin needs to be made clearer.

Introduction section (b)(i)

1. Is information still considered finite if something is entirely random? Say a radioactive decay. I think it still is (it would just have very high entropy) but I wanted to check I understood correctly.

2. In figure 1, (a), (b), (e) and (f) are functions of a single variable, X. But 1(c) is a function of two variables (ie: it's always multi-valued) as is the even more complex 1(d). Can we still straightforwardly apply the framework here with multiple dependence? Apologies if I'm misunderstanding this.
3. My previous question might stem from a poor understanding the mutual information theorem – as I understand it, MIT measures information shared by X and Y. In a circle above, X and Y are not independent, so perhaps that's fine. It would be worth clarifying this in the text, as this will not be known to the typical meta-researcher.

Introduction section (b)(ii)

1. F and g here are not clearly defined; are these functions?
2. Figure 3: This is hard to interpret as I'm still unsure of where equation 2.1 arrived from, and what values of K describe impressive 'knowledge' versus only middling. For example, the coefficient of determination in the first figure suggests that for perfect correlation, K is still small (~0.12): I think this might be easier to understand if the formal definition was clarified further.

Introduction section (b)(iii)

1. This alludes to the derivation of 2.1 – please make clearer. The rest of the section I quite like, but there is some risk of tautology – equation 2.16 taken to limits of infinity yields a K of 0, independent of tau, which is presented as a further argument for the postulate. But that is only the case if we're confident that K is well-derived. And again, while I am confident it is, I'd like to see it's derivation clearly presented so I might better understand.

section (c)(v)

1. The argument for the exponential form of 2.25 is quite technical, and not immediately clear (S6).

section 3(b)(i)

1. The electron mass is taken from NIST data to be $9.10938356 \times 10^{-31} \pm 0.00000011 \times 10^{-31}$; I am not sure where the current figure is from, or how the uncertainty is being reported?

Figure 1 - Simple examples of data points with no error perfectly described by an analytical relationship between x and y - should these not have higher K than that suggested in figure 3?

section 3(b) (General)

The K 's here seem much higher than the K 's in figure 3. This is causing me some confusion; if I have a perfected R^2 , it might quite likely be because I know the precise relationship between two things. For example, above. Shouldn't these two have high K scores?

Over all notes

I think this work is exceptionally promising, but as it stands I am still finding it difficult to ascertain precisely how the pivotal relationship asserted was arrived at. Consequently I feel I would be in a better place to interpret the work if this question (and the ones I have arrived at here) could be answered.

DRG

Appendix B

Summary

This highly ambitious manuscript attempts to provide a general theory of information as it applies to science that, the author hopes, can serve as a foundational core for a theoretical approach to meta-research/meta-science (the science of science). Drawing on information theory this manuscript introduces the broadly applicable variable of K entailing knowledge associated with a system, in particular the explanatory power that a body of evidence provides in any given domain. K has many similarities to more familiar constructs such as measures of effect sizes but the author argues has many advantages including being independent of measurement form, and sensitive to various factors such as sample size, accuracy of scale, and Ockham's razor. The author carefully explicates the central assumptions of the model, delineates its mathematical formulation and then goes on to illustrate its potential utility with respect to a number of pressing questions in meta-science. These include: determining how much knowledge is contained in a theoretical system, characterizing the amount of personality information gained from learning of a person's gender, providing an alternative to meta-analysis in assessing the cumulative meaning of multiple studies, evaluating reproducibility research, establishing the value of null results, characterizing pseudoscience, and determining what makes a science "soft". The paper concludes that this basic approach provides a foundational meta-theory for meta-science.

Evaluation

I must acknowledge at the outset that although I am familiar with information theory I am nowhere near sufficiently well versed with it, nor mathematically skilled enough to evaluate the details of its treatment here. The author could very well have some major flaws in his logic or math that would completely escape me. I thus find myself in the peculiar position of being extremely enthusiastic about a paper that I cannot claim to entirely understand. I do however understand the reason for my enthusiasm. The author has taken on the daunting task of trying to quantitatively formalize precisely what it is that we extract from science, and then to delineate how this formal approach can illuminate the nascent yet burgeoning field of meta-science. The general idea of using information theory to illuminate the knowledge that we extract from science strikes me as extremely promising and hitherto under explored, and the many distinct domains that the theory

is applied to is startling both in its breadth of topics and detail of analysis. I genuinely believe that this paper has the prospect of being truly seminal.

Although I am genuinely excited about this paper I feel that it is important that you find some reviewers who are sufficiently well versed in information theory that they can carefully work through all of the equations, and the details of their applications in the various domains. My expectation is that the math works, but I worry that there are additional assumptions embedded in the formalisms that the author may have neglected addressing. The devil is in the details, and I am concerned that there may be other assumptions (besides the two the author carefully defends) that might make this approach less palatable.

If, as I hope, some version of this work is ultimately published, given its remarkable breadth and potential significance, I would encourage the editors to invite several commentaries on it. I personally would not feel qualified to write a commentary, but I could happily suggest individuals who could.

Jonathan Schooler

I noticed a number of typos throughout the manuscript that I am confident would be caught by a good copy editor.

Appendix C

18 November 2018

Manuscript ID RSOS-181055

Dear Madame/Sir,

I thank both referees for the careful assessment of the manuscript, please find below my detailed responses to their comments.

Kind regards,
Daniele Fanelli

Reviewer 1 (Jonathan Schooler):

Summary

This highly ambitious manuscript attempts to provide a general theory of information as it applies to science that, the author hopes, can serve as a foundational core for a theoretical approach to meta-research/meta-science (the science of science). Drawing on information theory this manuscript introduces the broadly applicable variable of K entailing knowledge associated with a system, in particular the explanatory power that a body of evidence provides in any given domain. K has many similarities to more familiar constructs such as measures of effect sizes but the author argues has many advantages including being independent of measurement form, and sensitive to various factors such as sample size, accuracy of scale, and Ockham's razor. The author carefully explicates the central assumptions of the model, delineates its mathematical formulation and then goes on to illustrate its potential utility with respect to a number of pressing questions in meta-science. These include: determining how much knowledge is contained in a theoretical system, characterizing the amount of personality information gained from learning of a person's gender, providing an alternative to meta-analysis in assessing the cumulative meaning of multiple studies, evaluating reproducibility research, establishing the value of null results, characterizing pseudoscience, and determining what makes a science "soft". The paper concludes that this basic approach provides a foundational meta-theory for meta-science.

I thank the referee for a careful and accurate characterization of the manuscript's scope and content.

Evaluation

I must acknowledge at the outset that although I am familiar with information theory I am nowhere near sufficiently well versed with it, nor mathematically skilled enough to evaluate the details of its treatment here. The author could very well have some major flaws in his logic or math that would completely escape me. I thus find myself in the peculiar position of being extremely enthusiastic about a paper that I cannot claim to entirely understand. I do however understand the reason for my enthusiasm. The author has taken on the daunting task of trying to quantitatively formalize precisely what it is that we extract from science, and then to delineate how this formal approach can illuminate the nascent yet burgeoning field of meta- science. The general idea of using information theory to illuminate the knowledge that we extract from science strikes me as extremely promising and hitherto under explored, and the many distinct domains that the theory is applied to is startling both in its breadth of topics and detail of analysis. I genuinely believe that this paper has the prospect of being truly seminal.

Although I am genuinely excited about this paper I feel that it is important that you find some reviewers who are sufficiently well versed in information theory that they can carefully work through all of the equations, and the details of their applications in the various domains. My expectation is that the math works, but I worry that there are additional assumptions embedded in the formalisms that the author may have neglected addressing. The devil is in the details, and I am concerned that there may be other assumptions (besides the two the author carefully defends) that might make this approach less palatable.

I appreciate the reviewer's positive assessment of the potential of the manuscript, and I understand his concerns about the consistency of the mathematics. Reviewer 2 has taken on this task to a remarkable extent.

However, I welcome the suggestion of additional mathematical review and RSOS' editors would be very welcome to solicit an additional reviewer purely to check the soundness of the mathematics, to any extent that they deem this appropriate.

I noticed a number of typos throughout the manuscript that I am confident would be caught by a good copy editor.

I thank the reviewer for pointing this out. The manuscript was revised in some parts, and I hope to have caught a fair number of typos. I will be pleased, if and when the manuscript is accepted, to have it checked further for typos.

Reviewer 2:

This work is fascinating, but perhaps the most difficult paper I've ever had to review. If I have understood correctly, the scope of Dr. Fanelli's ideas are wide-ranging and of fundamental importance. However, this is a massive work, and I admit there are parts I found difficult to follow. I think if clarity could be improved, the manuscript would be much more beneficial. It is entirely possible too I have misunderstood aspects of the work at hand – while I am familiar with Shannon's theorem and AIT, I am certainly no expert and so I offer my comments in this vein, and could be very grateful if these could be addressed.

I thank the reviewer for the positive comments and the work that he/she has thus far put into examining my manuscript. I appreciate that its length and breadth may turn away readers, and I have extensively revised the manuscript to improve its clarity, both in the passages highlighted by the reviewers, as well as other passages.

Furthermore, I took the opportunity of this revision to add a few paragraphs, in particular:

-the discussion section has been structured around subsections, and includes a new section detailing testable predictions.

The abstract is a powerful statement of intent, and relatively clear – but QRP should be defined use.

The acronym has been removed from the abstract

It might be worth explaining what this means to the reader first, potentially with a simple coin flip example, for as the author himself points out, AIT and information theory are not commonly used in meta-research, and to encourage use requires demystification so that others may see the merit of this approach.

I appreciate this suggestion and have modified the relevant section by explaining in very simple and intuitive terms what information is, and what Shannon's entropy does.

Other parts of the text have been revised to ensure a broader accessibility.

RV is random variable presumably, but this should be defined.

The nature of the abbreviation is now explained at the start of the introduction.

My major issue with this section is that I do not see clearly how the pivotal equation 2.1 was arrived at, and from whence it came. I do not dispute it's rectitude, but I think as it's so fundamental to the work, it's origin needs to be made clearer.

The equation was originally derived following the "theoretical argument" presented in section b. This point has now been made clearer by adding: 1) further details in the last paragraph of the introduction (where the various sections of the manuscript are listed), 2) in section b, with sentences explaining that this was how the equation was arrived at.

Is information still considered finite if something is entirely random? Say a radioactive decay. I think it still is (it would just have very high entropy) but I wanted to check I understood correctly.

Yes, it would be. The reviewer's observation points to a genuine element of potential confusion (and a source of considerable

philosophical discussion): the distinction between the nature of phenomena and the nature of knowledge about those phenomena.

For the purposes of a theory of knowledge, what reality ultimately looks like is irrelevant, because what counts is how the world is represented to/by us.

The rate of radioactive decay, to be known, would need to be measured to some finite level of accuracy, thereby generating a finitely describable explanandum.

The description of this postulate has been modified to clarify this distinction (it now talks more explicitly about “representation”), and the relevant SI text has also been edited.

In figure 1, (a), (b), (e) and (f) are functions of a single variable, X. But 1(c) is a function of two variables (ie: it's always multi-valued) as is the even more complex 1(d). Can we still straight- forwardly apply the framework here with multiple dependence? Apologies if I'm misunderstanding this.

Yes, absolutely, although I appreciate the potential for confusion.

In the figure, the “r” of the circle radius is actually a constant, and so I replaced it with an actual value. That said, any number of variables, if combined, just produce another variable. This point was made clearer in the section that introduces the concept of information.

Moreover, some of the examples given in the results section show how multivalued functions (e.g. the reproductive skew example) are translated into K.

My previous question might stem from a poor understanding the mutual information theorem – as I understand it, MIT measures information shared by X and Y. In a circle above, X and Y are not independent, so perhaps that's fine. It would be worth clarifying this in the text, as this will not be known to the typical meta-researcher.

I may be misinterpreting the point, but the X and Y in all the figure's example are non-independent, except the first. The latter would be the

only case in which $I(Y;X)=0$. As mentioned above, I tried to make the concept of information and mutual information more accessible, and the reviewer is welcome to assess whether more background introductory material is necessary.

F and g here are not clearly defined; are these functions?

Yes, they were intended to represent generic (non-specified, but specifiable in principle) functions. The concept has now been explained in the relevant text. At the same time, parts of this section have been simplified and others (e.g. the SI about the Chi squared and that about Akaike's Information Criterion) have been removed to avoid distracting complications. These analyses are better left to future, dedicated articles.

Figure 3: This is hard to interpret as I'm still unsure of where equation 2.1 arrived from, and what values of K describe impressive 'knowledge' versus only middling. For example ,the coefficient of determination in the first figure suggests that for perfect correlation, K is still small (~0.12): I think this might be easier to understand if the formal definition was clarified further.

The value of K is "system-specific", in the sense that its specific value depends on numerous factors that, as was mentioned in the text corresponding to this figure, modulate the value of K. For example, for any given system, the R^2 can only vary between 0 and 1, depending on total and residual variances but the values of K, albeit always in one-to-one correspondence with the R^2 , will vary depending on the value of τ , n , and the entropy of explanandum (accuracy) and other factors.

The value of 0.12 in the future is due to the arbitrary combination of such values used to simulate the results. The details of such numbers are provided in the R code provided with the article.

In order to facilitate understanding, the figure caption has been re-created to have the same scale on the y axis, and the caption has been modified to explain what the value of τ is.

This alludes to the derivation of 2.1 – please make clearer. The rest of the section I quite like, but there is some risk of tautology – equation 2.16 taken to limits of infinity yields a K of 0, independent of tau, which is presented as a further argument for the postulate. But that is only the case if we’re confident that K is well-derived. And again, while I am confident it is, I’d like to see it’s derivation clearly presented so I might better understand.

As I hope the text now explains better, the equation is indeed derived from the same argument that this passage is referring to. i.e. the “theoretical” argument.

This is not the only argument for the finiteness of information. Another quantitative argument is given when discussing Ockham’s razor and when discussing the relation between K and resolution. More discursive arguments are also given in the SI.

The argument for the exponential form of 2.25 is quite technical, and not immediately clear (S6).

I thank the referee for paying attention to these aspects of the text, which are only apparently “supplementary”. In addition to being admittedly a little obscure, the passage also contained a few regrettable typos in the equations.

In addition to correcting the latter and generally revising the equations and texts for typos throughout the text, I have entirely re-written the section to make the argument more intuitive and pragmatic. The section is now much longer, because this concept is particularly important and deserves attention.

The electron mass is taken from NIST data to be $9.10938356 \times 10^{-31}$ +/- $0.00000011 \times 10^{-31}$; I am not sure where the current figure is from, or how the uncertainty is being reported?

The value reported came from the Particle Data Group 2014 edition <http://pdg.lbl.gov/2014/listings/rpp2014-list-electron.pdf>

However, I am happy to go with the recommended value in NIST, and I have amended the text accordingly. The corresponding K would be slightly higher, as the error in this estimation is smaller.

More importantly, however, I have now slightly changed the analysis: whereas before I was calculating the value for a nine-digit number, I decided that the most general case and in fact the most conservative approach ought to consider the entirety of the length of the number, which has 39 digits. The calculations have been amended accordingly.

I have also added some text in this example to emphasize how the value of K is “system specific” also in the sense that the scale of measurement used – being part of how a system is defined – will make a difference on the value of K .

The K's here seem much higher than the K's in figure 3. This is causing me some confusion; if I have a perfected R2, it might quite likely be because I know the precise relationship between two things. For example, above. Shouldn't these two have high K scores?

It would depend on how the system is specified, and consequently on the values of multiple factors.

Firstly, it would depend on the resolution, i.e. the number of partitions in which Y and X are measured, as this determines their entropies.

Second, it would depend on the value of n_Y and n_X .

Lastly, and most importantly, it would depend on what tau is assumed to consist in, and the language (alphabet, grammar) used to encode it.

In the example given, assuming the tau is solely represented by the mathematical relations, tau could be described as the number of alphanumeric characters of, say, eight bits each (256 bits).

Assuming the resolution and n values are the same in the two figures, the second curve would have a lower K compared to the first, because the description length of the tau is at least one bite longer. The longer

description length of course reflects the slightly higher complexity of a curve compared to a line.

This is also illustrated in the two examples of simulations with different levels of resolution, shown in S5 appendix.

This system-specific dependence is one of the key novelties of K. If K was identical to R2, then of course there would be no point in using it and not the latter.

I think this work is exceptionally promising, but as it stands I am still finding it difficult to ascertain precisely how the pivotal relationship asserted was arrived at. Consequently I feel I would be in a better place to interpret the work if this question (and the ones I have arrived at here) could be answered.

I am extremely grateful to the reviewer for the through attention paid in his/her assessment. I understand that this is an unusual paper and I shall be looking forward to receiving more invaluable feedback.

Appendix D

Fanelli Revised manuscript.

I liked this manuscript very much the first time I reviewed it but was concerned that my lack of mathematical depth precluded me from evaluating the nuances of its mathematical elements. I am encouraged by the deep and thoughtful review of the other referee, and the apparently thoughtful manner in which the author addresses the many issues raised. I feel confident that if the other reviewer approves of the revision that the technical arguments are sufficiently compelling to deserve vetting by the larger scientific community.

As I read this manuscript over with fresh eyes, several ways in which I could see it fruitfully developed came to mind. I will leave it to the editor whether these should be addressed in the current version or considered at a later date.

Anomalous cognition. It is easy to dispense with an area (e.g. astrology) for which there is only one credible study and it showed the domain had no merit whatsoever. But what about other areas of study that are often disparaged as pseudoscience but for which there are more regularly reported significant effects. Specifically, how would this approach evaluate domains of anomalous cognition (pre-cognition, psychokinesis and telepathy for which there is more, if admittedly still questionable evidence (see Schooler et al, 2018 for a list of meta-analyses). It would be interesting to see how this model handles the thorny issues of one of these domain.

Predictive coding- I was intrigued by the authors suggestion that this approach relates to much of cognition, and in particular the capacity to make predictions. Here it would be very interesting to learn how the author see his project relating to Friston, (e.g. Friston (2010) approach to predictive coding

Integrated information theory. The reliance on information theory to address the issues of meta-science has some parallels to its application to understanding consciousness. I am very curious about the relationship between this approach and that of IT theory of Tononi (e.g. Tononi's 2008).

Again I believe that the present work has already covered more than enough ground, though might be worthwhile briefly allude to some or all of these areas.

On page 21 progressively is misspelled

Jonathan Schooler

Cited refs

Friston K. (2010). The free-energy principle: a unified brain theory? *Nat. Rev. Neurosci.* 11, 127–138. 10.1038/nrn2787

Tononi, G (2008) "Consciousness as Integrated Information: a Provisional Manifesto," *The Biological Bulletin* 215, 216-242.

Schooler, J., Baumgart, S., Franklin, M. (2018). Entertaining Without Endorsing: The Case for the Scientific Investigation of Anomalous Cognition. *Psychology of Consciousness: Theory, Research, and Practice*, 5, 63–77

Appendix E

4 February 2019

Manuscript ID RSOS-181055

Dear Madame/Sir,

I again wish to thank both referees for their helpful comments, which I have taken into careful consideration and addressed as described below. I have also finalized the figures, reshaping some illustrations and adding clarifying titles.

Kind regards,
Daniele Fanelli

Reviewer 1

I have very little to add since my last review - apologies for my tardiness. The author has kindly addressed my major questions, and I am satisfied with the response. The examples are very useful, and I wonder if there's merit in applying these more early on, but this is a minor point given the manuscript is well laid out, but highly information dense. I am satisfied the paper will make a valuable addition to literature.

I thank the reviewer for helping improve the manuscript. Following the suggestion to present the examples earlier on, I have moved the table presenting the summary of results to the introduction, in order to give the reader an immediate sense of where the (admittedly dense) methodological section is heading. The table now includes a column with the section corresponding to each meta-scientific question, to point the reader to results and example that he/she may find of interest.

Reviewer 2 (Jonathan Schooler):

I liked this manuscript very much the first time I reviewed it but was concerned that my lack of mathematical depth precluded me from evaluating the nuances of its mathematical elements. I am encouraged by the deep and thoughtful review of the other referee, and the apparently thoughtful manner in which the author addresses the many issues raised. I feel confident that if the other reviewer approves of the revision that the technical arguments are sufficiently compelling to deserve vetting by the larger scientific community.

As I read this manuscript over with fresh eyes, several ways in which I could see it fruitfully developed came to mind. I will leave it to the editor whether these should be addressed in the current version or considered at a later date.

Anomalous cognition. It is easy to dispense with an area (e.g. astrology) for which there is only one credible study and it showed the domain had no merit whatsoever. But what about other areas of study that are often disparaged as pseudoscience but for which there are more regularly reported significant effects. Specifically, how would this approach evaluate domains of anomalous cognition (pre-cognition,

psychokinesis and telepathy for which there is more, if admittedly still questionable evidence (see Schooler et al, 2018 for a list of meta-analyses). It would be interesting to see how this model handles the thorny issues of one of these domain.

Predictive coding- I was intrigued by the authors suggestion that this approach relates to much of cognition, and in particular the capacity to make predictions. Here it would be very interesting to learn how the author see his project relating to Friston, (e.g. Friston (2010) approach to predictive coding

Integrated information theory. The reliance on information theory to address the issues of meta- science has some parallels to its application to understanding consciousness. I am very curious about the relationship between this approach and that of IT theory of Tononi (e.g. Tononi's 2008).

Again I believe that the present work has already covered more than enough ground, though might be worthwhile briefly allude to some or all of these areas.

On page 21 progressively is misspelled

I really appreciate the constructive suggestions for further developments of the manuscript. As the reviewer noticed, the essay is very long and there is no space to discuss the relation between the theories mentioned and K theory. Future work could explore the connections, but it should be emphasized that K theory differs from that of Friston and of Tononi both in its scope and in its mathematical structure. Therefore, I have added the following sentence in the discussion:

“Finally, K theory was developed independently from other recent attempts to give informational accounts of cognitive phenomena, for example the free-energy principle (e.g. [79]) and the integrated information theory of consciousness (e.g. [80]). Whereas these theories bear little resemblance to that proposed in this essay, they obviously share a common objective with it, and possible connections may be explored in future research.”

For what concerns the field of anomalous cognition, addressing its status is beyond the scope of this work. All I can say is that K theory offers the theoretical and empirical tools to assess the epistemological status of this and any other controversial research field.

The text has been proof-read once more, in the hope of catching other typographical errors, and further edited for clarity.